# Computationally efficient subglacial drainage modelling using Gaussian Process emulators: GlaDS-GP v1.0

Tim Hill[1], Derek Bingham[2], Gwenn E. Flowers[1], and Matthew J. Hoffman[3]

[1]Department of Earth Sciences, Simon Fraser University, Burnaby, BC, Canada
[2]Department of Statistics and Actuarial Science, Simon Fraser University, Burnaby, BC, Canada
[3]Fluid Dynamics and Solid Mechanics Group, Los Alamos National Laboratory, Los Alamos, NM, USA

**Correspondence:** Tim Hill (tim_hill_2@sfu.ca)

**Abstract.**

Subglacial drainage models represent water flow at the ice–bed interface through coupled distributed and channelized systems to determine water pressure, discharge and drainage system geometry. While they are used to understand processes such as the relationship between surface melt and ice flow, the number of uncertain model parameters and the computational cost of running models makes it difficult to adequately explore the high-dimensional parameter space and evaluate uncertainty in model predictions. Here, we develop Gaussian Process (GP) emulators that make fast predictions with associated uncertainty of subglacial drainage model outputs. Using a truncated principal component basis representation, we construct a GP emulator for diurnally averaged subglacial water pressure. We also explore emulation of scalar variables describing drainage efficiency and configuration. We train the emulators using ensembles of up to 512 simulations varying eight parameters of the Glacier Drainage System (GlaDS) model on a synthetic domain intended to represent an ice-sheet margin. The emulators make predictions $\sim$1000 times faster than GlaDS simulations, with error $< 3\%$ for the water pressure field and $\sim$5–9% for drainage efficiency and configuration. We apply the emulators to explore the eight-dimensional parameter space by computing variance-based parameter-sensitivity indices, finding that three parameters (ice-flow coefficient, bed bump aspect ratio and the subglacial cavity system conductivity) explain 90% of the variance in modelled water pressure in response to parameter changes. The GP emulator approach described here is well-suited to integrate observational data with models to make calibrated, credible predictions of subglacial drainage.

## 1 Introduction

Subglacial hydrologic processes influence a variety of local and downstream physical, chemical, and biological processes. Chemical weathering processes and the composition of subglacial microbial communities, and their influence on downstream environments, are modulated by water residence time (Crompton et al., 2015; Dubnick et al., 2017; Graly and Rezvanbehbahani, 2022), discharge rates (Prestrud Anderson et al., 1997; Hodson et al., 2008; Dubnick et al., 2017), the partitioning between channelized, distributed and disconnected portions of the subglacial drainage system (Hodson et al., 2008; Graly and Rezvanbehbahani, 2022; Dubnick et al., 2017), and the rate of mixing between water sourced from the surface and the bed (Sharp et al., 1999; Hodson et al., 2008; Crompton et al., 2015; Dubnick et al., 2017). Greenland fjord productivity is

25 influenced by buoyant subglacial meltwater plumes created by channelized discharge across the grounding line (e.g., Meire et al., 2017). Most commonly, studies of subglacial hydrology are motivated by the influence on ice-flow velocities of glaciers (e.g., Iken and Bindschadler, 1986; Mair et al., 2002) and ice sheets (e.g., Joughin et al., 2008; Moon et al., 2014) through effective-pressure-dependent basal slip.

Conventional physically based subglacial drainage models (e.g., Schoof, 2010; Werder et al., 2013; Sommers et al., 2018;
Hoffman et al., 2018) are computationally intensive, in part due to the short timescale associated with water flow driven by surface melt inputs that vary on sub-daily time scales and because models must resolve large spatiotemporal variations in drainage-system capacity and efficiency related to evolving drainage pathways. Their computational cost is compounded by uncertainty in model parameters and the necessarily limited subset of physics included in models. Field experiments and observations can be used to constrain the values of certain physical parameters, such as the friction factor in ice-walled channels
(e.g., Pohle et al., 2022; Werder and Funk, 2009) and channel size and sinuosity (e.g., Werder and Funk, 2009), infer plausible distributions of hydrology- and sliding-model parameters (e.g., Brinkerhoff et al., 2016, 2021), and learn about drainage system configuration (e.g., Irarrazaval et al., 2019, 2021; Rada and Schoof, 2018; Rada Giacaman and Schoof, 2023; Nanni et al., 2021; Gimbert et al., 2016). The high-dimensional space of uncertain parameters, combined with the cost of running model simulations, makes it difficult to adequately explore the input space to quantify the uncertainty in model outputs.

Emulators provide a fast approximation to physics-based models to overcome limitations related to long model runtimes. Emulation is gaining traction as a tool in the glaciological literature, for example, to speed up model simulations (e.g., Jouvet et al., 2022), compute probability distributions of sea-level change (e.g., Berdahl et al., 2021; Edwards et al., 2021), infer parameter values (e.g., Chang et al., 2014; Gilford et al., 2020; Wernecke et al., 2020) or parameterizations (e.g., Wernecke et al., 2020; Bolibar et al., 2020, 2023) and infer bed topography from surface observables (e.g., Jouvet, 2023). For subglacial
drainage modelling, neural network emulators have been used to calibrate subglacial drainage model and sliding law parameters given surface velocity data (Brinkerhoff et al., 2021) and to predict spatially distributed hydraulic potential given geometry (bed topography and ice thickness) and surface melt forcing for the purpose of providing improved basal boundary conditions for ice-sheet modelling (Verjans and Robel, 2024).

To overcome some of the limitations of subglacial drainage models identified above, and to increase the diversity of sub-
50 glacial drainage emulators, we explore Gaussian Process (GP) emulation of subglacial drainage model outputs. We choose to use GP emulation for its quantification of prediction uncertainty, where each emulator prediction is a probability distribution rather than a point estimate, and simplicity in terms of the number of parameters that must be estimated relative to a neural network. The Gaussian Process emulators we develop take subglacial drainage model parameters as their inputs and predict spatially and seasonally resolved flotation fraction (the ratio of water pressure to ice-overburden pressure) as well as
scalar descriptions of drainage configuration and efficiency. As an example of an application that is rendered computationally feasible by the emulators, we compute variance-based global sensitivity indices that precisely determine the combinations of parameters that most strongly control modelled subglacial hydrology.

**Table 1.** Subglacial drainage model variables (first group) and Gaussian Process model variables and parameters (second group)

| Symbol | Equation | Description |
| --- | --- | --- |
| $\phi$ | (1, A3) | Hydraulic potential (Pa) |
| $h_s$ | (A1)–(A2) | Water sheet thickness (m) |
| $S$ | (A4)–(A5) | Channel cross-sectional area (m$^2$) |
| $N$ | (2) | Effective pressure (Pa) |
| $f_w$ | (3) | Flotation fraction (unitless) |
| $f_q$ | — | Fraction of channelized discharge (unitless) |
| $T_s$ | — | Sheet transit time (a) |
| $L_c$ | — | Channel network length (m) |
| | | |
| $\mathbf{X}$ | — | Design matrix |
| $\mathbf{Y}$ | (9) | Simulation output matrix |
| $k$ | (13) | Covariance function |
| $\mathbf{\Sigma}$ | (4) | Covariance matrix |
| $\lambda$ | (13) | Gaussian Process precision (inverse variance) |
| $\boldsymbol{\beta}$ | (13) | Gaussian Process input sensitivity vector |
| $\boldsymbol{\theta}$ | (13) | $(\lambda, \boldsymbol{\beta})$ Gaussian Process hyperparameter vector |
| $d$ | — | Sample dimension (number of GlaDS parameters) |
| $m$ | — | Number of simulations |
| $n_s$ | — | Number of spatial locations in simulation outputs |
| $n_t$ | — | Number of time steps in simulation outputs |
| $p$ | (12) | Number of principal components |

## 2 Numerical and statistical models

### 2.1 Subglacial drainage model

We use the Glacier Drainage System (GlaDS) model (Werder et al., 2013) to simulate subglacial drainage. GlaDS computes water flow through coupled distributed and channelized drainage systems. Distributed drainage is modelled as macroporous sheet-flow with sheet geometry specified by the average water-layer thickness, aiming to describe area-averaged flow through subglacial cavities. Channelized drainage is modelled as flow through a network of one-dimensional ice-walled channels, with the channel radius determined by the balance between creep closure of ice and opening by melt. The governing equations, arising from conservation of mass and energy, are discretized on an unstructured triangular mesh. Variables describing the continuum distributed (sheet) drainage system are represented using finite elements with degrees of freedom located on the mesh nodes, with possible channel locations defined by element edges. We use the implementation of GlaDS within the Ice-sheet and

Sea-level System Model (ISSM; Larour et al., 2012; Ehrenfeucht et al., 2023), v4.24, with the laminar–turbulent sheet-flow parameterization introduced by Hill et al. (2024b). See Appendix A for a summary of the GlaDS governing equations.

For each timestep in a model run, GlaDS computes the subglacial hydraulic potential $\phi$ (units: Pa), the cavity height $h_s$ (units: m) and the channel cross-sectional area $S$ (units: m$^2$). From these three state variables, additional quantities of interest may be computed. For example, from the hydraulic potential and prescribed domain geometry, hydraulic potential can be rewritten in terms of water pressure $p_w$ (Pa), effective pressure $N$ (Pa), or flotation fraction $f_w$ (unitless):

$$\phi = p_w + \rho_w g z_b \tag{1}$$

$$N = p_i - p_w \tag{2}$$

$$f_w = \frac{p_w}{p_i}, \tag{3}$$

where gravitational acceleration $g$ and the densities of water ($\rho_w$) and ice ($\rho_i$) are given in Table 2, $z_b$ is the bed elevation and $p_i = \rho_i g H$ is the ice-overburden pressure for ice thickness $H$. For a mesh with $n_s$ nodes and for a simulation with $n_t$ timesteps, $\phi$, $N$ and $f_w$ are two-dimensional, time-varying fields with $n_s \times n_t$ values.

## 2.2  Gaussian Process model

The numerical model GlaDS, described above, is computationally intensive: model runs of ice-sheet outlet glaciers with seasonal surface melt forcing routinely take hours or days per modelled year. Emulators provide one solution to overcome the cost of running GlaDS simulations. This section briefly provides a high-level overview of the Gaussian Process (GP) model and the architecture that we use to emulate spatially and temporally resolved GlaDS outputs. For background on Gaussian

Processes see Jones et al. (1998) and Rasmussen and Williams (2005), and see Higdon et al. (2008) for a complete description of the emulators constructed here. Following tuning and evaluation of the emulators, we apply them to quantify the relationship between GlaDS parameters and GlaDS output.

Emulating GlaDS outputs is a type of regression problem: given a set of model simulations with various parameter values, we want to estimate the simulation response for untested parameter values for which we do not have GlaDS output. More

precisely, for an ensemble of $m$ (e.g., 256) GlaDS simulations, let $\boldsymbol{x}_i$ denote the $i^{\text{th}}$ vector of the $d = 8$ GlaDS parameter values. We store the parameter values for the entire ensemble in the $m \times d$ design matrix $\mathbf{X}$ (Fig. B1), where the rows specify the GlaDS parameter values used in each of the $m$ simulations. Each simulation produces spatiotemporally resolved hydraulic potential $\phi$, water sheet thickness $h_s$ and channel area $S$. Let $\boldsymbol{y}_i$ denote the vectorized model output of interest, with $n_s$ rows representing spatial positions and $n_t$ columns representing time steps, corresponding to the set of parameter values $\boldsymbol{x}_i$. The

ensemble of simulations is stored in the $n_s n_t \times m$ simulation output matrix $\mathbf{Y}$, with columns containing the simulation outputs corresponding to the parameters in $\mathbf{X}$. The emulation task is to predict the simulation output for new parameter values which are not included in the design matrix $\mathbf{X}$. Following the vocabulary of Higdon et al. (2008) and Verjans and Robel (2024), these GlaDS model parameters are called the inputs to the emulator. The steps involved in constructing the GP emulator and making predictions and summarized in Fig. 1.

**Table 2.** Constants (top group), fixed model parameters for GlaDS simulations (middle group) and GlaDS parameters and ranges used for training the GP emulator (bottom group).

| | Parameter | Value | Units |
|---|---|---:|---|
| $\rho_w$ | Density of water | 1000 | $\mathrm{kg\,m^{-3}}$ |
| $\rho_i$ | Density of ice | 910 | $\mathrm{kg\,m^{-3}}$ |
| $g$ | Gravitational acceleration | 9.81 | $\mathrm{m\,s^{-2}}$ |
| $L$ | Latent heat | $3.34 \times 10^5$ | $\mathrm{J\,kg^{-1}}$ |
| $c_w$ | Specific heat capacity of water | $4.22 \times 10^3$ | $\mathrm{J\,kg^{-1}}$ |
| $c_t$ | Pressure-melting coefficient | $-7.50 \times 10^{-8}$ | $\mathrm{K\,Pa^{-1}}$ |
| $\nu$ | Kinematic viscosity of water at $0°\mathrm{C}$ | $1.793 \times 10^{-6}$ | $\mathrm{m\,s^{-2}}$ |
| $\alpha_c$ | Channel-flow exponent | 5/4 | — |
| $\beta_c$ | Channel-flow exponent | 3/2 | — |
| $u_b$ | Basal velocity | 30 | $\mathrm{m\,a^{-1}}$ |
| $A_s$ | Ice flow-law coefficient when $N < 0$ | 0 | $\mathrm{s^{-1}\,Pa^{-3}}$ |
| $n$ | Ice-flow exponent | 3 | — |
| $\dot{m}_s$ | Basal melt rate | 0.05 | $\mathrm{m\,w.e.\,a^{-1}}$ |
| $k_s$ | Sheet conductivity | [0.01, 1] | $\mathrm{Pa\,s^{-1}}$ |
| $k_c$ | Channel conductivity | [0.05, 0.5] | $\mathrm{m^{3/2}\,s^{-1}}$ |
| $h_b$ | Bed bump height | [0.05, 1] | m |
| $r_b$ | Bed bump aspect ratio | [10, 100] | — |
| $A$ | Ice flow-law coefficient | $[10^{-24}, 10^{-23}]$ | $\mathrm{s^{-1}\,Pa^{-3}}$ |
| $l_c$ | Width of sheet beneath channels | [1, 100] | m |
| $\omega$ | Laminar–turbulent transition parameter | [1/5000, 1/500] | — |
| $e_v$ | Englacial void fraction | $[5 \times 10^{-5}, 5 \times 10^{-4}]$ | — |

### 2.2.1 Univariate Gaussian Process emulator

Gaussian Process regression as applied here may be thought of as an extension of kriging—where sparse observations are spatially interpolated to produce a gapless two-dimensional field (Krige, 1951)—to an arbitrary number of input dimensions that represent model parameters rather than spatial positions. This section describes the basics of Gaussian Process emulation for the case of scalar model outputs (i.e., where each $y_i$ is a scalar). Section 2.2.2 generalizes this approach to models with multivariate outputs (e.g., GlaDS).

A Gaussian Process emulator makes predictions of the GlaDS output of interest for new parameter values based on a weighted combination of the $m$ GlaDS simulations in the ensemble. The weights assigned to each simulation are based on

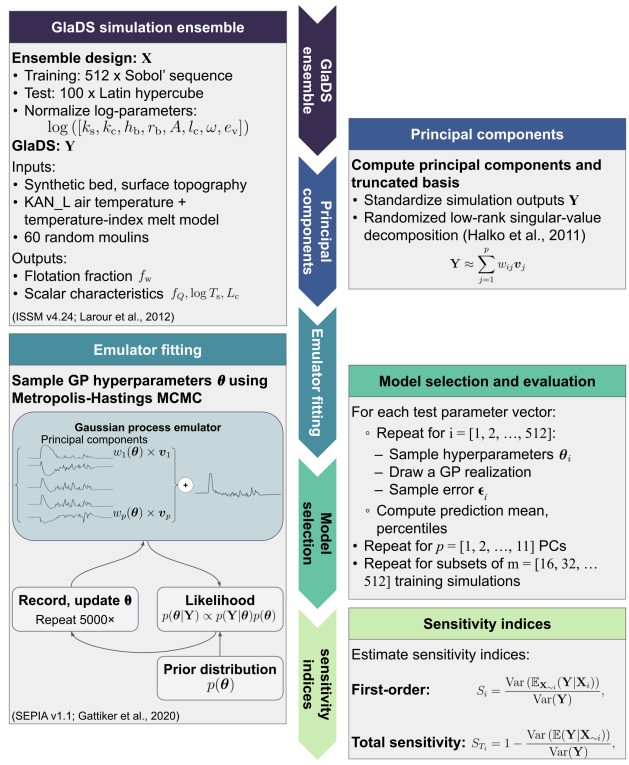

**Figure 1.** Overview of steps involved in constructing the Gaussian Process emulator for GlaDS flotation fraction. $\mathbf{X}$ is the design matrix of GlaDS parameters (defined in Table 2) with corresponding GlaDS outputs $\mathbf{Y}$. The Gaussian Process emulator is constructed as a truncated linear combination of $p$ principal components $w_i(\boldsymbol{\theta})$ and basis vectors $\boldsymbol{v}_j$ for $i = 1, \ldots, p$, where $\boldsymbol{\theta}$ are Gaussian Process hyperparameters that are inferred by Markov Chain Monte Carlo (MCMC) sampling. Emulators are fit using $m$-member subsets of the training data and constructed using different numbers of principal components $p$. The performance is evaluated on the independent set of 100 test simulations. The emulator is used to compute the sensitivity of model outputs to model parameters (Section 5).

the covariance corresponding to the parameter values of the simulation ensemble and the parameter values being used for emulator predictions. The GP is completely specified by the mean function $\mu(\boldsymbol{x})$ and the covariance function $k(\boldsymbol{x}_i, \boldsymbol{x}_j; \boldsymbol{\theta})$ with hyperparameters $\boldsymbol{\theta}$. The covariance matrix $\boldsymbol{\Sigma}$ is constructed to represent the pairwise covariance between model outputs in the simulation ensemble,

$$\boldsymbol{\Sigma}_{ij} = k(\boldsymbol{x}_i, \boldsymbol{x}_j; \boldsymbol{\theta}). \tag{4}$$

The parameters of the covariance function $\boldsymbol{\theta}$ are referred to as hyperparameters to distinguish them from GlaDS parameters. The hyperparameters typically control the variance of the Gaussian Process and the sensitivity to each input, but their interpretation depends on the type of covariance function that is used. Gaussian Processes typically have a similar number of hyperparameters as inputs to the emulator. The hyperparameters must be optimized to obtain an accurate emulator.

We make the common choice to set the prior mean to zero everywhere (e.g., Kennedy and O'Hagan, 2001). The zero-mean prior specifies that, before training the GP with the ensemble $(\mathbf{X}, \mathbf{Y})$, the pointwise mean of many GP predictions is zero for all parameter values. Since we standardize the simulation outputs, this corresponds to predicting the mean of the ensemble

once the GP predictions are converted back into physical units. Additional structure could be included by using a nonzero prior mean function, for instance a linear regression mean term. In cases where prior knowledge of the functional form of the model's response to parameter variations is available (for example, if it is known that the model responds nearly linearly to parameter variations), the mean function can be used to encode this knowledge (Rasmussen and Williams, 2005). We have found that the GP with zero prior mean is able to learn how the model output changes as parameters are varied without the

additional structure introduced by a mean function, while producing a simpler model with fewer hyperparameters to estimate.

The GP emulator is trained by using the simulation outputs $\mathbf{Y}$ as the basis for predictions and to infer the hyperparameter values $\boldsymbol{\theta}$. Since predictions with the trained emulator, called posterior predictions, depend on the simulation outputs $\mathbf{Y}$ and hyperparameters $\boldsymbol{\theta}$, they are said to be conditioned on $\mathbf{Y}$ and $\boldsymbol{\theta}$. Unlike a neural network that typically produces a deterministic prediction, the GP predictions form a probability distribution. For GlaDS parameters $\boldsymbol{x}_{\mathrm{p}}$, the corresponding emulator prediction

is denoted $y_{\mathrm{p}}$, with a corresponding normal distribution:

$$y_{\mathrm{p}}|\mathbf{Y}, \boldsymbol{\theta}, \boldsymbol{x}_{\mathrm{p}} \sim \mathcal{N}\left(\mu_{\mathrm{p}}, \sigma_{\mathrm{p}}\right), \tag{5}$$

with estimated mean

$$\hat{\mu}_{\mathrm{p}} = \boldsymbol{k}_{\mathrm{p}} \boldsymbol{\Sigma}^{-1} \mathbf{Y} \tag{6}$$

and variance

$$\hat{\sigma}_{\mathrm{p}}^2 = \sigma^2 - \boldsymbol{k}_{\mathrm{p}}^T \boldsymbol{\Sigma}^{-1} \boldsymbol{k}_{\mathrm{p}}. \tag{7}$$

The vector $\boldsymbol{k}_{\mathrm{p}} = k(\boldsymbol{x}_{\mathrm{p}}, \boldsymbol{x})$ contains the pairwise covariance between the model outputs from the simulation ensemble and the estimated output for parameters $\boldsymbol{x}_{\mathrm{p}}$, while $\sigma^2 = k(\boldsymbol{x}_{\mathrm{p}}, \boldsymbol{x}_{\mathrm{p}}; \boldsymbol{\theta})$ is the GP variance (Jones et al., 1998; Rasmussen and Williams, 2005). The form of the estimated mean (6) reveals that the mean prediction is a covariance-weighted sum of the model outputs $\mathbf{Y}$. The estimated variance (7) shows that the uncertainty in the emulator prediction is determined by the GP variance $\sigma^2$ and

140 the covariance $\boldsymbol{k}_p$, with lower uncertainty when predictions are made close to parameter values used in the ensemble (i.e., higher covariance $\boldsymbol{k}_p$).

The prediction mean (Eq. 6) and variance (Eq. 7) estimates depend on the values of the GP hyperparameters $\boldsymbol{\theta}$ through the covariance function (Eq. 4). To include a full assessment of uncertainty, we sample $\boldsymbol{\theta}$ from its posterior distribution using Markov Chain Monte Carlo (MCMC) sampling (e.g., Rasmussen and Williams, 2005; Higdon et al., 2008). From Bayes' rule

(e.g., Gelman et al., 2013), the probability distribution of the GP hyperparameters $\boldsymbol{\theta}$ given the simulation outputs $\mathbf{Y}$, called the posterior distribution, can be written

$$p(\boldsymbol{\theta}|\mathbf{Y}) \propto p(\mathbf{Y}|\boldsymbol{\theta})p(\boldsymbol{\theta}), \tag{8}$$

where $p(\mathbf{Y}|\boldsymbol{\theta})$, called the likelihood, takes the usual form for a multivariate Normal distribution with covariance $\boldsymbol{\Sigma}$,

$$p(\mathbf{Y}|\boldsymbol{\theta}) = \frac{1}{(2\pi)^{m/2}|\boldsymbol{\Sigma}|^{1/2}} \exp\left(-\frac{1}{2}\mathbf{Y}^T\boldsymbol{\Sigma}^{-1}\mathbf{Y}\right). \tag{9}$$

The prior distribution, $p(\boldsymbol{\theta})$, quantifies prior beliefs in plausible values of the emulator hyperparameters $\boldsymbol{\theta}$. MCMC sampling (Fig. B2) produces a set of $\boldsymbol{\theta}$ values which result in GP predictions that appropriately fit the simulation outputs. Emulator predictions $y_p$ are made by sampling hyperparameter values from the MCMC chain that are used to draw realizations of the GP predictions $y_p$ from the GP predictive distribution (Eq. 5). In addition to the probabilistic specification that includes emulator uncertainty (Eq. 5), the fact that the GP model is simple enough to allow for Bayesian inference of the emulator hyperparameter

values $\boldsymbol{\theta}$, where uncertainty in the hyperparameters is reflected in the uncertainty in the emulator predictions, is a key advantage compared to a neural network for uncertainty quantification.

### 2.2.2 Multivariate Gaussian Process emulator

The GP emulator described above does not directly transfer to multivariate or spatiotemporally resolved models. One approach to emulate spatiotemporal outputs would be to view the simulation output as a scalar value that is predicted as a function of

160 position, time and the parameters of the numerical model (i.e., GlaDS). To emulate GlaDS simulation outputs defined on $n_s$ mesh nodes, at $n_t$ timesteps (e.g., daily) and using $m$ simulations with different parameter values, this approach would result in a covariance matrix with $n_s n_t m$ rows and columns. For the GlaDS simulation ensembles, this would be at least $10^8$ rows and columns, making computing the determinants and inverses of the covariance matrix required to evaluate the GP likelihood (Eq. 9) infeasible. There are a variety of solutions to this issue of GP scalability (e.g., Liu et al., 2020), such as limiting

the quantity of training data (e.g., Chalupka et al., 2013) or reducing the rank (e.g., Smola and Bartlett, 2000; Quinonero-Candela and Rasmussen, 2005) or number of nonzero entries in the covariance matrix (e.g., Kaufman et al., 2011), and local approximations that make predictions based only on nearby points (e.g., Gramacy, 2016).

  We follow a common approach for multivariate outputs, proposed by Higdon et al. (2008), that views the simulation output as a spatiotemporal field which is a function only of the scalar parameter values of the numerical model. This choice corresponds

to the model

$$\boldsymbol{y}_i = \boldsymbol{\eta}(\boldsymbol{x}_i) + \epsilon_i, \tag{10}$$

where $\boldsymbol{y}_i$ is the simulation output (a spatiotemporal field) corresponding to GlaDS parameter values $\boldsymbol{x}_i$ and $\boldsymbol{\eta}(\boldsymbol{x}_i)$ is the vector-valued emulator. The error term $\epsilon_i \sim \mathcal{N}\left(\mathbf{0}, \frac{1}{\lambda_{\text{sim}}}\mathbf{I}\right)$ is taken to be multivariate normal, parameterized by the precision hyperparameter $\lambda_{\text{sim}}$. This error model assumes that errors at each spatial position and timestep are uncorrelated, which might

not be strictly true for our application. The corresponding likelihood is constructed from the univariate GP likelihood (Eq. 9) by augmenting the simulation covariance matrix $\boldsymbol{\Sigma}$ with the covariance associated with the error term $\epsilon_i$,

$$p(\mathbf{Y}|\boldsymbol{\theta}) = \frac{1}{(2\pi)^{m/2}\left|\boldsymbol{\Sigma} + \frac{1}{\lambda_{\text{sim}}}\mathbf{I}\right|^{1/2}} \exp\left(-\frac{1}{2}\mathbf{Y}^T\left[\boldsymbol{\Sigma} + \frac{1}{\lambda_{\text{sim}}}\mathbf{I}\right]^{-1}\mathbf{Y}\right). \tag{11}$$

In order to reduce the dimensionality of the simulation outputs, which leads to the obstacles described above, the multivariate output field is modelled directly by using a principal component (PC) decomposition. The PC decomposition represents the $n_s n_t \times m$ simulation output matrix with a smaller $p \times m$ matrix of principal components (where $p$ is much smaller than the number of simulations $m$) by encoding the space and time dimensions in the PC coefficients, rather than explicitly modelling the dependence. In this way, the principal component step can be viewed as a dimension-reduction of the simulation outputs.

The PC decomposition writes each simulation output $\boldsymbol{y}_i$ and emulator prediction $\boldsymbol{\eta}(\boldsymbol{x}_i)$ as the sum of scalar principal components (PCs) $w_{ij}$ multiplied by basis vectors $\boldsymbol{v}_j$. For example, for the simulation output $\boldsymbol{y}_i$,

$$\boldsymbol{y}_i \approx \sum_{j=1}^{p} w_{ij} \boldsymbol{v}_j. \tag{12}$$

Note that the truncated basis approximation becomes the exact singular value decomposition of $\mathbf{Y}$ by retaining all $p = m$ PCs. However, often a much smaller number of PCs can sufficiently approximate $\mathbf{Y}$. Since the simulation output matrix $\mathbf{Y}$ is large, with many more rows than columns (i.e., $n_s n_t \gg m$), the principal components $w_{ij}$ and basis vectors $\boldsymbol{v}_j$ are efficiently computed using a randomized low-rank singular value decomposition (Halko et al., 2011).

Following Higdon et al. (2008), since the PCs are orthogonal, independent univariate GPs are fit to model the relationship between the emulator inputs $\mathbf{X}$ and the PC vectors $\boldsymbol{w}_j = (w_{1j}, w_{2j}, \ldots, w_{mj})$ for $j = 1, \ldots, p$. For each univariate GP, letting $j$ index the principal components, we choose to use a squared-exponential covariance function,

$$k(\boldsymbol{x}, \boldsymbol{x}'; \lambda_j, \boldsymbol{\beta}_j) = \frac{1}{\lambda_j} \exp\left(-\sum_{l=1}^{d} \beta_{jl}(x_l - x_l')^2\right), \tag{13}$$

where $\lambda_j$ specifies the precision of the GP predictions for PC $j$ (inverse of the variance) and $\beta_{jl}$ specifies the sensitivity to each emulator input $X_l$ for PC $j$. The squared exponential covariance function (Eq. 13) results in highly smooth GP predictions (e.g., Rasmussen and Williams, 2005). Other covariance functions are available, for example the Matérn family, that are more permissive in the imposed degree of smoothness. While the flotation fraction field need not be smooth in space and in time, the principal components $w_{ij}(\boldsymbol{\theta})$ tend to vary smoothly with respect to the GlaDS parameters since the the spatiotemporal complexity is captured by the principal component basis.

Spatiotemporally resolved emulator predictions, $\boldsymbol{\eta}(\boldsymbol{x}_i)$, are constructed from the $p$ individual GPs by substituting posterior GP realizations for $w_{ij}$ in Eq. (12) using posterior samples of the GP hyperparameters $\boldsymbol{\theta}$. Predictions of the simulation outputs for GlaDS parameter values $\boldsymbol{x}_p$ are made by sampling from the GP $\boldsymbol{\eta}(\boldsymbol{x}_p)$ and the error $\epsilon_i$ (Eq. 10). The prediction mean and intervals containing 95% of the predictions can then be computed from these posterior samples of Eq. (12) (Fig. 1).

This PC-based approach for modelling multivariate simulation outputs assumes that the original data $\mathbf{Y}$ can be effectively modelled with just a small number ($p \ll m$) of PCs. We are optimistic that this assumption will hold, in part since Brinkerhoff et al. (2021) developed a neural network hydrology–dynamics emulator based on a principal component decomposition of annual-average surface velocities. We note that the complete statistical model (Eq. 10) has $p(d+1)+1$ hyperparameters to estimate (in addition to $p$ itself). To find an appropriate number of principal components $p$, we develop subglacial drainage emulators for a range of $p$ values. In practice, GP predictions can be less accurate for later principal components that explain a

small fraction of the ensemble variance (e.g., Higdon et al., 2008). Since including GPs for these later PCs does not meaningfully improve predictions, we will select a model with a modest number of principal components that nonetheless has similar performance obtained by using more components.

## 3   Experimental design

We run a large ensemble of simulations using an all-at-once parameter sampling strategy to uniformly explore the input parameter space and generate the simulation data used to fit the GP emulators. This section describes the setup of the GlaDS model, the experimental design to generate parameters for the simulation ensembles and the outputs that are extracted from the simulations for GP emulation.

### 3.1   Synthetic ice-sheet outlet glacier

#### 3.1.1   Domain and geometry

The GlaDS model is applied to a $100\,\mathrm{km} \times 25\,\mathrm{km}$ synthetic ice-sheet margin domain (Fig. 2b). The synthetic domain is modified from the synthetic ice-sheet geometry used in SHMIP experiments A–D (de Fleurian et al., 2018) to represent the land-terminating regions near the K-transect on the western Greenland Ice Sheet (van de Wal et al., 2005; Smeets et al., 2018). The synthetic geometry consists of a flat bed with an elevation of $350\,\mathrm{m}$ (adjusted from $0\,\mathrm{m}$ in the SHMIP experiment) and surface elevation between $390$–$1910\,\mathrm{m}$ (adjusted from $1$–$1520\,\mathrm{m}$ in the SHMIP experiment) to match the observed elevation range of this part of the ice sheet.

#### 3.1.2   Melt forcing

GlaDS is forced with prescribed basal melt rates and surface melt inputs through moulins. We impose a steady basal melt rate of $0.05\,\mathrm{m\,w.e.\,a^{-1}}$, representing the total melt rate from the geothermal flux and basal sliding. This basal melt rate is in line with modelled basal melt rates in western Greenland (e.g., $0.001$–$0.1\,\mathrm{m\,w.e.\,a^{-1}}$, Karlsson et al., 2021), but lacks the seasonality associated with basal sliding. Surface melt rate is computed with a positive degree-day model forced with daily mean air temperatures recorded in 2014 at the PROMICE lower K-transect (KAN_L, 670 m asl.) weather station (Fausto et al., 2021; How et al., 2022) (Fig. 2a). We randomly place 60 moulins within the domain following a moulin density that varies with elevation. Based on a satellite-derived supraglacial drainage map (Yang and Smith, 2016), moulin density is parameterized by a normal distribution with mean 1138 m and standard deviation 280 m. Surface melt is accumulated within sub-catchments surrounding each moulin defined by a Voronoi diagram and instantaneously routed to the bed.

#### 3.1.3   Boundary and initial conditions

The model is posed on an unstructured triangular mesh consisting of 3693 nodes with a mean edge length of $\sim 900\,\mathrm{m}$. We impose an atmospheric pressure boundary condition along the 25 km-wide terminus to represent a land-terminating outlet and

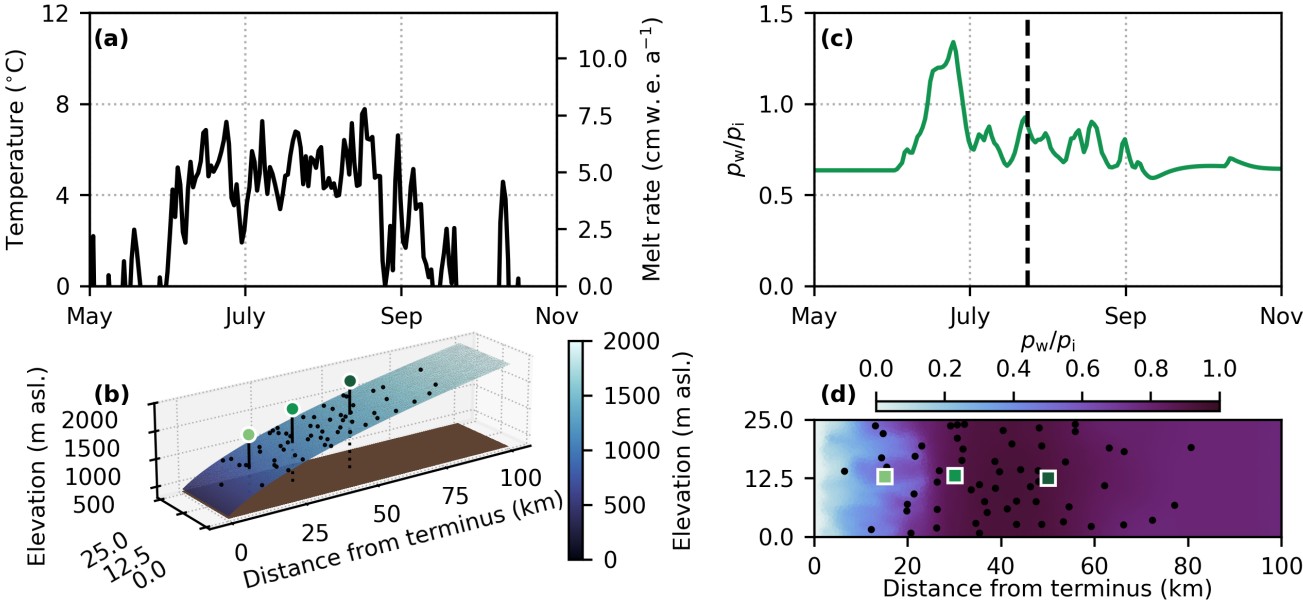

**Figure 2.** Synthetic ice-sheet margin experiment configuration. (a) Terminus (390 m asl.) air temperature forcing and surface melt rate derived from KAN_L weather station daily temperature (How et al., 2022) assuming a lapse rate of $5°C\,km^{-1}$ and degree-day factor $0.01\,m\,w.e.\,d^{-1}\,°C^{-1}$. (b) Surface elevation, bed elevation, moulins (black dots) and glacier centerline positions 15, 30, and 50 km from the terminus (indicated by corresponding squares in (d)). (c) Example of flotation fraction timeseries at 30 km position. (d) Example of flotation fraction field on 24 July (indicated by dashed vertical line in (c)) with moulins (black dots) and glacier centerline positions.

apply a zero-flux condition elsewhere. Model runs are initialized with zero channel area, cavity height equal to 20% of the bed bump height and water pressure equal to ice-overburden pressure. The model evolves from this initial condition to a steady state with respect to hydraulic potential within the first winter. We run the model for two complete years with identical melt forcing, discarding the first year as a spinup period to minimize the influence of channel initialization. For certain parameter combinations, a steady state channel network may not be reached after these two melt seasons as large channels that develop under thin ice near the terminus may take five or more years to reach a periodic steady state. Since the focus of the current work is developing a subglacial drainage model emulator rather than studying the subglacial hydrology of a particular physical system, we limit the simulations to two years in an effort to minimize the time required to run large ensembles of simulations.

### 3.2 Ensemble design

#### 3.2.1 Model parameters

We vary eight uncertain GlaDS model parameters in ensembles of simulations (Table 2). Consistent with Brinkerhoff et al. (2021), we apply a log transformation to all GlaDS parameters to capture variations over the ranges that span one or more orders of magnitude. The ranges for model parameters used as emulator inputs (Table 2) have been chosen to balance maximizing the

variation in each parameter and minimizing the proportion of nonphysical model outputs. Given the large variations in surface melt rates on diurnal and multi-day timescales which result in large variations in water pressure, we have had to choose narrower parameter ranges than used by Brinkerhoff et al. (2021). We have found that broadening the parameter ranges results in numerous nonphysical simulations with nearly zero water pressure during the melt season, transient negative flotation fraction as low as $f_\mathrm{w} < -10$ or flotation fraction as high as $f_\mathrm{w} \gg 100$ which degrade the performance of the principal component decomposition. These problems are symptomatic of the limitations of the GlaDS model physics, and since we are not interested in training the emulator to reproduce these nonphysical outputs, we restrict the parameter ranges (and therefore restrict the domain over which GP predictions can be reliably made) to curate an ensemble with fewer nonphysical members. We sample from the entire region described by the bounds listed in Table 2 without filtering or discarding nonphysical training runs (c.f., Jantre et al., 2024). The ensemble still contains some instances of negative or extremely high flotation fraction, but these do not appear to negatively impact the principal component decomposition nor the emulator predictions. Since the goal of the emulator is to reproduce GlaDS outputs as closely as possible, we do not constrain the emulator to predicting realistic flotation fraction $f_\mathrm{w} \geq 0$.

We construct separate ensembles for emulator training and evaluation. The training design consists of 512 samples from a space-filling Sobol' sequence (Sobol', 1967). Sobol' sequences were chosen for their sequential design properties: each $2^k$ subset (for positive integer $k$) of the sequence is itself an approximately uniformly space-filling design (Fig. B1). The trade-off between computational investment to generate training data and GP prediction performance can then be assessed by fitting GP models with subsets of the original sequence. The test design for model evaluation consists of 100 samples from a space-filling, centred-discrepancy-minimized Latin hypercube (McKay et al., 1979). A different sampling strategy is used for the test design to minimize the overlap between the training and test sets to ensure a fair evaluation of model performance.

### 3.2.2   Simulation outputs

The primary output from each subglacial drainage model run is a flotation fraction field (Eq. 3) defined at $n_\mathrm{s}$ spatial locations and for $n_t$ time steps (Fig. 2c, d). We could alternatively treat water pressure, effective pressure, or hydraulic potential as our target variable to measure distributed water pressure, but have chosen flotation fraction for its natural scaling that lies mostly between $[0, 1]$ and since it removes much of the baseline geometric signal related to bed elevation and ice thickness. Other model outputs (e.g., sheet thickness $h_\mathrm{s}$, channel area $S$) could equally be considered targets for emulation, but here we focus on measures of distributed water pressure.

In addition to emulating the spatiotemporal flotation fraction $f_\mathrm{w}$, we also explore the possibility of emulating scalar quantities derived from the full output fields to describe the distributed and channelized drainage systems. We define three scalar quantities of interest as proxies for key physical processes taking place in the subglacial drainage system. Each drainage system component (sheet, channel network) is described by an aggregate quantity of interest, with a third quantity describing the partitioning between the components:

1. The channel discharge fraction $f_Q$ quantifies the partitioning between distributed and channelized drainage. For a flux-gate at a fixed distance from the terminus, the channel discharge fraction is defined as the ratio of the melt-season-integrated discharge crossing the fluxgate through channels to the total melt-season-integrated discharge crossing the fluxgate through both the sheet and channel network. We evaluate the channel discharge fraction for flux gates placed every 5 km between 5 km and 30 km from the terminus and compute the average across the six fluxgate positions to limit the influence of the particularities of moulin position on this metric. The channel discharge fraction $f_Q$ potentially influences glacial ecosystems (e.g., Hodson et al., 2008; Dubnick et al., 2017), subglacial chemical weathering (e.g., Graly and Rezvanbehbahani, 2022) and channelized grounding line discharge of tidewater glaciers (e.g., Meire et al., 2017).

2. Sheet transit time $T_s$ describes the efficiency of the distributed (sheet) drainage system. Starting at a fluxgate placed a fixed distance from the terminus, the sheet transit time is the downstream-integrated and width-averaged time it would take a parcel of water to travel to the terminus through the distributed drainage system. The sheet transit time is computed as the average time starting from each of the same six fluxgate positions between 5 km and 30 km. As a proxy for water residence time in the subglacial drainage system, sheet transit time $T_s$ may be a factor in controlling chemical weathering rates (e.g., Crompton et al., 2015; Graly and Rezvanbehbahani, 2022) and subglacial microbial composition (e.g., Dubnick et al., 2017).

3. The extent of channel network development is described by the total channel network length $L_c$. Given a prescribed channel radius threshold $R$, the total channel network length is the sum of the length of all channel segments (defined on mesh edges) with radius $\geq R$. The total channel network length is computed for radius threshold $R = h_b = 0.5\,\mathrm{m}$, corresponding to the maximum bed bump height so that a channel segment is never smaller than the largest cavities. By approximately determining the extent of the bed which is affected by channelized discharge, the total channel network length $L_c$ potentially influences weathering rates (e.g., Sharp et al., 1999) and glacial ecosystems (e.g., Hodson et al., 2008).

## 4 Evaluation of the Gaussian Process emulator

### 4.1 Principal component decomposition

Before evaluating the GP emulator for flotation fraction, we investigate the assumption that the GlaDS simulation outputs can be appropriately represented by the first several principal components. The error in truncating the PC approximation to retain only the first $p$ PCs is quantified by (1) the root-mean-square-error (RMSE) between the original simulation outputs and the PC approximation and (2) the cumulative proportion of explained variance (Fig. 3). For all ensembles, the RMSE and cumulative proportion of variance rapidly converge as the number of PCs increases. For example, no more than 9 PCs are needed to capture at least 95% of the variance of the simulation ensemble, with RMSE not exceeding 0.045, compared to the system response expected to be between 0 and 1. Larger ensembles (which include more simulations) require more PCs to obtain the same

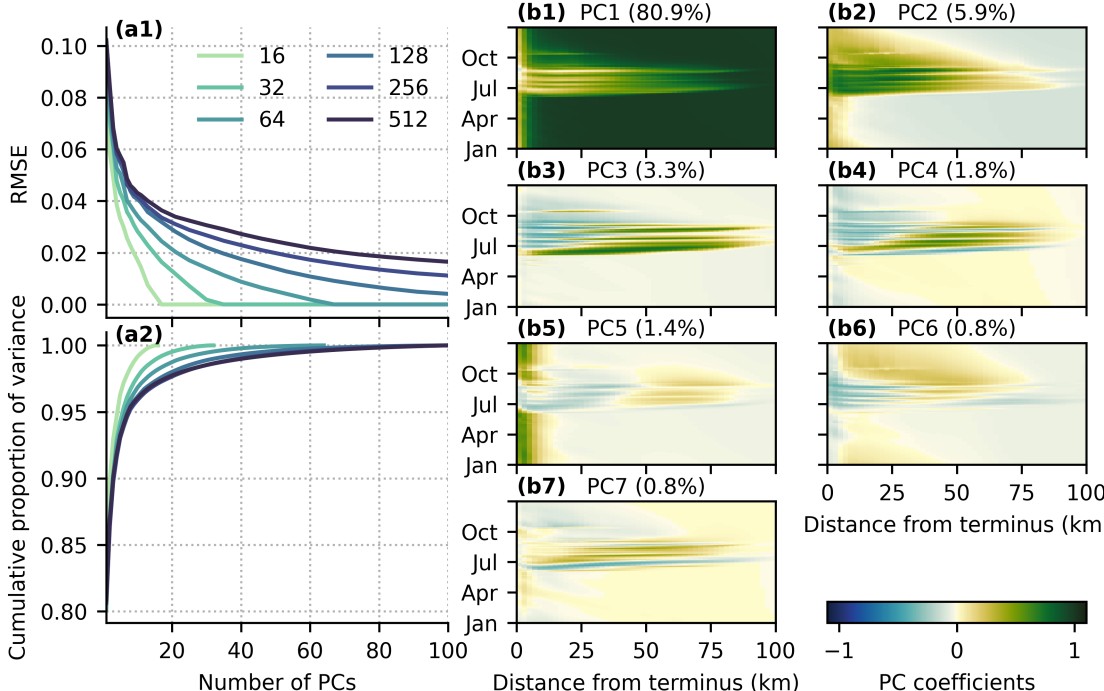

**Figure 3.** Principal component decomposition of spatiotemporal flotation fraction field. (a1) RMSE of the truncated principal component representation and (a2) cumulative proportion of variance explained for the first 100 PCs of ensembles of 16–512 simulations. (b) Width-averaged representation of the first seven flotation fraction $f_\mathrm{w}$ spatiotemporal principal component basis vectors $\boldsymbol{v}_j$ (Eq. (12)) for the 256 simulation ensemble. Note that the PC basis is defined in time and both horizontal dimensions but has been width-averaged for ease of visualization.

RMSE value and explain the same cumulative proportion of variance, perhaps since the input space has been explored more thoroughly. The RMSE (0.045) and cumulative proportion of variance (95%) with 9 PCs together provide evidence that the GlaDS simulation outputs can be effectively represented by a small number ($\lesssim 10$) of PCs. This PC representation compresses the original simulation data by a factor of approximately $10^5$, reducing the number of columns in the simulation output matrix from $\mathcal{O}(10^6)$ to $\mathcal{O}(10)$.

The principal component decomposition produces a set of basis vectors representing the dominant modes of flotation fraction variation in space and time (Fig. 3b1–b7). Note that the sign of the PCs and basis vectors are arbitrary since inverting the sign of both the basis $\boldsymbol{v}_j$ and the PC $w_{ij}$ in Eq. (12) yields identical results. Based on the first PC basis vector being nonzero in winter and upstream of the maximum surface melt extent ($\sim 80\,\mathrm{km}$), and not contributing to the solution at low elevations during the melt season, the first and most important PC in terms of its explained variance (80.6%) appears to control the baseline water pressure in the absence of surface melt inputs. PC2 is expressed most strongly in the lower half of the domain and during the

melt season, suggesting that PC2 controls surface-melt-influenced summer water pressure. The remaining PCs are expressed as mixed positive/negative regions mostly confined to the melt season, making them more difficult to interpret.

## 4.2 Emulator performance for flotation fraction field

### 4.2.1 Model selection

With PC truncation error quantified, we apply the Gaussian Process emulator described in section 2.2.2 to the GlaDS simulation outputs. Rather than selecting an appropriate number of principal components based on an arbitrary explained variance threshold (e.g., 99%), we fit GPs for a range of 1–11 PCs and evaluate the resulting performance based on error metrics (RMSE and mean absolute percent error, MAPE) and whether GP predictions capture the essential features of the simulation data. Since the convergence rate of the PC truncation error varies with ensemble size (Fig. 3), we repeat fitting each GP using subsets of the Sobol' design (section 3.2.1) to identify how the appropriate number of PCs changes for different training ensembles. Prediction performance is quantified by the RMSE and MAPE between the test simulation data (not seen by the emulators during the fitting) and emulator predictions on the 100-member test ensemble. These metrics are complementary since RMSE is an absolute error that is more sensitive to large deviations from the test data while MAPE is a relative error that is more sensitive to consistent bias (Fig. 4).

The RMSE and MAPE curves provide some evidence that more PCs should be included for larger ensembles. For the 32 simulation ensemble, there is no significant benefit, in terms of RMSE and MAPE reduction, to using more than four PCs. For ensembles with at least 128 simulations, there are slight reductions in RMSE and MAPE by adding additional PCs. However, the reductions are within the interquartile range of the test errors, suggesting the improvement in model prediction performance compared to simpler models is small.

To identify the appropriate reference model for further evaluation, we look in more detail at the prediction performance of a GP based on 256 training simulations. Figures 5a–c show the distribution of the RMSE, MAPE and the spatiotemporally averaged 95% prediction interval evaluated on the test ensemble for emulators constructed using different numbers of PCs. The RMSE and MAPE distributions (Fig. 5a, b) again suggest that including more than four PCs slightly reduces the median and range of prediction performance across the test ensemble. Prediction uncertainty as measured by the spread of the central 95% of emulator predictions (Fig. 5c) initially decreases as more PCs are included, reflecting the reduction in PC truncation error, with minimal further reduction in uncertainty for 8–11 PCs. The fact that the 95% prediction intervals (Fig. 5c) overlap nearly 95% of the simulated values for all but the smallest subset of 16 trainings runs (Fig. B4) suggests that the emulators have well-calibrated uncertainty estimates. Figure 5 seems to suggest that the RMSE and MAPE are converging to nonzero values. This is an expected outcome since the total error represents the sum of the basis truncation error associated with using at most 11 PCs (Fig. 3) and error in the GP predictions of the principal components.

Insight into the common modes of GP prediction error can be found by computing the mean RMSE spatial patterns and timeseries. To assess how the spatial and temporal patterns change as more PCs are included in the model, we consider models based on 256 training simulations and 2, 5 and 8 PCs (Fig. 6). For all models, the spatial pattern of prediction error is highest

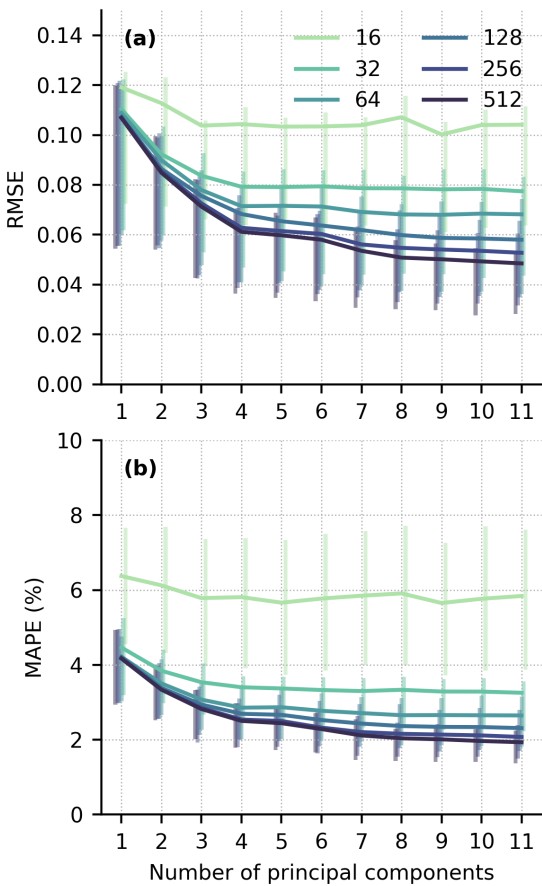

**Figure 4.** Median root mean square error (RMSE, a) and mean absolute percent error (MAPE, b) for GP emulator predictions across the 100-member test ensemble based on 16–512 simulations in the training set and using 1–11 principal components. Vertical bars indicate interquartile (25th to 75th percentile) ranges of RMSE and MAPE.

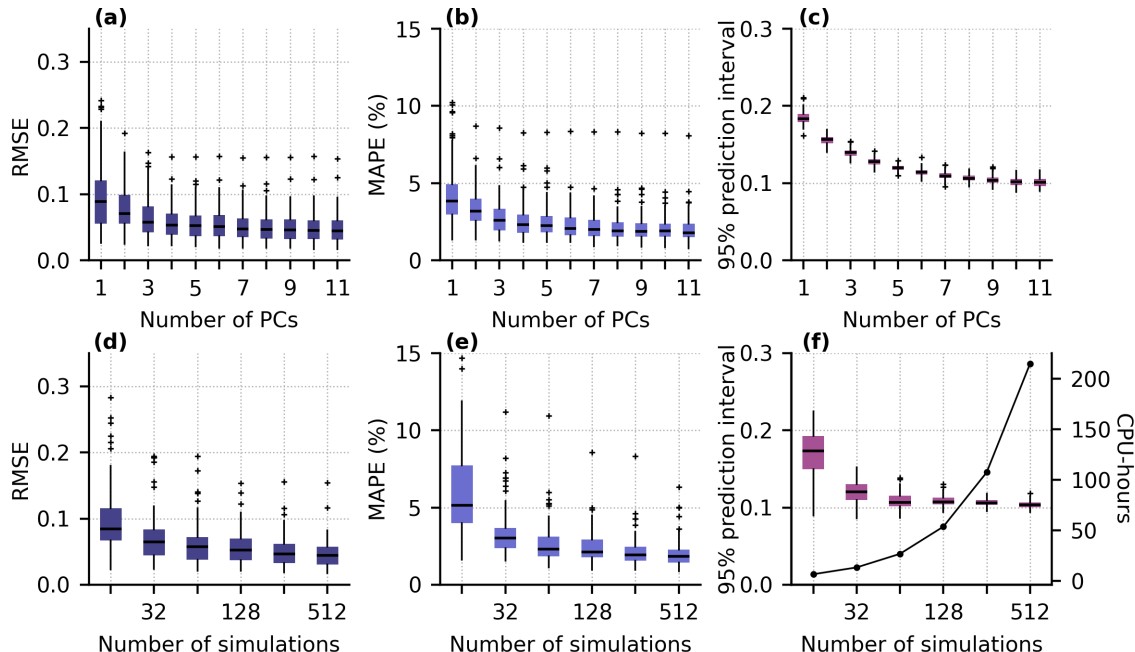

**Figure 5.** Gaussian Process emulator prediction performance, evaluated using the common 100 member test set, for varying model complexity (a–c) and using subsets of the training data (d–f). (a, d) RMSE of predictions. (b, e) MAPE of predictions. (c, f) Width of 95% prediction intervals for each test simulation (boxes, whiskers and crosses). The solid line in (f) indicates the number of CPU-hours needed to run the GlaDS training simulations for each subset. Boxplots show the median (horizontal line), interquartile range (shaded box), and the interval between the lowest and highest data points within 1.5 times the interquartile range from the median (whiskers), with outliers beyond the whisker extents (crosses). Note the logarithmic horizontal axis for (d–f).

near the terminus, where seasonal variations in flotation fraction are largest, and near moulins where surface melt is injected to the bed. The simple model with 2 PCs has larger and more widespread errors over the lowest ∼60 km of the domain, while the models with 5 and 8 PCs have similar moulin-influenced spatial distributions (Fig. 6 a–c). The seasonal evolution of prediction error (Fig. 6 d) reaches a maximum in the spring and decreases through summer, with higher error occurring during periods with high melt inputs. The simplest model results in higher maximum RMSE in spring and in each melt event through the summer. Using 8 PCs reduces the height of the RMSE peaks in summer relative to the 5 PC model. All models have a similar RMSE during the September melt event, with relatively little improvement obtained by including more PCs. We take the GP with 8 PCs as the reference model for further evaluation.

The trade-off between computation time to run the simulation ensemble and the resulting GP prediction accuracy is evaluated by comparing the prediction RMSE, MAPE and uncertainty for GPs with the same architecture ($p = 8$ PCs) but restricted to subsets of the training data (Fig. 5d–f). Increasing the number of simulations monotonically decreases prediction RMSE and MAPE since the distance the GP must interpolate from training points to test points is smaller (Fig. 4). Prediction uncertainty is

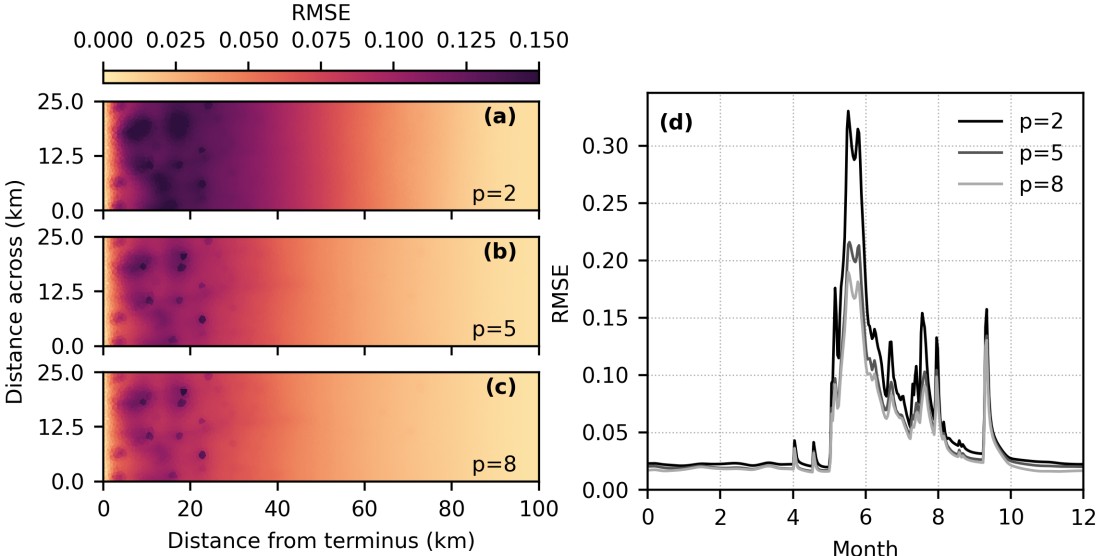

**Figure 6.** GP prediction RMSE for emulators using 2, 5 and 8 PCs. (a–c) Time and input-averaged RMSE spatial patterns. (d) Domain and input-averaged RMSE timeseries.

similar for models trained with 64–512 simulations (Fig. 3). We select 256 simulations as our reference training set. However, the similar median RMSE and MAPE, and the overlapping prediction uncertainty distributions, suggest that using 128 or 512 simulations would also be an appropriate trade-off between investment in generating the ensemble and GP performance.

**4.2.2 Model evaluation**

With the reference model architecture selected, consisting of 8 PCs and trained using the first 256 GlaDS simulations, we evaluate the GP emulator in more detail by comparing predictions on the set of 100 test parameter values to the unseen GlaDS simulations. The median on the test set of the GP prediction RMSE is 0.047 and the median MAPE is 2.0% (Table 3). Emulator predictions have negligible consistent bias and explain 97% of the variance of the GlaDS ensemble. Errors are higher in the
380 lower part of the domain and in the summer (months JJA), although the median coefficient of determination $R^2$ is higher below 30 km (0.95) than above 30 km (0.92).

To assess how emulator performance varies across the test set, we evaluate the performance on test simulations with 95th-percentile ("high error"), 50th-percentile ("median error") and 5th-percentile ("low error") RMSE. While emulator predictions capture the primary spatial (Fig. 7) and seasonal (Fig. 8) flotation fraction patterns for each of these three test simulations,
many of the GlaDS simulations produce unrealistically high water pressure exceeding overburden ($f_w \gg 1$) for long periods of time and over a large portion of the domain (e.g., Fig. 7a2 and Fig. 8a2, b2) for the ranges of parameters used in designing the ensembles. However, the purpose of the emulator is to produce predictions similar to the simulation values, rather than to produce flotation fraction fields that align with expectations of realistic subglacial hydrology.

**Table 3.** Median spatiotemporally averaged Gaussian Process emulator prediction RMSE, MAPE, bias and coefficient of determination $R^2$. Bracketed numbers indicate the 5th and 95th percentile values across the 100 test simulations.

|  | RMSE | MAPE | Bias | $R^2$ |
|---|---|---|---|---|
| Overall | 0.047 | 0.020 | 0.00 | 0.97 |
|  | (0.022, 0.088) | (0.012, 0.036) | (-0.009, 0.009) | (0.90, 0.99) |
| Lower 30 km | 0.075 | 0.12 | -0.001 | 0.95 |
|  | (0.034, 0.14) | (0.070, 0.21) | (-0.021, 0.017) | (0.85, 0.98) |
| Upper 70 km | 0.025 | 0.010 | 0.00 | 0.92 |
|  | (0.011, 0.059) | (0.006, 0.020) | (-0.008, 0.006) | (0.78, 0.97) |
| Winter (DJF) | 0.010 | 0.006 | 0.00 | 0.998 |
|  | (0.006, 0.035) | (0.002, 0.023) | (-0.005, 0.007) | (0.98, 0.999) |
| Summer (JJA) | 0.078 | 0.045 | 0.00 | 0.92 |
|  | (0.038, 0.16) | (0.027, 0.075) | (-0.037, 0.024) | (0.78, 0.97) |

For the high-error and median-error simulations, with unrealistically high flotation fraction values, prediction error is highest during the spring pressure peak, when simulated values are least reasonable (Fig. 7, 8). For test simulations with lower error, and at times of year where flotation fraction values are more realistic, emulator predictions are closer to simulated values. The 95% prediction intervals mostly overlap the simulated values (Fig. 8), suggesting the emulator has reasonably accounted for interpolation and basis truncation error. In the higher-error simulation, however, the prediction intervals do not overlap the simulation outputs in spring, when the mean prediction significantly overestimates flotation fraction (Fig. 8a1, b1). In other words, the emulator has amplified the unrealistically high GlaDS flotation fraction in the case of the 95th-percentile RMSE test simulation.

The GP emulator predicts flotation fraction fields significantly faster than running GlaDS directly (Table 4). Each GlaDS run takes ∼24 minutes, and with simulation ensemble in hand, it takes 47 minutes to fit the emulator by drawing 5000 hyperparameter samples. From that point on, emulator predictions of GlaDS outputs take $< 2$ seconds on the same hardware using 32 posterior samples, or 22 s using 512 posterior samples, allowing for a denser exploration of the GlaDS input space.

In section 2.1, effective pressure, water pressure, and hydraulic potential were mentioned as alternative variables to flotation fraction. For the flotation fraction emulator, performance varies when predictions are subsequently re-expressed in terms of each of these variables (Fig. 9). Prediction performance as measured by the coefficient of variation $R^2$ is slightly worse when flotation fraction predictions are subsequently converted into effective pressure ($R^2 = 0.936$) compared to using flotation fraction directly ($R^2 = 0.961$), while predictions are best if they are converted into hydraulic potential ($R^2 = 0.993$). The higher coefficient of variation obtained by converting predictions into hydraulic potential suggests that $\phi$ is the weakest of the three indicators of GP prediction performance. Since converting emulator outputs changes the coefficient of variation, any subglacial drainage emulation application should identify the most relevant variable to model. For example, if the output of the subglacial emulator will be used for ice-flow coupling with an effective pressure-dependent basal slow law, the subglacial

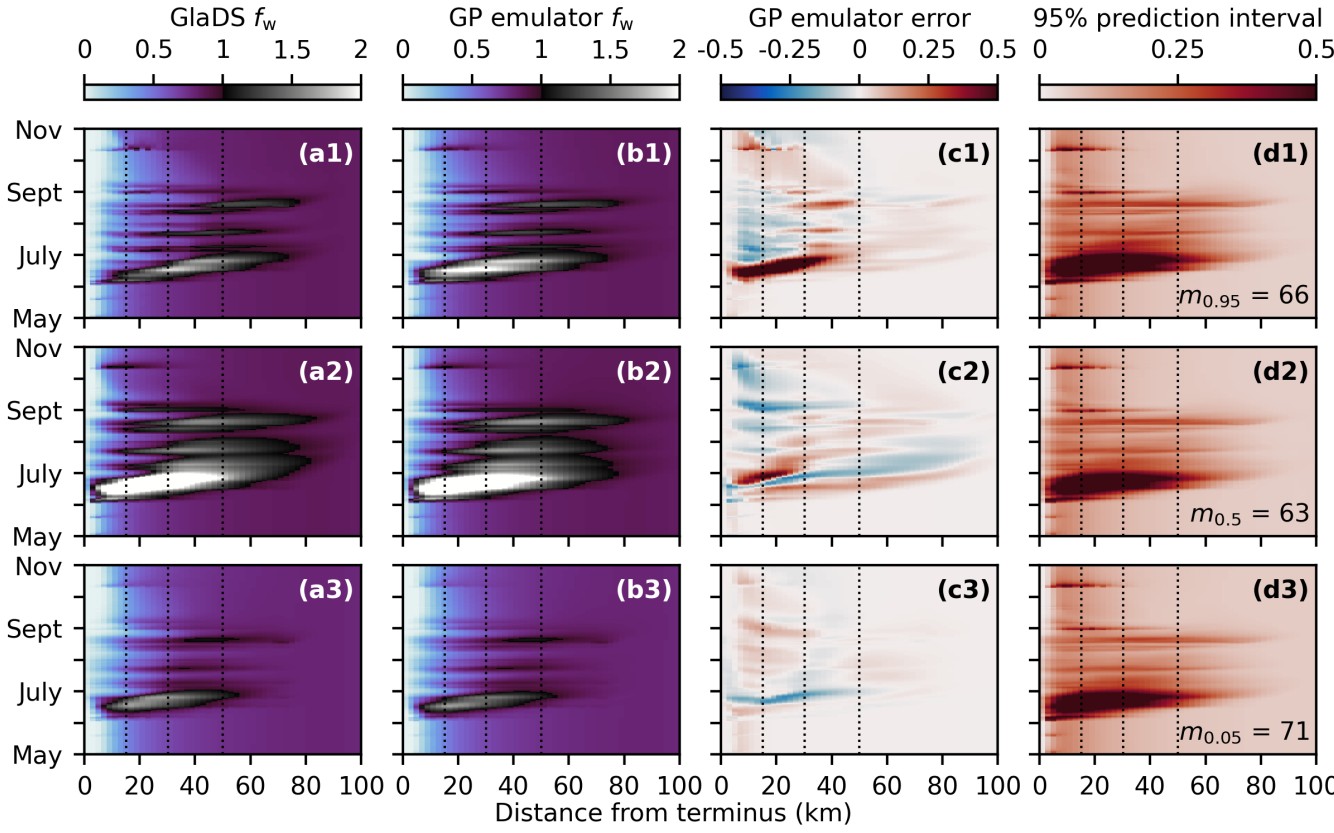

**Figure 7.** Width-averaged seasonal evolution of simulated (a) and emulated (b) flotation fraction $f_w$, GP prediction error (c) and width of the 95% prediction interval (d). Predictions are compared for three input settings corresponding to high (95th-percentile, row 1), median (50th-percentile, row 2) and low (5th-percentile, row 3) prediction error. Rows are labelled with the index of the corresponding simulation (i.e., the index of the test simulation with median error is $m_{0.5} = 24$) and associated parameter values are listed in Table B1.

**Table 4.** Computation time for GlaDS simulations, PC representation of the ensemble of simulations, estimation of GP hyperparameters (i.e., training) and GP prediction for the reference emulator with 8 PCs and the 256-member training ensemble. Computations were timed on AMD Rome 7532 CPUs on the Digital Research Alliance of Canada Narval cluster.

| Task | Computation time (HH:MM:SS) |
|------|------------------------------|
| Single GlaDS simulation | 00:25:00 |
| 512 simulation ensemble | 216:00:00 |
| Principal components | 00:00:18 |
| MCMC sampling (5000 draws) | 00:47:01 |
| GP prediction | 00:00:1.4 |

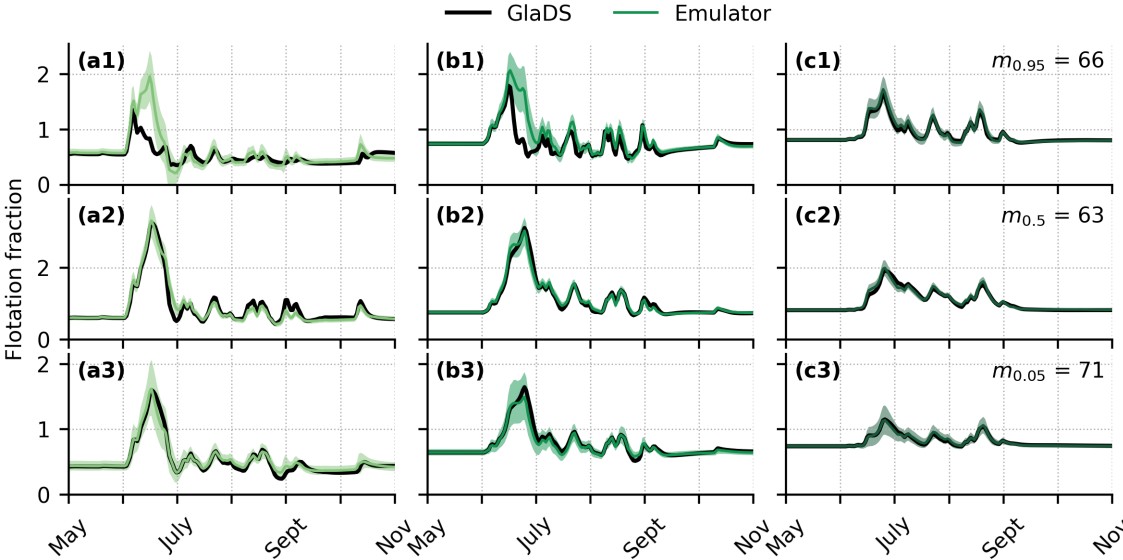

**Figure 8.** Timeseries of GlaDS simulated floatation fraction and mean GP emulator predictions on the test set with shaded 95% prediction intervals for nodes along the glacier centerline at (a) 15 km, (b) 30 km, and (c) 50 km from the terminus. Timeseries are extracted for predictions with the 95th-percentile (row 1), median (row 2) and 5th-percentile (row 3) RMSE. Rows are labelled with the index of the corresponding simulation and associated parameter values are listed in Table B1.

emulator should be trained to predict effective pressure $N$ directly based on an ensemble of effective-pressure fields, rather than computing effective pressure from flotation fraction or hydraulic potential.

### 4.3   Emulator performance for scalar variables

In addition to the flotation fraction emulator described above, we explore GP emulation of the three scalar variables. Taking the same strategy as with the flotation fraction emulator to assess the computation time–prediction performance trade-off for
these aggregate variables, we fit GPs using subsets of the simulation ensemble (Fig. 10). These scalar variables are emulated directly with univariate GPs so we do not need to tune the appropriate number of PCs.

As expected, adding simulations to the training ensemble improves prediction accuracy and decreases prediction uncertainty for each of the scalar variables. The slow rate at which prediction error decreases for 128–512 simulations suggests that 128 simulations may be sufficient for predictions of the scalar variables. For consistency with the flotation fraction emulator, we use
$m = 256$ training simulations. We obtain the lowest percent error for the channel discharge fraction emulator (MAPE 5.2%), with similar percent error for the sheet transit time (8.8%) and channel network length emulators (9.0%; Table 5). For all three scalar variables, the MAPE is higher than for the flotation fraction field (2.0%). However, it is important to consider that the flotation fraction field consists of many closely spaced, correlated time steps and nodes and a long winter period with minimal

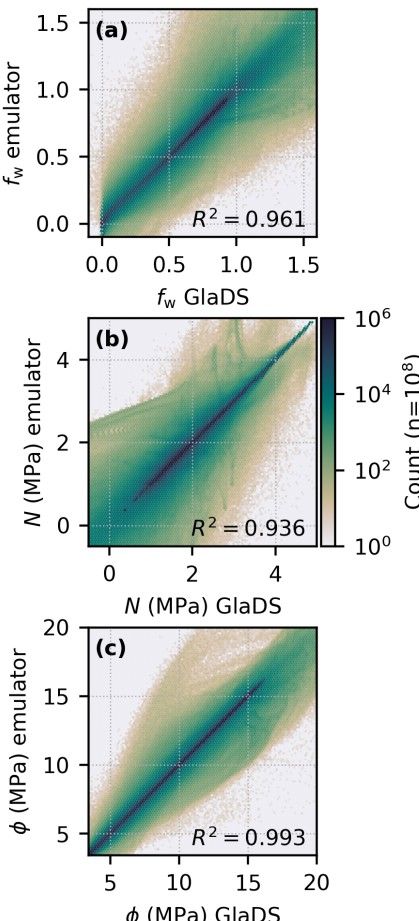

**Figure 9.** Pointwise comparison of simulated and emulated variable values, labelled with the coefficient of determination ($R^2$). (a) Flotation fraction $f_w$ (this study). (b) Flotation fraction re-expressed as effective pressure $N$ and (c) hydraulic potential $\phi$. Since the elevation potential is constant for the flat synthetic bed topography, performance would be identical between water pressure and hydraulic potential. Note the logarithmic colour scale.

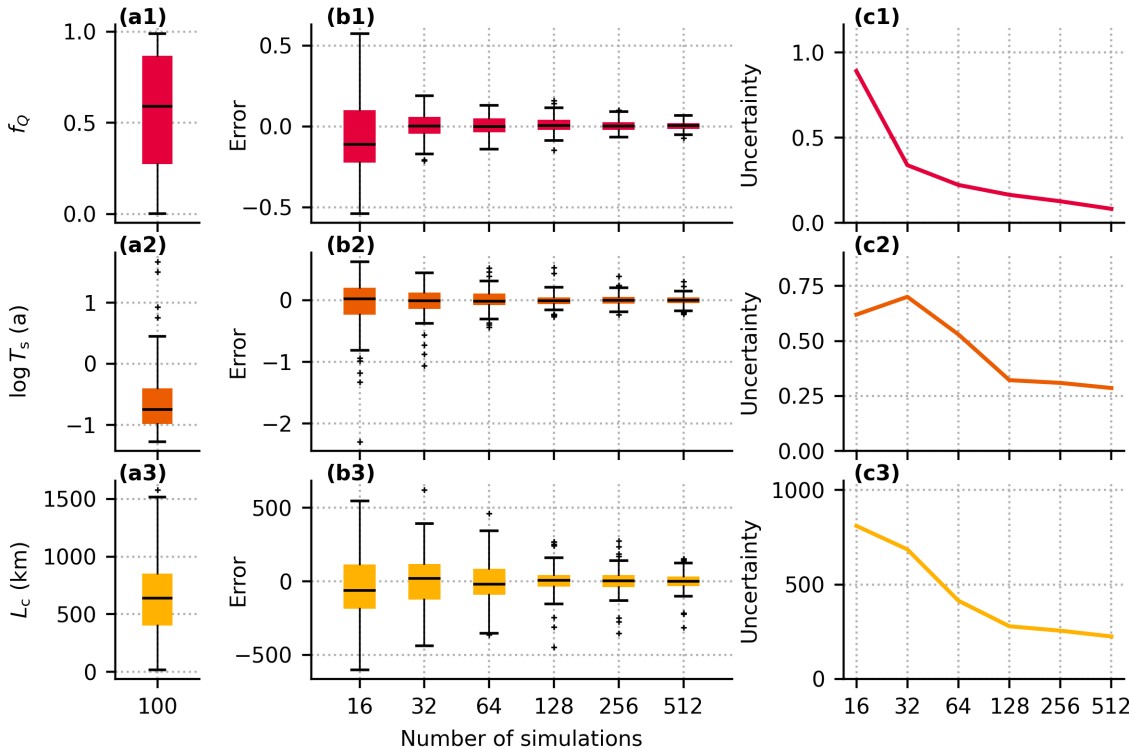

**Figure 10.** GP prediction performance for channel discharge fraction ($f_Q$, row1), log sheet transit time ($\log T_\mathrm{s}$, row 2) and channel network length ($L_\mathrm{c}$, row 3). (a) Distribution of simulated values from the test ensemble. (b) Prediction error evaluated on the 100 test simulations. (c) Mean 95% prediction interval.

**Table 5.** Emulator performance statistics using 256 training simulations on the 100-member test set for channel discharge fraction $f_Q$, sheet transit time $\log T_\mathrm{s}$, and channel network length $L_\mathrm{c}$ variables. The range of values from the test simulation ensemble is provided to indicate the scale of each variable to interpret the error metrics.

| Variable | Range | Units | RMSE | MAPE (%) | Median error (bias) | (5%, 95%) error |
|---|---|---|---|---|---|---|
| $f_Q$ | [0.0015, 0.99] | — | 0.034 | 5.2 | 0.0029 | (-0.041, 0.059) |
| $\log T_\mathrm{s}$ | [-1.28, 1.67] | a | 0.097 | 8.8 | $-7.5\times10^{-5}$ | (-0.14, 0.18) |
| $L_\mathrm{c}$ | [16, 1581] | km | 90 | 9.0 | -0.89 | (-116, 126) |

variance which may influence the error metrics. With this caveat in mind, the MAPE suggests it is not necessarily more difficult
to model the spatiotemporal field itself than scalar descriptions of drainage configuration and efficiency.

## 5 Sensitivity analysis

As an application of the fast emulator, we investigate the model parameters and processes that most strongly control the seasonal flotation fraction fields and the scalar variables using variance-based first-order and total sensitivity indices (e.g., Sobol', 2001; Saltelli et al., 2007). These global indices, sometimes called Sobol' indices, differ from the common practice of carrying out one-at-a-time sensitivity tests previously used for subglacial drainage models (e.g., Werder et al., 2013; Dow, 2022; Khan et al., 2024) by varying all parameters simultaneously. We compute both first-order sensitivity indices and total sensitivity indices to capture the individual and cumulative effects of each parameter.

The first-order effect of parameter $x_i$ on $\mathbf{Y}$ quantifies how varying $x_i$ alone controls the output $\mathbf{Y}$, while averaging out the effect of all other parameters. The first-order sensitivity index for parameter $x_i$ (e.g., Saltelli et al., 2007) can be written

$$S_i = \frac{\mathrm{Var}\left(\mathbb{E}_{\mathbf{X}_{\sim i}}(\mathbf{Y}|\mathbf{X}_i)\right)}{\mathrm{Var}(Y)}, \tag{14}$$

where $0 \leq S_i \leq 1$ and a value close to 1 indicates that the parameter $x_i$ is important. The notation $\mathbb{E}_{X_{\sim i}}$ means the expectation over all parameters except $x_i$. A low value near 0, however, does not indicate that $x_i$ is unimportant, since its influence may come in the form of interactions with other groups of parameters. We therefore compute the total effect of parameter $x_i$ on $\mathbf{Y}$. The total effect, quantified by the total sensitivity index $S_{T_i}$,

$$S_{T_i} = 1 - \frac{\mathrm{Var}\left(\mathbb{E}(\mathbf{Y}|\mathbf{X}_{\sim i})\right)}{\mathrm{Var}(\mathbf{Y})}, \tag{15}$$

considers all ways that $x_i$ may influence the output $\mathbf{Y}$, such as through pairwise or higher order interactions. We use the recommended estimators in Table 2 of Saltelli et al. (2010) to compute the indices. Confidence intervals for the sensitivity estimates are computed by bootstrap resampling. Confidence intervals extending $> 1$ are a result of numerical errors in the estimators, which are only guaranteed to converge in the limit of infinite simulation runs (Saltelli et al., 2010).

These methods are generally expensive, requiring thousands of simulations or more to accurately estimate sensitivity (Saltelli et al., 2010), depending on the dimensionality of the input space. Here, we use a total of 5120 simulations to quantify the sensitivity indices, which would require $\sim 100$ CPU-days using GlaDS directly. Using the emulator, the computation takes $\sim 4$ h. As with all sensitivity metrics, it is important to highlight that these sensitivity indices are dependent on the prescribed range of each model input (Table 2), with the ranges in this study chosen based on requirements placed on model outputs (section 3.2.2).

### 5.1 Flotation-fraction field

These sensitivity indices are well-defined for scalar model outputs but are not directly applicable to the modelled flotation-fraction fields. Given the principal component-based GP emulator, it is natural to apply these sensitivity methods to each individual principal component (e.g., Lamboni et al., 2011; Xiao et al., 2017). For the flotation fraction field, we recover $p \times d$ sensitivity indices, specifying the groups of parameters that most strongly control each principal component. A generalized sensitivity index measuring the total impact of each input on the output field is computed as the sum of the sensitivity indices

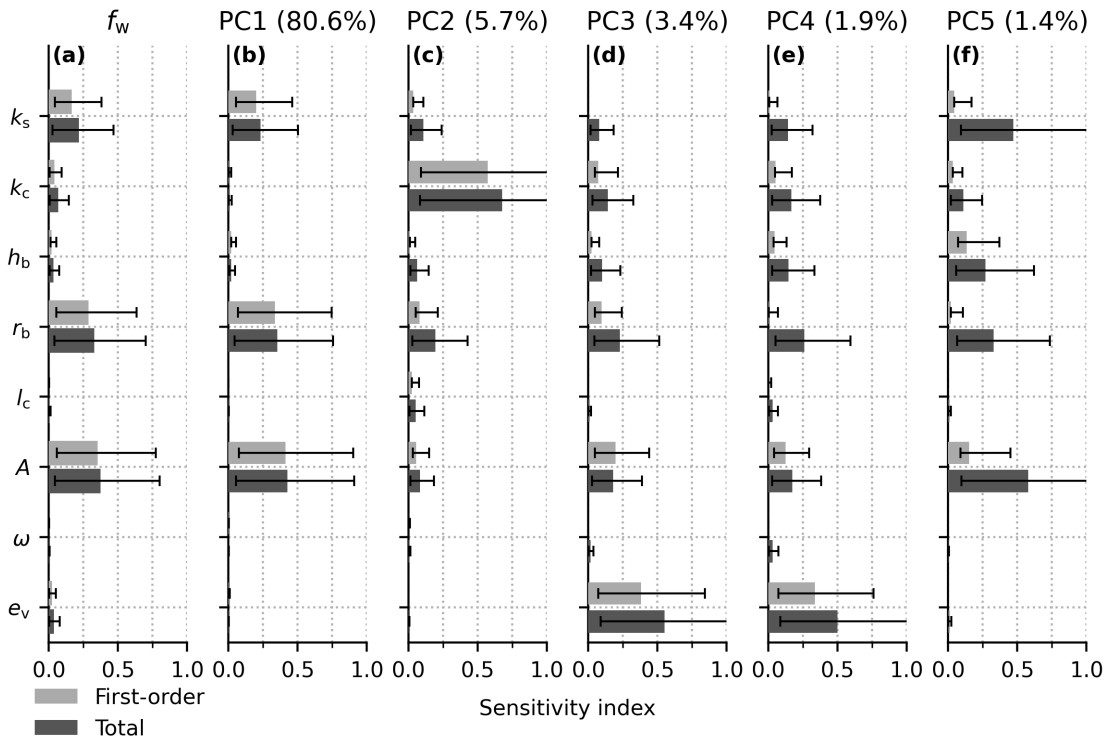

**Figure 11.** First-order and total sensitivity indices (bars) with bootstrap confidence intervals for the eight GlaDS parameters (Table 2). (a) Full flotation fraction field. (b–f) For the five PCs that individually explain at least 1% of variance in the flotation fraction field (b–f).

for each of the $p$ PCs weighted by the squared singular values (Lamboni et al., 2011; Xiao et al., 2017). The first-order and total sensitivity indices for the first five PCs reveal that a few distinct groups of parameters control each PC (Fig. 11b-f). The first PC is most sensitive to the sheet conductivity ($k_s$), bed bump aspect ratio ($r_b$) and the ice flow-law coefficient ($A$). These variables control the sheet capacity and hydraulic gradient in the absence of channels, supporting the interpretation of PC1 as representing water pressure in the absence of surface melt inputs (Fig. 3). The second PC is most sensitive to channel

conductivity with minimal sensitivity to other parameters, providing some evidence that PC2 represents summer water pressure controlled by channelized drainage. PC3 is most sensitive to the englacial storage parameter ($e_v$), with small sensitivity to most other parameters. Since $e_v$ controls the amplitude and duration of summer pressure peaks, PC3 appears to represent corrections to the amplitude and duration of summer pressure fluctuations controlled by englacial storage. PC4 and PC5 have less definitive first-order sensitivity with higher total sensitivity indices, suggesting they are driven by interactions between parameters.

Since PC1 explains ∼80% of the total variance, the sensitivity profile of the full flotation fraction field (Fig. 11a) resembles the sensitivity of PC1. Only the sheet conductivity, bed bump aspect ratio and ice flow-law coefficient have sensitivity indices greater than 0.1. The weak sensitivity to channel conductivity, bed bump height and englacial storage suggests that these

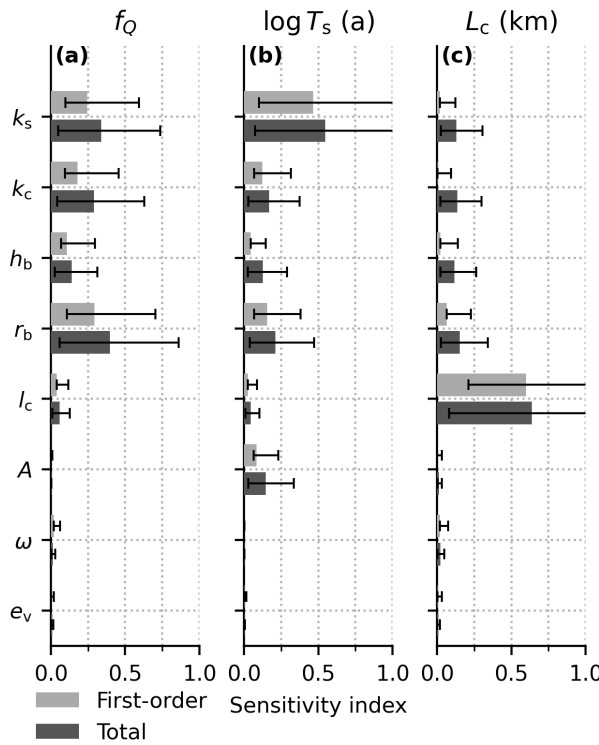

**Figure 12.** First-order and total sensitivity indices (bars) with bootstrap confidence intervals for the effect of the eight GlaDS parameters (Table 2) on channel discharge fraction $f_Q$ (a), log sheet transit time $\log(T_s)$ (b) and channel network length $L_c$ (c).

parameters together contribute to minor corrections to the flotation fraction field related to channelization and summer pressure fluctuations.

## 5.2   Scalar variables

Each scalar quantity is sensitive to a large group of parameters, with distinct sensitivity profiles between outputs (Fig. 12, B5). In terms of physical processes, the channel discharge fraction appears to be controlled by the rate of cavity opening, as evidenced by the high sensitivity to the bed bump aspect ratio, $r_b$. Sheet transit time is most sensitive to the sheet conductivity, $k_s$, perhaps due to the linear scaling between conductivity and flow velocity. The extent of the channel network is controlled by

the rate of channel initiation through $l_c$. We see very weak sensitivity of both the flotation fraction field and the scalar variables to the transition parameter $\omega$, suggesting it might be reasonable to remove $\omega$ from the ensemble by holding it at a physically realistic value.

## 6 Discussion

### 6.1 What is the fidelity of the subglacial drainage model emulator?

The GP emulators capture the essential features of seasonal variations in modelled subglacial hydrology on a synthetic domain. Predicted flotation fraction fields exhibit similar timing, amplitude and duration of variations as GlaDS simulations, with the highest discrepancy in the early melt season (Fig. 8). Across the space of model parameters, prediction errors are small relative to the variations across the ensemble of simulations: prediction RMSE is $< 20\%$ of the standard deviation of the 100-member test ensemble. The flotation-fraction emulator, which captures 97% (90–99%) of the variance of GlaDS simulations (Table 3), therefore has sufficient fidelity to resolve the influence of model parameter values on spatiotemporally resolved flotation fraction, with similar results for the scalar variables describing drainage configuration and efficiency. Emulator performance has not been assessed when extrapolating outside of the range of parameters used for training the model. For predictions far outside the training range, the zero-mean GP that we have used will revert to predicting the mean of the ensemble of simulations, likely producing significantly higher error than we have found on the test data. Predictions should therefore only be made within the parameter ranges used in the GlaDS simulation ensemble.

Prediction accuracy is limited by both GP error (reflecting the finite number of training simulations) and the error associated with truncating the basis after at most 11 PCs. Of the two error sources, PC truncation error is the larger contributor for the $p = 8$ PCs and $m = 256$ training runs used for the reference emulator. PC truncation RMSE on the test set for the reference model with 8 PCs is 0.034 (Fig. 3a1). GP prediction RMSE, representing truncation error and error in the GP predictions of the PCs, is 0.055, suggesting the PC truncation error contributes more than half of the total prediction error (Fig. B3). Increasing the number of principal components from 8 to 11 reduces basis truncation error (Fig. 3a1) but only slightly reduces prediction error (Fig. 5a–c), indicating that the balance between basis truncation error and GP error depends on the number of principal components used. The upper bound on prediction performance introduced by the truncation error suggests that alternative methods for multivariate simulation outputs (e.g., Liu et al., 2020) may be helpful in constructing emulators of spatiotemporally resolved subglacial drainage model outputs if significantly improved prediction accuracy is needed. Given the spatiotemporal density of GlaDS outputs, local approximations (e.g., Gramacy, 2016) may be an attractive alternative.

### 6.2 What is the trade-off between CPU investment in generating training data and emulator fidelity?

Based on the diminishing returns in prediction accuracy obtained by including more simulations (Fig. 4), using 128–512 simulations appears to be an appropriate trade-off between emulator performance and the investment in generating training data ($\sim$50–210 CPU-hours) for the emulation tasks considered here. This ensemble contain a large volume of data: the ensemble consists of up to 512 simulations, each with 365 time steps and $\sim$4000 nodes. Some improvement can be obtained with additional simulations, since the emulator will have closer neighbouring simulations to constrain predictions for new input values. However, since only a modest reduction in error is obtained while doubling computation time, this may not be worth the cost for all applications.

Once the ensemble of $\sim$128–512 simulations has been run, fitting the GP takes less time than a single additional simula-
tion (Table 4). For any applications requiring more than $\sim$128–512 simulations, constructing the GP emulator is worthwhile
since additional GP mean predictions are obtained in $< 2$ seconds. These results, however, are specific to the geometry and
experimental design that we have used.

## 6.3   What are the spatial and temporal scales of emulator prediction errors?

For the reference emulator, predictions recover spatial patterns in flotation fraction better than they capture variations in time
(Fig. 6). The time-integrated spatially distributed error patterns show that the highest emulator errors occur within a few km
of moulins (Fig. 6a–c). These localized errors would be partially averaged out if the emulated fields were used as part of
the basal boundary condition for ice-flow modelling, given that ice-flow models act as a low-pass filter with a wavelength
threshold of several ice thicknesses (e.g., Kamb and Echelmeyer, 1986; Joughin et al., 2004). The impact of large errors in
predicting the spring pressure maximum may be reduced for ice-flow modelling applications if the difference is only in the
amplitude and not the duration of the pressure maximum. Both the GlaDS simulations and the emulator predict water pressure
exceeding overburden (i.e., $f_{\mathrm{w}} > 1$; Fig. 7). Ice-sheet models would typically cap effective pressure, and therefore restrict
flotation fraction, in order to ensure basal drag does not become negative (e.g., $f_{\mathrm{w}} \leq 0.94$, Ehrenfeucht et al., 2023; Verjans
and Robel, 2024). By contrast, errors in emulator predictions with lower flotation fraction values, or errors in the duration of
water pressure exceeding the prescribed cap, would propagate through the ice-sheet model to produce discrepancy to some
extent in modelled velocity fields relative to using GlaDS directly (Verjans and Robel, 2024).

## 6.4   How does the GP emulator compare to neural network emulators?

Table 6 compares characteristics of the GP emulator presented here to previous neural network-based models (Brinkerhoff et al.,
2021; Verjans and Robel, 2024). In terms of prediction skill, the GP emulator has similar performance to the model of Verjans
and Robel (2024), who report the coefficient of variation ($R^2$) between simulated and emulated hydraulic potential fields of
0.96–0.998. When we convert our flotation fraction predictions into hydraulic potential, we obtain $R^2 = 0.993$. Relative to the
neural network models, the GP has fewer parameters to fit ($p(d+1)+1 = 73$ for the reference model). The small number of
parameters may help avoid overfitting without requiring additional steps such as bootstrap aggregation (e.g., Brinkerhoff et al.,
2021).

Verjans and Robel (2024) suggest that, without varying GlaDS parameters, it is more difficult to predict hydraulic poten-
tial for ice thickness and bed elevation fields that the emulator has not seen during training than for different surface melt
forcing timeseries. While significant differences in model setup, emulator architecture and experimental design prevent a di-
rect comparison between our study and theirs, the similar performance that the GP achieves when generalizing to new values
of eight model parameters suggests that the parameter generalization task is of a similar difficulty to domain or melt input
transferability.

Compared to a neural network, which typically predicts a single flotation fraction field without an accompanying prediction
uncertainty, the GP emulator predicts a distribution of plausible flotation fraction values (e.g., shaded intervals in Fig. 8).

This foundation in uncertainty quantification is important for parameter inference tasks (e.g., Brinkerhoff et al., 2021), for example, since the resulting posterior distributions of model parameters derived using the emulator will be broader than would be obtained using the numerical model directly (e.g., Downs et al., 2023). Prediction uncertainty can be approximated for neural network models (e.g., Gawlikowski et al., 2023), but it is an inherent component of the GP emulator.

We have chosen to force GlaDS and emulate its output at daily resolution, resulting in high variance in flotation fraction timeseries. Smoother, averaged melt inputs (e.g., monthly, Table 6) would likely lead to reduced PC truncation error and therefore more accurate GP predictions since GlaDS simulations tend to have smaller variations in time and between simulations with lower-frequency melt inputs (Hill et al., 2024b). For certain applications, daily resolution is excessively fine. For example, for projecting long-term changes in subglacial hydrology forced by climate model projections, daily resolution may be incompatible with running 100–300-year simulations, despite the fact that daily resolution is important for eliciting realistic water-pressure responses (e.g., Werder et al., 2013). For other applications, such as constraining the model using timeseries observations of borehole or moulin water pressure, the amplitude and phase of daily variations may be important.

The GP emulator approach that we have described is closest in spirit and in practical applications to that of Brinkerhoff et al. (2021). By emulating model outputs for different model parameter values, the GP emulator constructed in this study and the Brinkerhoff et al. (2021) neural network emulator are well-suited for quantifying parametric uncertainty, calibrating model parameters given data and exploring parameter sensitivity (e.g., Fig. 11). Both approaches use a principal component decomposition that nicely introduces interpretability for the emulator (e.g., Fig. 3, 11). Aside from structural differences in the type of emulator, the major differences between our work and that of Brinkerhoff et al. (2021) is that we explicitly resolve subglacial water pressure and drainage characteristics and we obtain a built-in prediction uncertainty estimate, whereas Brinkerhoff et al. (2021) implicitly represent subglacial conditions through the influence on surface velocities and take extra steps to estimate prediction uncertainty. Both approaches are tied to a particular study area, limiting their utility for large-scale forward modelling. On the other hand, Verjans and Robel (2024) use a convolutional neural network that can generalize to arbitrary melt forcing and study areas, making it an ideal tool for forward modelling of ice-sheet evolution forced with a basal boundary condition that is influenced by the hydrology emulator. Since Verjans and Robel (2024) do not predict water pressure for different model parameters, their emulator is not ideally suited for uncertainty quantification, calibration of drainage model parameters or sensitivity analysis.

None of the subglacial drainage model emulators have yet included structural constraints or constructed loss functions to enforce governing equations or other physical constraints. For example, Jouvet and Cordonnier (2023) use a loss function that is based on conservation of momentum as part of a neural network ice-flow velocity emulator. Such "physics-informed" models have improved skill for interpolating between sparse observations and extrapolating outside the training data, but come with additional cost and complexity (e.g., Lai et al., 2024). Considering the discontinuous nature of the channelized drainage system and the tendency of GlaDS to produce unrealistically high water pressure, it remains an open problem to apply physical constraints, such as mass conservation, to the subglacial drainage model emulation task and determine the applications which would benefit from such constraints.

**Table 6.** Comparison of subglacial drainage model emulation studies

| Criteria | Brinkerhoff et al. (2021) | Verjans and Robel (2024) | This study |
|---|---|---|---|
| | *Emulator configuration* | | |
| Emulator architecture | Artificial neural net | Convolutional neural net | Gaussian Process |
| Principal components retained | 50+ PCs to retain 99.99% variance | N/A | 8 PCs based on prediction error and uncertainty |
| Emulator inputs | 5 subglacial drainage ($k_s$, $k_c$, $h_b$, $r_b$, $e_v$) and 3 sliding-law parameters (friction coefficient $\gamma^2$, friction exponents $p$, $q$) | Bed topography, ice thickness, ice-surface velocity, surface melt rate aggregated over 6 preceding epochs | 8 GlaDS parameters ($k_s$, $k_c$, $h_b$, $r_b$, $l_c$, $A$, $\omega$, $e_v$) |
| Emulator output | Multi-year average logarithmic surface velocity | Spatially and seasonally resolved hydraulic potential | Spatially and seasonally resolved flotation fraction; scalar channel discharge fraction, sheet transit time, channel network length |
| Training minimization objective | Area-integrated squared velocity misfit | Surface velocity-weighted squared hydraulic potential misfit | GP posterior, (Eq. 8) |
| Number of training simulations | 5000 | 8 (each simulation spanning 1970–2009) | 256 |
| Spatially transferable | No | Yes | No |
| Temporally transferable | N/A (multi-year average) | Yes | No |
| Physics or conservation law constraint | None | None | None |
| | *Subglacial drainage model configuration* | | |
| Subglacial drainage model | Custom GlaDS implementation with updated boundary conditions, cavity opening rate and numerical discretization | GlaDS (ISSM) | GlaDS (ISSM) with laminar–turbulent sheet-flow parameterization (Hill et al., 2024b) |
| Model domain(s) | Russell Glacier, western Greenland | 8 Greenland outlet glaciers including Russell Glacier | Synthetic outlet glacier |
| Surface melt rate forcing | 1992–2015 monthly modelled mean runoff | 30-day moving average dEBM modelled melt rate | Daily mean air temperature and temperature-index melt model |
| Surface meltwater input locations | Spatially distributed source | 30 random positions per 100 km × 100 km image | 68 (synthetic) and 171 (Greenland) random positions with elevation-dependent density |
| | *Coupling and calibration data* | | |
| Ice-flow model coupling | Two-way: first-order ice-flow model with power-law sliding | One-way: Provide simulated and emulated $N$ fields to determine basal sliding | None |
| Calibration data | Multi-year annual average inSAR-derived surface velocity | None | None |
| | *Emulator performance* | | |
| Evaluation variables and metrics | Annual-average surface velocity: percent error | Hydraulic potential: $R^2$ and RMSE | Flotation fraction: RMSE, MAPE and prediction uncertainty |
| Computation time | Not listed | GlaDS simulation: 859.9 hours. Prediction: 1.0 hours | GlaDS simulation: 0.5 hours. Prediction: 2.1 s |

## 6.5 Limitations

The PC-based multivariate GP emulator introduces additional truncation error and prediction uncertainty when only a small number of PCs ($p = 8$) are retained. Performance can be minimally improved by running more training simulations and adding more PCs, but the additional cost may not be worth the modest improvements. While the interpretability of the PC-based emulator is attractive, the choice of using GlaDS parameters as the only emulator inputs combined with the basis representation produces an emulator that is not transferable to different domains or melt inputs since the fixed basis encodes both (c.f., Verjans and Robel, 2024).

Relative to the GlaDS simulations, the GP loses some fidelity, especially in the spring melt event and in capturing the local influence of moulins on water pressure. For some parameter combinations, the 95% prediction intervals nearly span the full range of temporal variations in melt-season flotation fraction. These predictions, associated with high-variance portions of the input space or being far from training simulations, include a wide range of plausible flotation fraction timeseries, suggesting there are some cases where the emulator may not be suitably constrained by the training data.

The synthetic ice-sheet domain excludes realistic variations in bed topography and ice thickness which influence the location and configuration of channelized drainage and the spatial patterns of flotation fraction. The simple spatial patterns of flotation fraction in our simulations may partially explain the relatively higher prediction skill in space than in time (Fig. 6). This may not be the case for transferable emulators, as Verjans and Robel (2024) found their emulator had higher prediction skill generalizing to new melt forcing timeseries than new geometries. Future case studies should explore the application of GP emulation of the subglacial hydrology of real ice-sheet basins.

## 6.6 Applications and considerations

When should an emulator be used instead of the numerical model directly? Since each simulation in the ensemble used to train the emulator is independent, the ensemble of simulations is efficiently parallelized, so that generating training data is not a serious bottleneck in absolute time. Applications for subglacial drainage emulation include: inverse problems (computing distributions of model inputs or parameters that produce model outputs consistent with data), forward sensitivity problems (i.e., propagating distributions of input parameters through the model to derive distributions of output variables) and forward modelling where aggregating the subglacial drainage modelling cost offline is more convenient (e.g., ice-sheet modelling using emulated effective pressure as an input to a sliding law).

The number and type of emulator inputs (e.g., scalar model parameters, model input fields such as topography and melt forcing) and outputs are another important consideration since they will influence the flexibility and complexity of the emulator. Increasing the number of emulator inputs will degrade performance for a given computational budget for running ensembles, especially if the variables to be emulated are sensitive to the additional inputs, since adding dimensions to the input space reduces the density of samples. Does a specific task require predicting large fields (e.g., predicting basal effective pressure as an input for ice-sheet model simulations)? Or is it sufficient to predict scalar quantities or a measure of model–observation

mismatch (e.g., Downs et al., 2023)? Predicting scalar fields is an easier problem, especially for GP emulators, and may require less time to train the emulator and fewer training simulations for the same prediction performance.

The contribution of the GP emulator model constructed here is primarily methodological since it has been applied to a synthetic ice-sheet outlet glacier. These particular methods are ideally suited for Bayesian calibration of model parameters given the probabilistic formulation (e.g., Hill et al., 2024a). Since we do not include an ice-flow component (c.f., Brinkerhoff et al., 2021), calibration data would need to consist of quantities that can be derived from GlaDS outputs directly, such as basal water pressure measured in boreholes (e.g., Hubbard et al., 1995; Irarrazaval et al., 2021; Rada Giacaman and Schoof, 2023), moulin water level (e.g., Andrews et al., 2014; Hoffman et al., 2016), tracer transit times (e.g., Irarrazaval et al., 2021), or channel characteristics inferred from passive seismic measurements (e.g., Nanni et al., 2021).

Emulated GlaDS effective-pressure fields could be used as inputs to a sliding law (e.g., Verjans and Robel, 2024), reducing the computational cost of ice-sheet model runs forced by varying subglacial effective pressure. We see this type of coupling as integral for assessing the uncertainty in ice flow and solid ice discharge that is sourced from uncertainty in subglacial drainage characteristics and model parameters. For instance, the sensitivity analysis from Section 5 could be extended to assess the contribution of subglacial drainage model parameters to variations in modelled solid-ice discharge for marine-terminating outlet glaciers. Additional uncertainties introduced by modelling ice flow, such as the choice of a sliding law (Gagliardini et al., 2007; Zoet and Iverson, 2020; Gilbert et al., 2023) and the relevant rheology of ice (e.g., Millstein et al., 2022; Schohn et al., 2025), urge caution in such a coupled approach. The same steps that we have outlined in Section 2 and Fig. 1 to construct the flotation fraction emulator can be used to emulate any vector-valued model output as a function of scalar model parameters. The uncertainty-aware GP emulator methods used here provide one path forward for acknowledging and quantifying uncertainties in a range of glaciological processes.

## 7    Conclusions

We have described a Gaussian Process emulator for spatially resolved subglacial hydrology at daily resolution. The emulator uses eight parameters of the physics-based model GlaDS as inputs and, using a principal component decomposition for the multivariate model outputs, predicts water pressure normalized by ice-overburden (flotation fraction). For simulations with the GlaDS model that run in $\sim 0.5$ h, each GP prediction can be as fast as $< 2$ s, with one-time overhead costs of $\sim 5$ CPU-days to run the 256-member training ensemble and $\sim 6.5$ minutes to compute the principal component decomposition and sample the GP hyperparameters. Emulator predictions have 2.5% average error, with locally higher errors in the early melt season and near moulins.

In addition to spatiotemporally resolved flotation fraction fields, we have explored emulating scalar variables that describe subglacial drainage morphology, such as the total length of the channel network, and efficiency, including the fraction of channelized drainage and the transit time through the distributed drainage system. Based on computing global sensitivity indices, made tractable by the fast emulators, the flotation fraction field and these scalar drainage variables are influenced by distinct sets of model parameters. The flotation fraction field is most sensitive to the ice-flow coefficient ($A$), the bed bump

aspect ratio ($r_b$) and the sheet conductivity ($k_s$). In contrast, flotation fraction is insensitive to the sheet-width below channels ($l_c$) and the laminar–turbulent transition parameter ($\omega$), suggesting these parameters are unlikely to be constrained by data.

There is no universally optimal emulator. The emulator described here is well-suited to uncertainty quantification and model
calibration objectives given its built-in estimate of prediction uncertainty, but it does not generalize to different melt forcings or glacier geometries (c.f., Verjans and Robel, 2024). It also relies on a truncated principal component representation of the multivariate GlaDS outputs that introduces additional error and uncertainty. The GP emulator nonetheless unlocks a wealth of future research possibilities, including Bayesian calibration of subglacial drainage models and probabilistic approaches to ice-flow modelling driven by emulated subglacial drainage.

*Code and data availability.* Code, simulation outputs, trained emulators and example notebooks are available at https://doi.org/10.5281/ zenodo.14933592 under MIT software license (Hill et al., 2025). GP emulators were constructed using the SEPIA package v1.1 (Gattiker et al., 2020), available at https://github.com/lanl/SEPIA/ via BSD-3 license. The Ice-sheet and Sea-level System Model (ISSM) v4.24 used for GlaDS simulations is available at https://issm.jpl.nasa.gov/ (Larour et al., 2012) under BSD 3-Clause license. Air temperature data are available from PROMICE at https://doi.org/10.22008/FK2/IW73UU (How et al., 2022).

**Appendix A: Subglacial drainage model governing equations**

The GlaDS model solves for hydraulic potential $\phi$, water sheet geometry described by sheet thickness $h_s$ and subglacial channel geometry described by the cross-sectional area $S$ based on conservation of mass combined with sheet and channel discharge parameterizations (Schoof et al., 2012; Hewitt et al., 2012; Werder et al., 2013). The model is posed on an unstructured triangular mesh where the continuum description of the distributed water sheet is applied over the two-dimensional domain
and subglacial channels are defined along the network of one-dimensional channel edges.

Using the sheet-flow parameterization described by Hill et al. (2024b), which allows flow to transition between laminar and turbulent regimes depending on the local Reynolds number, the distributed water sheet evolves according to

$$-k_s h_s^3 \nabla \phi = \boldsymbol{q}_s + \omega \mathrm{Re} \boldsymbol{q}_s \tag{A1}$$

$$\frac{\partial h_s}{\partial t} = f_b \frac{h_b - h_s}{r_b h_b} u_b - \tilde{A} h_s |N|^{n-1} N \tag{A2}$$

$$\frac{e_v}{\rho_w g} \frac{\partial \phi}{\partial t} + \nabla \cdot \boldsymbol{q}_s + f_b \frac{h_b - h_s}{r_b h_b} u_b - \tilde{A} h_s |N|^{n-1} N = m_s, \tag{A3}$$

where $\boldsymbol{q}_s$ is the discharge-per-unit-width in the water sheet, $\mathrm{Re} = \frac{|\boldsymbol{q}_s|}{\nu}$ is the Reynolds number for kinematic viscosity $\nu$, $f_b$ is a switch that turns off the sliding-opening term when $h_b > h_s$, $m_s$ is the basal melt rate, $\tilde{A} = \frac{2}{n^n} A$ is a geometric scaling of the ice flow-law coefficient and remaining parameter names and values are listed in Table 2. We have written the sliding-opening parameterization (Eq. (A2)) in terms of the bed bump aspect ratio $r_b$, rather than the bed bump length $l_b$ used in the original
GlaDS formulation (Werder et al., 2013), where the aspect ratio $r_b = \frac{l_b}{h_b}$ is the ratio of the bump length to the bump height.

The subglacial channel network is governed by the set of equations

$$Q = -k_c S^{\alpha_c} \left| \psi_c \right|^{\beta_c - 2} \psi_c \tag{A4}$$

$$\frac{\partial Q}{\partial s} + \frac{\Xi - \Pi}{L} \left( \frac{1}{\rho_i} - \frac{1}{\rho_w} \right) - \tilde{A} S |N|^{n-1} N - m_c = 0 \tag{A5}$$

$$\frac{\partial S}{\partial t} = \frac{\Xi - \Pi}{\rho_i L} - \tilde{A} S |N|^{n-1} N \tag{A6}$$

$$\Xi = \left| Q \frac{\partial \phi}{\partial s} \right| + \left| l_c q_c \frac{\partial \phi}{\partial s} \right| \tag{A7}$$

$$\Pi = -c_t c_w \rho_w (Q + f l_c q_c) \frac{\partial}{\partial s} (\phi - \phi_m), \tag{A8}$$

where $Q$ is the channel discharge, $\psi_c = \frac{\partial \phi}{\partial s}$ is the along-channel potential gradient, $\frac{\partial Q}{\partial s}$ is the along-channel discharge gradient, $\Xi$ and $\Pi$ represent potential energy dissipation and sensible heat changes, $m_c$ is a channel source term representing inflow from the adjacent water sheet, $f$ is a switch that disables refreezing when it would produce negative channel areas and remaining parameter names and values are listed in Table 2. Equations (A4–A8) are applied to individual channel segments, with the channel network assembled by connecting individual channel segments at nodes and enforcing mass conservation, with no storage permitted at nodes and including possible mass inputs from moulins.

The governing equations (A1–A8) are solved using finite elements. By integrating over the nodes, edges and elements of the mesh, the sheet–channel exchange term ($m_c$) is computed implicitly by assuming the hydraulic potential $\phi$ is continuous between the sheet and channel systems (Werder et al., 2013). The model allows for an arbitrary combination of Dirichlet boundary nodes with fixed hydraulic potential and Neumann boundary nodes with specified boundary fluxes.

## Appendix B

**Table B1.** Parameter values corresponding to test simulations with 5th-percentile, median and 95th-percentile RMSE (GlaDS parameters defined in Table 2).

| Parameter | 95th-percentile (66) | Median (63) | 5th-percentile (71) | Units |
|---|---|---|---|---|
| $k_s$ | 0.37 | 0.12 | 0.031 | $\mathrm{Pa\,s^{-1}}$ |
| $k_c$ | 0.31 | 0.059 | 0.16 | $\mathrm{m^{3/2}s^{-1}}$ |
| $h_b$ | 0.064 | 0.063 | 0.36 | m |
| $r_b$ | 75 | 39 | 25 | — |
| $l_c$ | 1.3 | 28 | 41 | m |
| $A$ | $2.1 \times 10^{-24}$ | $2.9 \times 10^{-24}$ | $1.2 \times 10^{-24}$ | $\mathrm{s^{-1}Pa^{-3}}$ |
| $\omega$ | $1.2 \times 10^{-3}$ | $1.3 \times 10^{-4}$ | $1.4 \times 10^{-3}$ | — |
| $e_v$ | $1.2 \times 10^{-4}$ | $1.3 \times 10^{-4}$ | $2.7 \times 10^{-4}$ | — |

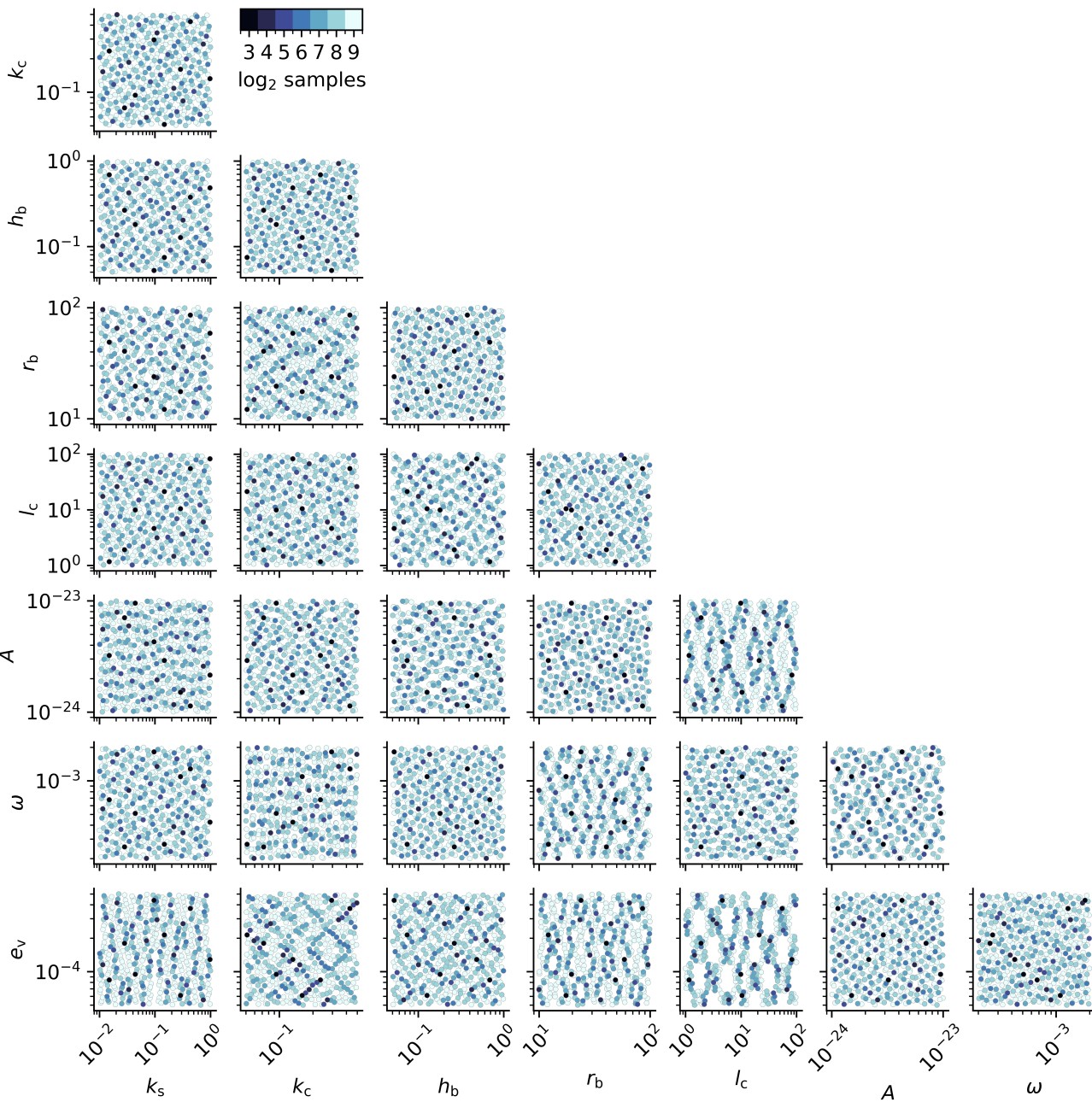

**Figure B1.** Pairwise two-dimensional projections of the parameter design matrix. Markers are coloured by each $2^k$ subset for $k = 3, 4, \ldots, 9$.

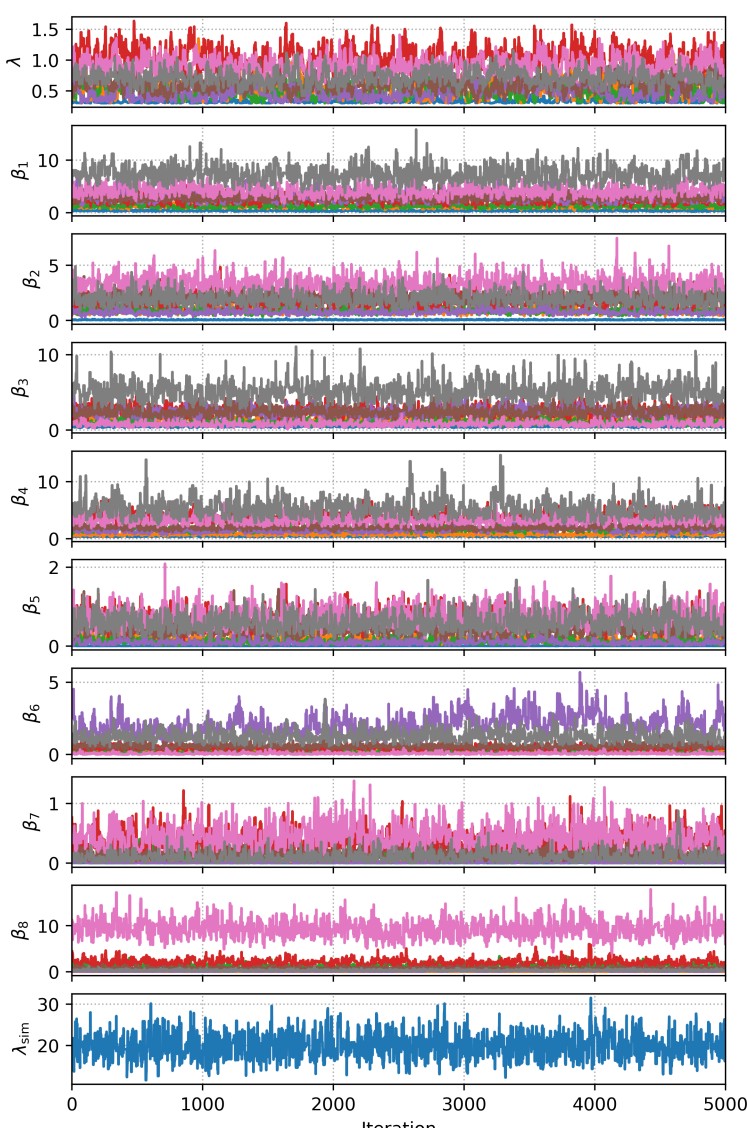

**Figure B2.** MCMC trace plot of 5000 samples of the GP hyperparameters for the reference emulator ($m = 256$ training simulations with $p = 8$ principal components), where $\lambda$ is the GP precision for each of the $p$ independent GPs, the $\beta_i$ are the GP rate parameters for each of the GlaDS parameters and $\lambda_{\mathrm{sim}}$ is the simulator precision associated with the principal component truncation error. The first 2500 samples are discarded as a warm-up. Note that parameters $\lambda$ and $\beta_1, \ldots, \beta_8$ have traces associated with each of the $p = 8$ principal components. Convergence of the MCMC chains is evaluated by the potential scale factor reduction $\hat{R}$ diagnostic, which measures the within- and between-chain correlation, and the effective sample size, which reduces the sample size based on the autocorrelation, using 4 chains with random starting points (Gelman et al., 2013). The median $\hat{R}$ across GP hyperparameters and PCs is 1.008 (min 1.001, max 1.04), where $\hat{R} < 1.1$ is often used to indicate convergence. The median effective sample size is 576 using 4 chains (min 75, max 1562), indicating that most chains are converged based on the empirical threshold of 10 independent samples per chain (Gelman et al., 2013). The acceptance rates are between 31%–52%.

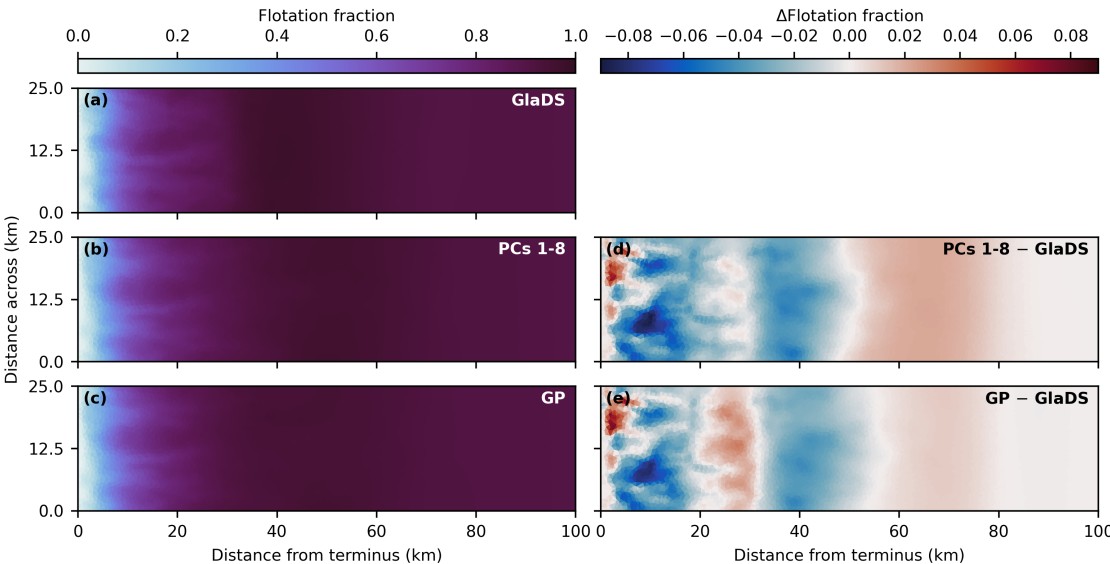

**Figure B3.** Principal component truncation error and GP prediction error. (a) GlaDS-simulated flotation fraction for the test simulation with median RMSE on 29 July, (b) corresponding principal component representation of the GlaDS flotation fraction using 8 PCs, and (c) Gaussian Process (GP) emulator prediction. Difference maps show the principal component representation (d) and the Gaussian Process emulator prediction (e) minus the GlaDS output.

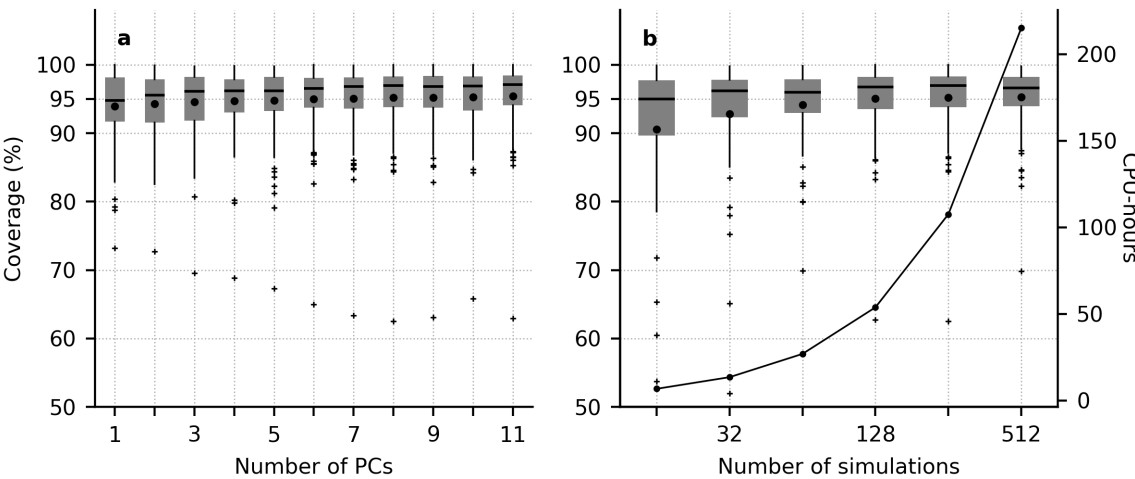

**Figure B4.** Proportion of GP emulator-predicted flotation fraction values that overlap the simulated values within the 95% prediction interval across the test set. (a) For emulators constructed using different numbers of principal components. (b) For emulators using subsets of the training ensemble. Horizontal lines indicate the median and black dots indicate the mean across the test set. The right axis in (b) indicates the number of CPU-hours associated with running each subset of the GlaDS training ensemble. Note the logarithmic x-axis in (b).

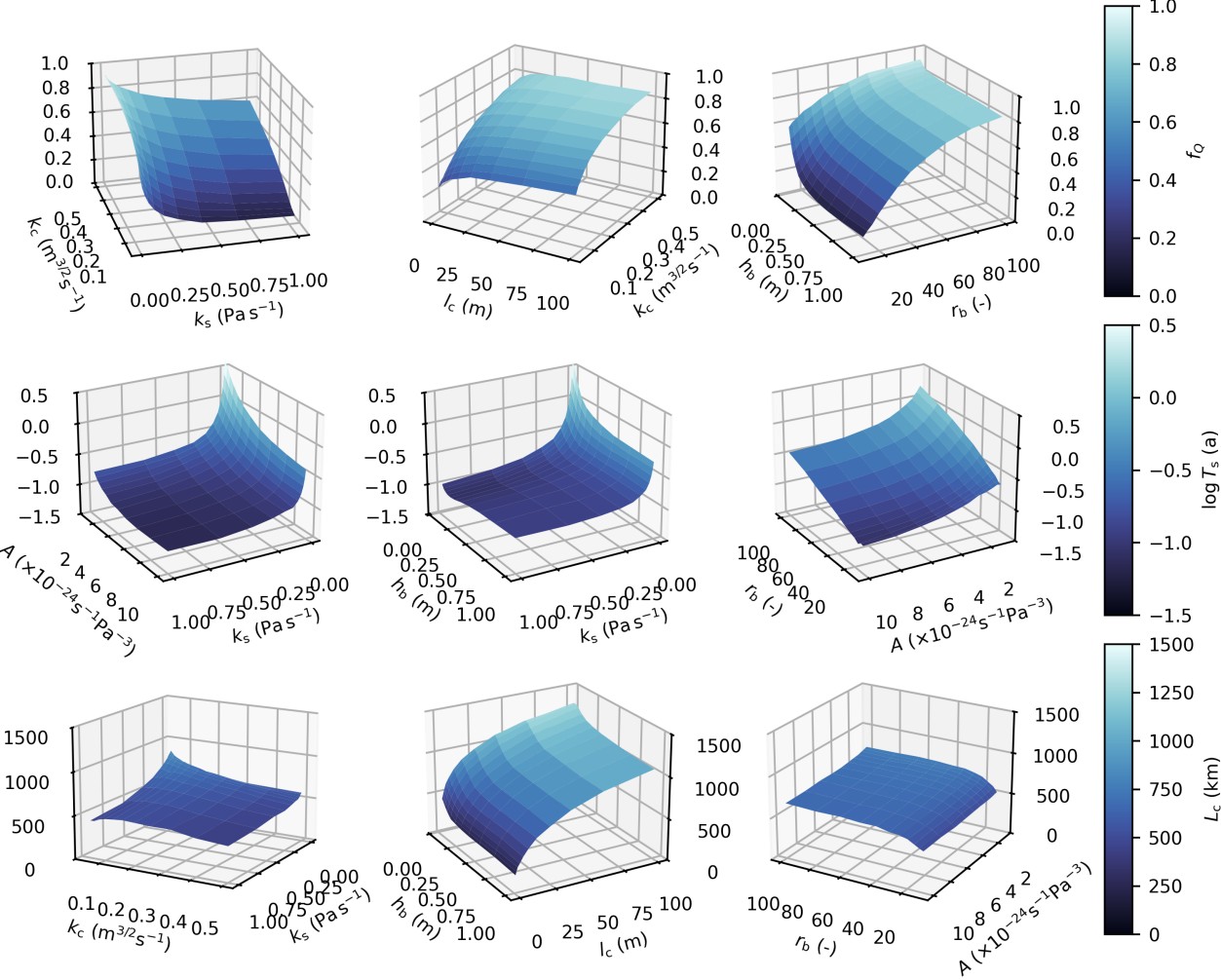

**Figure B5.** Select pairwise mean channel discharge fraction ($f_Q$), log transit time ($\log T_s$), and channel length ($L_c$) response surfaces (GlaDS parameters defined in Table 2). Note the different vertical and colour scales between each model output.

*Author contributions.* DB and GF conceived of the idea of statistically emulating subglacial drainage models. DB and TH developed the statistical methodology. GF, MH and TH designed the GlaDS experiments. TH ran the simulation ensembles, fit emulators and analyzed outputs. All authors interpreted outputs. TH prepared the manuscript with contributions from DB, GF and MH.

*Competing interests.* The authors declare that they have no conflict of interests

*Acknowledgements.* TH was supported by Simon Fraser University and the Natural Sciences and Engineering Council of Canada (NSERC) Canada Graduate Scholarship program. GF and DB received support from the NSERC Discovery Grants program. This research was enabled in part by support provided by WestDRI (https://training.westdri.ca), Calcul Quebec (https://www.calculquebec.ca/) and the Digital Research Alliance of Canada (https://alliancecan.ca). Support for MH was provided through the Scientific Discovery through Advanced Computing (SciDAC) program funded by the U.S. Department of Energy (DOE), Office of Science, Advanced Scientific Computing Research and Biological and Environmental Research Programs. The authors thank Vincent Verjans, Jacob Downs and an anonymous reviewer for comments on an earlier version of this manuscript.

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
