# Peer review of "Computationally efficient subglacial drainage modelling using Gaussian Process emulators: GlaDS-GP v1.0"

_EGUsphere, 2024_

## Referee Comment (RC1)

Review of "Computationally efficient subglacial drainage modelling using Gaussian Process emulators: GlaDS-GP v1.0" by Hill et al.
Reviewer: Vincent Verjans

This study develops a Gaussian Process (GP) emulator to emulate output from the subglacial hydrology model GlaDS (Werder et al., 2013). In particular, the GP is trained to reproduce the sensitivity of flotation fraction output to 8 different GlaDS parameters. A principal component truncation is performed to reduce the dimensionality of the outputs to be emulated. Training is performed on an idealized glacier configuration, with a pre-specified melt input forcing. The performance of the emulator is then evaluated on 100 test combinations of GlaDS parameters, unseen during the emulator training.

This study contributes positively to efforts towards computationally efficient solutions to simulate subglacial hydrology. It also offers a promising tool to evaluate parametric uncertainty of subglacial hydrology models. This latter aspect is important, since subglacial hydrology models are heavily parameterized, with very few physical constraints on parameter values. I value positively the technical approach used for the emulator development. On the idealized configuration tested here, the emulator shows a good performance on non-training data samples. The manuscript is clearly structured and well-written. I have nonetheless a concern regarding the impact of the study. The authors have developed a subglacial hydrology emulator, but the real scientific value of this lies in the implications for areas where subglacial hydrology plays a role, many of which are provided as motivations in the introduction. As presented, both the emulator performance and the potential for uncertainty quantification are hard to interpret, because no application of the emulator is demonstrated. I detail this concern in my Major comment below, and I emphasize that this lack of impact (1) is my personal opinion, and it is the editor who decides which impact is expected from studies published in Geoscientific Model Development, and (2) does not influence my positive opinion about the quality of the work performed by the authors, but only on what more could be done. My review further includes a Minor comment regarding the quantitative evaluation of the emulator, and Technical comments aiming to improve the structure and clarity of the manuscript. Line numbers in this review correspond to the preprint manuscript.

**Major comment: Impact**
As mentioned in my introduction, emulation and uncertainty quantification of subglacial hydrology by themselves are not of great scientific interest. It is really the implications of subglacial hydrology for different fields, primarily ice flow modeling but also others listed in the introduction, that make it a critical research topic. However, none of these implications is explored here. As such, I feel like the prediction performance and the potential for uncertainty quantification from the GP emulator are not very meaningful as presented.
I note that GlaDS has been run with the Ice-sheet and Sea-level System Model (ISSM, Larour et al., 2012). As such, it should not be a big step to compute ice flow simulations (1) using the GlaDS output and (2) using the emulator output in order to evaluate the implications of emulator performance on modeled ice flow. Furthermore, it would be very interesting to see differences in modeled ice flow across the range of GlaDS parameters investigated in this study. This would really demonstrate the benefits of uncertainty quantification of subglacial hydrology when it comes to modeling ice flow velocities. Even though this study focuses on a single idealized glacier and melt forcing configuration, such ice sheet model experiments would be a great contribution to constraining ice flow uncertainty caused by subglacial hydrology.
If the authors are concerned about the length of the manuscript if such experiments are included,

I would recommend reconsidering the inclusion of the simulations of scalar quantities ($f_Q$, $T_s$, and $L_c$). In my view, these experiments are not of great relevance, as I do not see which research area would benefit from predictions and uncertainty quantification of these variables.

Again, I repeat here that the decision of sufficient impact from this study for publication in Geoscientific Model Development is ultimately a decision of the editor. I express here my personal opinion. And I emphasize that the work presented in this manuscript is of good quality, with only a single Minor comment and some Technical comments that I provide below.

**Minor comment: Quantitative evaluation**

The title of Section 6.1 is "What is the fidelity of the subglacial drainage model emulator?". In my view, this question has not been evaluated thoroughly enough. I think that simply adding a table with important evaluation metrics would be sufficient to address this concern. Evaluation metrics could be averaged spatio-temporally as well as across the 100 test simulations. It would be insightful to provide 5th, 50th, and 95th percentiles of RMSE, MAPE, coefficient of determination ($R^2$), and bias across the 100 test simulations, where these metrics are time- and spatially-averaged. In addition, it would be nice to add the same metrics but (i) for the upper and lower 30 km parts of the domain separately, and (ii) for the DJF and JJA months separately. Such evaluation metrics would give the reader a better and more quantitative appreciation of the performance of the GP emulator. Finally, for each of these metrics, I recommend also providing between parentheses the same metric but computed on the training data. This would be insightful to evaluate the potential degradation of the GP emulator performance when used on inputs unseen during training.

**Technical comments**

- General (1): The authors make an excessive use of parentheses throughout the text. In my view, parentheses should only be used to provide additional non-essential details in the text. I recommend that the authors clean up their parentheses by making more sentence separations instead of overloading single sentences.

- General (2): Throughout the manuscript, the authors use both the terms "inputs" and "parameters" to refer to the same notion: the parameters of GlaDS passed to the emulators. To avoid any confusion, a single term should be used consistently in the entire manuscript.

L5
Replace "construct robust" by evaluate uncertainty in.
L6
uncertainty quantification.
L14
"of the water pressure variance": it is unclear if this refers to spatial variance, temporal variance, and/or variance across the samples of the parameter space.
L15
I believe that the mention to observational data is misused here, as no observational data is integrated in this study.
L25
"well-established": I understand what the authors mean here. However, this wording is misleading, because although there is consensus about the existence of an influence of subglacial hydrology on ice flow, this influence remains highly uncertain.
L26
Replace flow by sliding.

L31-36

This sentence is too long, and I do not know what "which" (L36) refers to.

L36

Replace large by high-dimensional.

L52

Typo: "GP emulators we develop".

L53

I suggest this definition for flotation fraction: ratio of water pressure to ice-overburden pressure.

L55

Please explain here the meaning of "global sensitivity indices".

L63

If possible, please use another word than "emerging".

L69

Please provide units of variables.

Eq. (1)

Although obvious, please define $g$.

L77

" each of these quantities defines a two-dimensional, time-varying field": this is confusing because I believe that both $z_b$ and $p_i$ are not time-varying.

L83

Please specify: For details about GPs

L85

I do not understand what is meant by: "in terms of the proportion of output variance corresponding to each GlaDS parameter". Please clarify.

L91

Rephrase: Let $\boldsymbol{y}_i$ denote the vectorized model output of all variables (...).

L94

Typo: "which are not a part of".

L104

Add a comma after $\boldsymbol{\theta}$.

L110

Refer to $\mu(\boldsymbol{x})$ after "mean function".

L110

"to set the mean to zero" should be to set the prior mean to zero. In the following sentences, it is also important to emphasize that it is only the prior mean that is zero.

Eq. (5)

This should be $y_p | \mathbf{Y}, \theta, x_p$

L124

I recommend being more specific here: (...) contains the pair-wise covariances between $x_p$ and each entry of $x$ (...).

L126

Specify: The prediction mean

L127

Refer to Eq. (4) after "covariance function".

L136 and L137

Replace "will" by would.

L138

Typo: "a variety solutions".

L148

Please specify here that Eq. (8) assumes uncorrelated errors. This may not be entirely valid in this case.

L152

"can be viewed as": this wording is inappropriate, because it is a dimension reduction by definition.

L162

The authors can also invoke the orthogonality property of the PC decomposition to motivate their univariate approach.

L169

Replace "permissive" by flexible.

L169

"variations in the principal components that tend to be smooth with respect to the input parameters": why is that? I would expect a strong sensitivity of GlaDS to some of its parameters, even in the PC subspace. Could the authors please clarify this statement?

L173

How many GP realizations are sampled?

L179-181

I am not sure to agree here. As I understand it, each univariate GP is fitted to a single series of PC coefficient, regardless of the number $p$ of PCs retained. The number of parameters scales linearly with $p$, being $p(d + 1) + 1$. However, the amount of data used for fitting also scales linearly with $p$, because increasing $p$ by 1 implies that one more series of PC coefficients is used. As such, I do not see why "a simpler model with fewer PCs and therefore fewer hyperparameters to estimate is desirable as it will have less prediction variance (i.e., less tendency to overfit)". On the other hand, I believe that using an increasingly high number of PCs would imply increasingly many GPs fitted to low-variance component of the GlaDS output, which can be regarded as noisy features of GlaDS results rather than dominant components of the variability.

Section 3.1

In general, I think that more details are needed in this Section.

L192

Add one sentence to explain what the K-transect is.

L192-193

"adjusted from 0 m" and "increased to 40–1560 m": are these adjectives with respect to the SHMIP configuration? If so, please specify.

L196

Please put the basal melt rate imposed into a glaciological context. For example, how does it compare with basal melt rate estimates in Greenland, or with the SHMIP forcing?

L199

" following a moulin density that varies with elevation computed from a satellite-derived supraglacial drainage map": is it possible to provide the formulation of the moulin density as a function of elevation?

L200

"within each sub-catchment": this is not explained.

Figure 1

Is it possible to add the melt rate using the right y-axis in Fig. 1a? What does the color scheme represent in Fig. 1b? Is it possible to indicate the moulin locations in Fig. 1d?

L203

Replace "posed" by configured.

L204

The variable $x$ is already used to denote the input to the GPs. Please do not use the same symbol for two different variables.
L216

"Following the vocabulary of Higdon et al. (2008) and Verjans and Robel (2024)": this is not needed here.
L222

Concerning the parameter ranges, the ranges provided in Table 2 are most likely not intuitive to a majority of readers. I recommend adding a column in Table 2 specifying the ranges of parameter values used in previous studies focused on uncertainty quantification from GlaDS parametric uncertainty (e.g., Brinkerhoff et al., 2021).
L223

"flotation fraction $f_w < -10$": in principle, any $f_w < 0$ is nonphysical because it implies $p_w < 0$. Are all these simulations rejected from the training data? And/or is the GP constrained to predict $p_w > 0$?
L234

As I understand, the test data do not include any extrapolation beyond the parameter space used for training. Therefore, it should be mentioned here and in the Discussion that the extrapolation capability of the GP emulator has not been evaluated.
L242

Please change this sentence to "In addition to emulating the spatiotemporal flotation fraction, (...)".
L277

Remove "small".
L279

"perhaps since the input space has been explored more thoroughly": I do not think this is the case. In my view, more PCs are needed simply because the rank of the output space increases. For example, if a single simulation is run, it is fully characterized by a single PC. As more simulations are included, the number of PCs required to fully characterize the outputs increases, and thus the number of PCs to characterize a given % of the output variance also increases.
L284

"only the absolute value, not the sign, of the PC basis vectors should be interpreted": I disagree. Opposite signs indicate opposite phasing of variability. It would be more correct to say that the sign of any given PC basis vector is arbitrary, but only looking at the absolute value would be wrong.
Figure 3

Specify if the lines show the mean or median of RMSE and MAPE taken across the test simulations.
Figure 4c,f

I think that the presentation of the 95% prediction intervals is both unclear and misleading. Firstly, I understand that the RMSE and MAPE boxplots show the errors averaged in both time and space. But for the 95% prediction intervals, do the boxplots show the entire population of 95% prediction intervals taken at each grid cell and each time step of each test simulation? Secondly, this Figure suggests that broad prediction intervals are a bad thing. However, the purpose of a prediction interval is to communicate about the uncertainty in the output. Thus, it is a good thing that prediction intervals are broader for cases with high RMSE (i.e., the simulations with low PC numbers in this Figure). This means that the true GlaDS value may still lie within the 95% prediction interval despite the larger error in the mean estimate. For this reason, I recommend to show the percentage of GlaDS values falling outside of the 95% prediction intervals in Figure 4c,f, rather than the 95% prediction intervals themselves. If the GPs are well-calibrated, this percentage should be 5%.

Figure 4d,e,f

Please mention in the caption that the x-axis uses a logarithmic scale.

Figure 4 caption

"Black circles indicate the total integrated prediction uncertainty": I do not understand this.

L320

I find that it is worth mentioning that the RMSE for the three GPs is very similar for the late-September melt event, and I suggest to provide a succinct explanation of why this is.

L321

Typo: "reduces by the height".

L335

Table C1 should be referenced here.

L337

Please explain why $f_w > 2$ is considered unrealistic.

L343

"suggesting the emulator has reasonably accounted for basis truncation error": please note that this also suggests that the GP can correctly estimate uncertainty due to interpolation towards unseen parameter values.

L345

Add comma after "spring".

L345

"the mean prediction significantly overestimates flotation fraction": this suggests that the GP tends to further amplify the unrealistic GlaDS output. Please mention this explicitly.

Table 3

Specify: Single GlaDS simulation. L353 and Figure 8

Please use coefficient of determination ($R^2$) as an evaluation metric, rather than the squared correlation coefficient ($r^2$).

L350-359 and Figure 8

These comparisons are misleading, because the GP emulator has been trained to reproduce $f_w$. It is impossible to know what the performance of the GP would be if it had been trained to reproduce $\phi$ or $N$. The discussion here should be rephrased as an evaluation of the error introduced by the conversion from $f_w$ to $\phi$ and/or $N$, rather than "indicators of GP prediction performance" or "different prediction skill" (L355).

Figure 9 column c

Same comment concerning the prediction intervals as for Figure 4c,f.

L368

"Based on RMSE, MAPE and bias": I am not sure that these performances can be compared so easily from Table 4. For example, RMSE and bias have different units for the three variables. MAPE could serve as a better comparison basis, but $T_s$ has been log-transformed, and MAPE may not be representative of the error on $T_s$ itself. In addition, as mentioned by the authors, the MAPE values depend on the degree of variability in each quantity and on the values themselves (e.g., the percentage error when $\log T_s$ is 0 tends to infinity). Finally, when considering the ranges of values provided in Table 4, it is not clear to me that "the channel discharge fraction emulator has the best performance".

L389

"Sensitivity indices for the flotation field are defined as a variance-weighted sum of the sensitivity indices for each principal component": This one-sentence explanation is not clear to me. If possible, I recommend providing the mathematical formulation for the sensitivity indices. That is, which formula is used to compute the values shown in Figure 10? Furthermore, the difference between

first-order and total sensitivity indices should be explained.

Figures 10 and 11

Some whiskers extend beyond the value of 1.0. While the value of 0 can be intuitively interpreted as no sensitivity, what do values >1.0 mean?

L418

"prediction RMSE is < 20% of the ensemble standard deviation": across the 100 test simulations?

L422

"PC truncation RMSE on the test set for the reference model with 8 PCs is 0.034, while GP prediction RMSE is 0.054, suggesting the PC truncation error contributes more than half of the prediction error.": I think that this statement requires a more thorough justification, and I am also not sure that I agree with the authors about it. First, does the 0.034 value correspond to the case of 8 PCs for the curve 256 simulations in Figure 2a1? If so, please refer to Fig. 2a1 in the text. Second, Figure 4 shows that there seems to be a baseline RMSE of the GP predictions of about 0.05, which does not decrease when going from 7 to 11 PCs. On the other hand, the PC truncation error must decrease when going from 7 to 11 PCs. As such, this indicates that there is a balance between (1) using only the first few PCs that seem to be relatively easy to predict for the GP, and (2) including low-variance PCs that allow to reduce the truncation error, but that seem to be harder to predict for the GP. As a consequence of this balance, the baseline RMSE stagnates at 0.05. But saying that "Of the two error sources, PC truncation error is the larger contributor" is misleading. I believe that if more PCs had been included, PC truncation error would decrease, but GP error would increase. Thus, this conclusion seems to be due to the choice of truncating at 8 PCs, rather than an inherent attribute of the GP. At least, this is how I understand the results. I would welcome any thoughts from the authors about this.

L434

"$5 - 10 \times 10^4$ time steps": why such a range? I thought that all simulations had been performed over the same time period and with the same temporal discretization.

L446

"The impact of large errors in predicting the spring pressure maximum is also reduced for ice-flow modelling applications": I disagree with this statement. For example, Fig. 4d (magenta curve) and Fig. S4c of Verjans and Robel (2024) show that the highest ice flow velocity errors due to the subglacial hydrology emulation occurs in the spring pressure maximum (their Fig. 4d) and at the ice velocity peaks (their Fig. S4c), which generally coincide with the spring pressure maximum. So, it is impossible to verify this claim from the authors if they do not actually compare ice flow model realizations forced with the GlaDS versus the GP output.

L454-455

" the model of Verjans and Robel (2024), who report squared correlations ($r^2$)": this is not correct. Verjans and Robel (2024) report the coefficient of determination ($R^2$), which is not the same metric as $r^2$.

L474

"since PC truncation typically preferentially dampens high-frequency variations": I disagree with the authors here. PC truncation selects the components of variability with maximum variance. If most of the variance lies in high-frequency bands of the spectrum, there would not be any damping of high-frequency variations. Whether PC truncations dampens high-frequency variability or not depends on the power spectrum of the data.

L477-479

It is also important for resolving sub-annual ice flow variability.

L487

I believe that it would be relevant to add a sentence here about the propensity of GlaDS to produce

nonphysical output.

L493

Citing Verjans and Robel (2024) here is misleading, because their emulator is transferable to different domains or melt inputs, i.e., the opposite of the sentence given here.

L521-528

I found this entire paagraph a little vague and hand-waving. I recommend that the authors focus on the current work and future developments. For example, why discussing emulator predictions of global mean sea-level rise? This is clearly not the focus of this study.

L542

Typo: "is is".

L548-549

"fully Bayesian time-dependent calibration to provide observationally constrained distributions of subglacial drainage variables": I do not understand what the authors mean here.

L561-562

Add regimes after "laminar and turbulent".

Eq. A3

To make the notation less clumsy, I suggest replacing the third and fourth terms on the left-hand-side by $\frac{\partial h_s}{\partial t}$.

L603

Please specify: of a multivariate Normal distribution.

Eq. B3

For this equation to be valid, there needs to be an additional constant term on the right-hand-side.

L616

Typo: "includes" should be include.

L625

Typo: "by condition" should be by conditioning.

Eq. B5

$\theta$ should be boldfaced.

L629

then sampling from equation (B5)

L634

a major benefit of using

L635

Can the authors please remind here what are $p$ and $d$ so that the reader does not need to go back tp the main text?

Figure C2

It seems to me that some of the Markov Chains are not well-mixed, although it is hard to tell from the scale of the y-axes. Did the authors compute convergence diagnostics? I recommend providing R-hat values and effective sample sizes (Gelman et al., 2013).

Figure C3

These figures should be shown in two dimensions rather than 3 for better clarity.

**References**

Douglas Brinkerhoff, Andy Aschwanden, and Mark Fahnestock. Constraining subglacial processes from surface velocity observations using surrogate-based bayesian inference. *Journal of Glaciol-*

*ogy*, 67(263):385–403, 2021.

Andrew Gelman, John B Carlin, Hal S Stern, David Dunson, Aki Vehtari, and Donald B Rubin. *Bayesian data analysis*. Chapman and Hall/CRC, 2013.

Eric Larour, Helene Seroussi, Mathieu Morlighem, and Eric Rignot. Continental scale, high order, high spatial resolution, ice sheet modeling using the ice sheet system model (issm). *Journal of Geophysical Research: Earth Surface*, 117(F1), 2012.

Vincent Verjans and Alexander Robel. Accelerating subglacial hydrology for ice sheet models with deep learning methods. *Geophysical Research Letters*, 51(2):e2023GL105281, 2024.

Mauro A Werder, Ian J Hewitt, Christian G Schoof, and Gwenn E Flowers. Modeling channelized and distributed subglacial drainage in two dimensions. *Journal of Geophysical Research: Earth Surface*, 118(4):2140–2158, 2013.

---

## Referee Comment (RC2)

**Review: "Computationally efficient subglacial drainage modelling using Gaussian Process emulators: GlaDS-GP v1.0" by Hill et al.**

Reviewer: Jacob Downs

This work outlines a Gaussian Process based emulator of the GlaDS subglacial hydrology model. Inputs to the emulator consist of 8 scalar model parameters, while outputs include either spatio-temporal flotation fraction fields or scalar metrics describing bulk properties of the subglacial drainage system. Performance of the emulator is thoroughly examined by comparing emulators using different training sets as well as using different numbers of principal components to represent flotation fraction fields.

Overall, this manuscript is well written, the methods are clear, and I appreciate the rigorous evaluation of the emulator's accuracy. In light of this, I believe this work represents a good contribution to the field. However, I believe that this work would benefit from a more detailed discussion of how the GP emulator compares to existing emulators. In particular, I think the discussion should outline more of the key differences between this work and Verjans and Robel [2024], including differences in their potential use cases.

The authors should outline in specific terms where their emulator could be applied versus other existing methods. For instance, is this purely a tool for calibrating subglacial hydrology parameters or assessing sensitivity? Is there a path for using this as a substitute for GlaDS to predict effective pressure when coupled with an ice sheet model? What do the authors intend to do with this emulator, or what might future work on the emulator entail? As someone interested in this space, I was hoping the authors might delve more deeply into some of these topics in the discussion.
* * *
**Line 52**

Might benefit from rewording to "The Gaussian Process emulators we develop take subglacial drainage model parameters as inputs and predict spatially and seasonally resolved flotation fraction ... "

**Line 63**

Maybe this could be reworded as as "where the radius of channels is modeled as a balance between the creep closure of ice and opening by melt. "

**Line 65**

"The continuum distributed (sheet) drainage system is defined on the mesh nodes, with possible channel locations defined by element edges."

This could be clarified. Maybe you could say that variables describing the distributed system are represented by linear finite elements with degrees of freedom on mesh nodes? This would help clarify that the distributed drainage system isn't "confined" to the mesh nodes, but that's where the degrees of freedom are defined.

**Line 83**

"For details on Gaussian Processes"

**Line 130**

Please mention why there are d+1 hyperparameters. Also, depending on the kernel, this number could vary.

**Line 170**

That seems reasonable. If I understand, would it be fair to say that you hope that much of the spatial / temporal complexity is captured in the principal, components while the response of the coefficients of the principal components to variations in parameters is expected to be smooth?

**Line 285**

Interesting, I like the commentary on the first couple principal components, and I think interpretability is a really nice advantage of this method.

**Line 321**

"Using 8 PCs reduces the height..."

**Figure 4**

I'm not sure what is meant by 95% prediction interval in c and f. Is this the prediction uncertainty of the emulator integrated through time and space? Also, when considering the prediction error, is this accounting for uncertainty in $\theta$ (sampled via MCMC) as well as the GP prediction uncertainty? Or is it just the latter, and you are using the most probable estimate for the hyperparameters? Presumably the advantage of using MCMC on $\theta$ as opposed to maximum likelihood estimation is to characterize its effect on uncertainty as well?

**Line 340**

Before discussing results for the test cases with different levels of error, please introduce what the "high-error" and "median-error" simulations are. I think I found this information in the figure 6 and 7 captions, but it would be helpful to include in in the text.

**Figure 9**

It's difficult to see the differences in the median error across different numbers of simulations due to the scale of the outliers.

**Section 6.4**

I think some of this discussion could be expanded to highlight more of the nuances of the different approaches. For example, the emulator in Verjans and Robel [2024] aims to be fairly general purpose, and its more of a one-to-one substitute for GLaDS. Hence, I see their neural network based approach as fundamentally different from your yours in its intent. This also makes the comparison of the number of parameters difficult as the input / output spaces in Verjans and Robel [2024] is very broad, meaning more parameters are likely needed.

There are certainly a number of appealing elements to your GP approach, including interpretability, the speed at which it can be trained, and how you also obtain uncertainty estimates without additional work. But I feel like its difficult to directly compare your emulator to Verjans and Robel [2024], and I have a hard time seeing the two emulators being used in the same way. In this sense, I see your emulator as being far more comparable to Brinkerhoff et al. [2021].

**Line 481**

To me, saying that IGM enforces conservation of momentum sounds as if it is enforced strictly. Although IGM uses a physics-based loss function, conservation of momentum is only upheld approximately. Contrast this with something like Horie and Mitsume [2024], in which a value is strictly conserved in a neural network.

**Section 6.6**

I think this manuscript would significantly benefit from elaborating on specific use cases for this emulator. For instance, do you see the emulator being used more or less as is, or do you think the value of this work is in the general approach that you present? You mention uncertainty in future sea level rise, but not a clear application of the emulator for this purpose.

There is a pretty clear use case for using the emulator to do Bayesian calibration of subglacial hydrology model parameters, but what other use cases might it have? Do you see a path forward for coupling effective pressure fields from the emulator to an ice sheet model? Clarifying the intended use of this emulator or more concrete pathways for other applications would really strengthen the discussion.

**References**

Douglas Brinkerhoff, Andy Aschwanden, and Mark Fahnestock. Constraining subglacial processes from surface velocity observations using surrogate-based bayesian inference. 67 (263):385–403, 2021. ISSN 0022-1430, 1727-5652. doi: 10.1017/jog.2020.112. URL `https://www.cambridge.org/core/journals/journal-of-glaciology/article/constraining-subglacial-processe` Publisher: Cambridge University Press.

Masanobu Horie and Naoto Mitsume. Graph neural PDE solvers with conservation and similarity-equivariance. 2024. URL `https://openreview.net/forum?id=WajJf47TUi`.

Vincent Verjans and Alexander Robel. Accelerating subglacial hydrology for ice sheet models with deep learning methods. 51(2):e2023GL105281, 2024. ISSN 1944-8007. doi: 10.1029/2023GL105281. URL `https://onlinelibrary.wiley.com/doi/abs/10.1029/2023GL105281`. _eprint: https://onlinelibrary.wiley.com/doi/pdf/10.1029/2023GL105281.

---

## Author Comment (AC1)

**Author Response to RC1: "Computationally efficient subglacial drainage modelling using Gaussian Process emulators: GlaDS-GP v1.0"**

Reviewer: Vincent Verjans

Tim Hill, Derek Bingham, Gwenn E. Flowers, Matthew J. Hoffman

Reviewer comments are in black and we provide our responses in blue.

This study develops a Gaussian Process (GP) emulator to emulate output from the subglacial hydrology model GlaDS (Werder et al., 2013). In particular, the GP is trained to reproduce the sensitivity of flotation fraction output to 8 different GlaDS parameters. A principal component truncation is performed to reduce the dimensionality of the outputs to be emulated. Training is performed on an idealized glacier configuration, with a pre-specified melt input forcing. The performance of the emulator is then evaluated on 100 test combinations of GlaDS parameters, unseen during the emulator training.

This study contributes positively to efforts towards computationally efficient solutions to simulate subglacial hydrology. It also offers a promising tool to evaluate parametric uncertainty of subglacial hydrology models. This latter aspect is important, since subglacial hydrology models are heavily parameterized, with very few physical constraints on parameter values. I value positively the technical approach used for the emulator development. On the idealized configuration tested here, the emulator shows a good performance on non-training data samples. The manuscript is clearly structured and well-written. I have nonetheless a concern regarding the impact of the study. The authors have developed a subglacial hydrology emulator, but the real scientific value of this lies in the implications for areas where subglacial hydrology plays a role, many of which are provided as motivations in the introduction. As presented, both the emulator performance and the potential for uncertainty quantification are hard to interpret, because no application of the emulator is demonstrated. I detail this concern in my Major comment below, and I emphasize that this lack of impact (1) is my personal opinion, and it is the editor who decides which impact is expected from studies published in Geoscientific Model Development, and (2) does not influence my positive opinion about the quality of the work performed by the authors, but only on what more could be done. My review further includes a Minor comment regarding the quantitative evaluation of the emulator, and Technical comments aiming to improve the structure and clarity of the manuscript. Line numbers in this review correspond to the preprint manuscript.

Thank you for the detailed review and sharing your expertise in this area. We appreciate your suggestions to improve the manuscript and plan to implement them in the revised version. We have responded to your comments individually below.

**Major comment: Impact**
As mentioned in my introduction, emulation and uncertainty quantification of subglacial hydrology by themselves are not of great scientific interest. It is really the implications of subglacial hydrology for different fields, primarily ice flow modeling but also others listed in the introduction, that make it a critical research topic. However, none of these implications is explored here. As such, I feel like the

prediction performance and the potential for uncertainty quantification from the GP emulator are not very meaningful as presented.

I note that GlaDS has been run with the Ice-sheet and Sea-level System Model (ISSM, Larour et al., 2012). As such, it should not be a big step to compute ice flow simulations (1) using the GlaDS output and (2) using the emulator output in order to evaluate the implications of emulator performance on modeled ice flow. Furthermore, it would be very interesting to see differences in modeled ice flow across the range of GlaDS parameters investigated in this study. This would really demonstrate the benefits of uncertainty quantification of subglacial hydrology when it comes to modeling ice flow velocities. Even though this study focuses on a single idealized glacier and melt forcing configuration, such ice sheet model experiments would be a great contribution to constraining ice flow uncertainty caused by subglacial hydrology.

If the authors are concerned about the length of the manuscript if such experiments are included, I would recommend reconsidering the inclusion of the simulations of scalar quantities ($f_Q$, $T_s$, and $L_c$). In my view, these experiments are not of great relevance, as I do not see which research area would benefit from predictions and uncertainty quantification of these variables. Again, I repeat here that the decision of sufficient impact from this study for publication in Geoscientific Model Development is ultimately a decision of the editor. I express here my personal opinion. And I emphasize that the work presented in this manuscript is of good quality, with only a single Minor comment and some Technical comments that I provide below.

Thank you for suggesting ways to make the work more interesting to a broader audience. However, we disagree with the statement that subglacial drainage modelling by itself is unimportant and uninteresting. For example, Ehrenfeucht et al. (2024) recently pushed physics-based subglacial drainage modelling forward by producing full-Antarctic GlaDS runs. That this work was published in GRL suggests community interest in subglacial drainage modelling. Moreover, recent work towards understanding the physics of subglacial drainage models (e.g., Sommers et al., 2023; Warburton et al., 2024), and constructing models with approximate physics in order to improve computational scaling (Kazmierczak et al., 2024) is evidence that model developments are a community priority. These latter works have highlighted open questions about drainage models that warrant attention separately from their application to ice-sheet modelling.

The suggestion to include an ice-flow component in the current work is an interesting idea that we have considered throughout the process of the larger project that this paper is one part of. We have not done so here because we believe that including an appropriately detailed and nuanced ice-flow application would be out of scope for this model description paper. Given uncertainty arising from the choice of a sliding law, the representation of basal drag would be a nontrivial addition. Moreover, forcing an ice-sheet model one-way with emulated effective pressure fields would miss important two-way hydrology–dynamics feedbacks (e.g., Hoffman et al., 2014), and running two-way coupled simulations remains non-trivial even with ISSM-GlaDS. It is also not clear how interesting the ice-flow application would be for the synthetic geometry that we have used. The uncertainty-aware emulation methods that we have used can be one part of accepting and quantifying these uncertainties, but this is out of the scope of the current work. We have summarized this discussion in an expanded Section 6.6 (Applications and Considerations).

Finally, we would like to highlight a recent preprint where we explore calibrating the subglacial drainage model with borehole water pressure data (https://doi.org/10.31223/X5GQ68). We had intended to include an ice-flow component in this work to allow for calibration with both borehole water pressure and seasonal surface velocities. However, this preprint identifies serious shortcomings in the model relative to the data, indicating that further work is needed on the subglacial drainage models themselves before their potential for ice-sheet modelling can be fully realized.

**Minor comment: Quantitative evaluation**
The title of Section 6.1 is "What is the fidelity of the subglacial drainage model emulator?". In my view, this question has not been evaluated thoroughly enough. I think that simply adding a table with important evaluation metrics would be sufficient to address this concern. Evaluation metrics could be averaged spatio-temporally as well as across the 100 test simulations. It would be insightful to provide 5th, 50th, and 95th percentiles of RMSE, MAPE, coefficient of determination ($R^2$), and bias across the 100 test simulations, where these metrics are time- and spatially-averaged. In addition, it would be nice to add the same metrics but (i) for the upper and lower 30 km parts of the domain separately, and (ii) for the DJF and JJA months separately. Such evaluation metrics would give the reader a better and more quantitative appreciation of the performance of the GP emulator. Finally, for each of these metrics, I recommend also providing between parentheses the same metric but computed on the training data. This would be insightful to evaluate the potential degradation of the GP emulator performance when used on inputs unseen during training.

Thank you for the suggestion to add the table and include the coefficient of determination ($R^2$). We have added the suggested table (copied below), that provides mean and 5th and 95th percentiles for each suggested statistic, to Section 4.2.2 "Model evaluation" and have referenced this table in Section 4.2.2 and Section 6.1. Since the GP interpolates the training data (c.f., neural nets), we fear that providing those statistics would be misleading, so we provide only statistics on the test data.

**Table 3.** Median values evaluated on the test set of the spatiotemporally averaged Gaussian Process emulator prediction RMSE, MAPE, bias and coefficient of determination $R^2$. Bracketed numbers indicate the 5th and 95th percentile values.

|  | RMSE | MAPE | Bias | $R^2$ |
|---|---|---|---|---|
| Overall | 0.046 | 0.020 | 0.00 | 0.97 |
|  | (0.021, 0.089) | (0.012, 0.036) | (-0.009, 0.008) | (0.90, 0.99) |
| Lower 30 km | 0.075 | 0.12 | -0.001 | 0.95 |
|  | (0.033, 0.14) | (0.070, 0.21) | (-0.020, 0.017) | (0.85, 0.98) |
| Upper 70 km | 0.025 | 0.010 | 0.00 | 0.92 |
|  | (0.011, 0.060) | (0.006, 0.020) | (-0.008, 0.006) | (0.78, 0.97) |
| Winter (DJF) | 0.010 | 0.006 | 0.00 | 0.998 |
|  | (0.005, 0.035) | (0.002, 0.023) | (-0.006, 0.007) | (0.98, 0.999) |
| Summer (JJA) | 0.079 | 0.045 | 0.00 | 0.92 |
|  | (0.038, 0.17) | (0.026, 0.076) | (-0.037, 0.026) | (0.77, 0.97) |

**Technical comments**

- General (1): The authors make an excessive use of parentheses throughout the text. In my view, parentheses should only be used to provide additional non-essential details in the text. I recommend that the authors clean up their parentheses by making more sentence separations instead of overloading single sentences.

  We have simplified the language throughout where it has been appropriate and there are now fewer parentheses than the original manuscript. In many cases, especially in Section 2.2 (Gaussian Process model), we retain parentheses to provide in-line definitions or examples of technical language to ensure that non-experts can follow the text.

- General (2): Throughout the manuscript, the authors use both the terms "inputs" and "parameters" to refer to the same notion: the parameters of GlaDS passed to the emulators. To avoid any confusion, a single term should be used consistently in the entire manuscript.

  Thank you for highlighting that these two terms were used interchangeably for the same object. As suggested, we have changed most instances of "inputs" to "parameters" or "GlaDS parameters". We have retained a few instances of "inputs" since it has sometimes been useful to speak about "inputs to the emulator" more abstractly than by referring to the GlaDS parameter values, particularly in Section 2 where we provide a high-level overview of the GP emulator. We have acknowledged this by stating that "Following the vocabulary of Higdon et al. (2008) and Verjans et al. (2024), these GlaDS model parameters are called the inputs to the emulator". We have moved this statement into the beginning of Section 2 where we introduced the concept of a GP emulator and before the first instance of "inputs". Whenever "inputs" is now used, we have clarified that we mean "emulator inputs" or "inputs to the emulator" to indicate that we are talking about the GP methodology more abstractly.

L5: Replace "construct robust" by evaluate uncertainty in.
Done.

L6: uncertainty quantification.
If we understand the reviewer correctly, this suggestion is to replace "uncertainty" with "uncertainty quantification" in the sentence: "Here, we develop Gaussian Process (GP) emulators that make fast predictions **accompanied by uncertainty [quantification]** of subglacial drainage model outputs".
What we mean to say here is that predictions have associated uncertainty, so we have revised this sentence to read: "Here, we develop Gaussian Process (GP) emulators that make fast predictions **with associated uncertainty** of subglacial drainage model outputs".

L14: "of the water pressure variance": it is unclear if this refers to spatial variance, temporal variance, and/or variance across the samples of the parameter space.
Thank you for highlighting this point, we mean variance as parameters are changed. We have revised this statement to: "[...] 90% of the variance in modelled water pressure in response to parameter changes"

L15: I believe that the mention to observational data is misused here, as no observational data is integrated in this study.

We have included a reference to integrating observational data to highlight this as an extension of the methods that we describe in this work: "The GP emulator approach described here is well-suited to integrate observational data with models to make calibrated, credible predictions of subglacial drainage". As part of another reviewer's comments, we have added a paragraph to Section 6.6 Applications and considerations that outlines how this work could be extended to calibrate model parameters. We hope that this discussion that we have added helps to provide additional details and context for this suggestion of related future work.

L25: "well-established": I understand what the authors mean here. However, this wording is misleading, because although there is consensus about the existence of an influence of subglacial hydrology on ice flow, this influence remains highly uncertain.

This is a good suggestion, we have rephrased this statement to: "Most commonly, studies of subglacial hydrology are motivated by the influence on ice-flow velocities of glaciers …"

L26: Replace flow by sliding.

We are trying to be careful here to not explicitly imply hard-bed sliding, including also the possibility of effective pressure-dependent sediment deformation. We have changed the two instances of "basal flow" to "basal slip", a more common catch-all term to encompass both of these processes (e.g., Cuffey & Paterson, 2010; Zoet & Iverson).

L31-36: This sentence is too long, and I do not know what "which" (L36) refers to.

We agree and have removed ", which guides the selection of model physics" since it is not necessary to make our point. Each set of references is already directly associated with a specific contribution to our understanding of subglacial hydrology.

L36: Replace large by high-dimensional.

Done.

L52: Typo: "GP emulators we develop".

We have corrected this to "The Gaussian Process emulators we develop take subglacial drainage model parameters as their inputs"

L53: I suggest this definition for flotation fraction: ratio of water pressure to ice-overburden pressure.

Thank you, we will take this definition as it is more clear.

L55: Please explain here the meaning of "global sensitivity indices".

We have changed "to" to "that" in the following sentence to provide a definition of the global sensitivity indices: "we compute variance-based global sensitivity indices  **that** precisely determine the combinations of parameters that most strongly control modelled subglacial hydrology"

L63: If possible, please use another word than "emerging".

This has been changed to "[...], with the channel  **radius determined by** the balance between creep closure of ice and opening by melt"

L69: Please provide units of variables.
We have added units for hydraulic potential (Pa), sheet thickness (m) and channel area (m$^2$)

Eq. (1): Although obvious, please define g.
Added to the description of Eq. (1), g is gravitational acceleration (Table 2).

L77: " each of these quantities defines a two-dimensional, time-varying field": this is confusing because I believe that both zb and pi are not time-varying.
Thanks for pointing this out, we have clarified that "$\phi$, N and fw are two-dimensional, time-varying fields", since zb and pi are not time-varying.

L83: Please specify: For details about GPs
Following another reviewer's suggestion, we have integrated the content from Appendix B into the main text. We have revised this reference to more precisely point to the appropriate material: "For background on Gaussian Processes see Jones et al. (1998) and Rasmussen and Williams (2005), and see Higdon et al. (2008) for a complete description of the emulators constructed here."

L85: I do not understand what is meant by: "in terms of the proportion of output variance corresponding: to each GlaDS parameter". Please clarify.
What we mean to say here is that we used the emulators to determine how each parameter influences the GlaDS outputs. We have clarified this statement to: "Following tuning and evaluation of the emulator, we apply the emulator to quantify the relationship between GlaDS parameters and GlaDS output." We provide a full technical definition, now including defining equations, in Section 5.

L91: Rephrase: Let $\mathbf{y}_i$ denote the vectorized model output of all variables (...).
If we understand the reviewer correctly, it appears that this statement was suggesting that we were concatenating the variables into a joint vector. Here, we intend $\mathbf{y}_i$ to be a placeholder for a single variable (e.g., flotation fraction). We have clarified this in the text by letting "$\mathbf{y}_i$ denote the vectorized model output of interest".

L94: Typo: "which are not a part of".
It appears the reviewer is pointing out the typo that it is incorrect to refer to $\mathbf{y}_i$ as the prediction for new parameter values since $\mathbf{y}_i$ is previously defined as the output corresponding to parameters $\mathbf{x}_i$. We will avoid overloading $\mathbf{y}_i$ by writing that "The emulation task is to predict the simulation output for new input values which are not a part of the design matrix $\mathbf{X}$". Elsewhere, we use $\mathbf{y}_p$ to represent the emulator prediction.

L104 Add a comma after $\boldsymbol{\theta}$.
If we are interpreting this suggestion correctly, it was unclear whether the hyperparameters were associated with the covariance function or with the mean function. We have reversed the ordering of the mean function and covariance function in this sentence to avoid this potential source of ambiguity:

"The GP is completely specified by the mean function μ(x) and the covariance function k(x$_i$, x$_j$; θ) with hyperparameters θ".

L110: Refer to $\mu(x)$ after "mean function".
Thank you for catching this, we will ensure to refer to variable symbols after their names throughout the text.

L110: "to set the mean to zero" should be to set the prior mean to zero. In the following sentences, it is also important to emphasize that it is only the prior mean that is zero.
Thank you for suggesting this technical correction, we have changed the quoted statement to: "to set the prior mean to zero", with corresponding changes throughout this section.

Eq. (5): This should be $y_p$ |**Y**, **θ**, $x_p$
Thank you for catching this mistake, we have corrected this as indicated

L124: I recommend being more specific here: (...) contains the pair-wise covariances between $x_p$ and each: entry of $x$ (...).
This is a good suggestion to clarify that $k_p$ is also a pairwise covariance vector. However, we would like to emphasize that the covariance function $k$ represents covariance between model outputs, not covariance between model parameters. We have now clarified that "the vector k$_p$ = k(x$_p$, x) contains the pairwise covariance between the model outputs from the simulation ensemble and the estimated output for parameters x$_p$".

L126: Specify: The prediction mean
Done.

L127: Refer to Eq. (4) after "covariance function".
Good suggestion, done.

L136 and L137: Replace "will" by would.
Done.

L138: Typo: "a variety solutions".
Corrected: "a variety of solutions".

L148: Please specify here that Eq. (8) assumes uncorrelated errors. This may not be entirely valid in this case.
The reviewer is correct, this has been revised to acknowledge that "This error model assumes that errors at each spatial position and timestep are uncorrelated, which might not be strictly true for our application."

L152: "can be viewed as": this wording is inappropriate, because it is a dimension reduction by definition.
We have added a more direct statement about the PC decomposition to the beginning of this paragraph: "In order to reduce the dimensionality of the simulation outputs, which leads to the obstacles described

above, the multivariate output field is modelled directly by using a principal component (PC) decomposition.". We were trying to be cautious not to lead the readers to think that a PC decomposition is the only way to address the size of model outputs.

L162: The authors can also invoke the orthogonality property of the PC decomposition to motivate their univariate approach.
Good suggestion, we have added the explanation: "Following Higdon et al. (2008), since the PCs are orthogonal, independent univariate GPs are fit to model the relationship between the inputs…"

L169: Replace "permissive" by flexible.
We prefer "permissive" since, in our view, one reason to use a covariance function with a weaker smoothness constraint is to permit a larger class of random functions. "Permissive" reflects this effect on the space of random functions.

L169: "variations in the principal components that tend to be smooth with respect to the input parameters": why is that? I would expect a strong sensitivity of GlaDS to some of its parameters, even in the PC subspace. Could the authors please clarify this statement?
We would like to clarify that strong sensitivity does not imply non-smoothness. The PC value can have large variations as parameters are varied. As long as these variations are reasonably smooth, the squared-exponential kernel is a reasonable choice. For clarity, we have revised this statement to: "While the flotation fraction field need not be smooth in space and in time, the principal components $w_{ij}$ ($\theta$) tend to vary smoothly with respect to the GlaDS parameters since the the spatiotemporal complexity is captured by the principal component basis."

L173: How many GP realizations are sampled?
We draw 64 GP realizations to compute the mean and prediction quantiles for these comparisons. We have found the mean and quantiles to be reasonably converged with this number of samples. For the more detailed evaluation of the reference emulator, we use 512 GP realizations. We have added this detail to this line.

L179-181: I am not sure to agree here. As I understand it, each univariate GP is fitted to a single series of PC coefficient, regardless of the number p of PCs retained. The number of parameters scales linearly with p, being p(d + 1) + 1. However, the amount of data used for fitting also scales linearly with p, because increasing p by 1 implies that one more series of PC coefficients is used. As such, I do not see why "a simpler model with fewer PCs and therefore fewer hyperparameters to estimate is desirable as it will have less prediction variance (i.e., less tendency to overfit)". On the other hand, I believe that using an increasingly high number of PCs would imply increasingly many GPs fitted to low-variance component of the GlaDS output, which can be regarded as noisy features of GlaDS results rather than dominant components of the variability.
The reviewer is right in saying that we are expanding the amount of data used in training each time we increase the number of principal components (p). Since the individual GPs are independent, each receives a different orthogonal vector of PCs that are used to infer the hyperparameters for that particular GP. Adding another GP therefore means that we use an additional column of the PC matrix to fit another set of hyperparameters. As in Higdon et al. (2008), we find that GP prediction ability levels off beyond a

certain number of principal components *p* (Fig. 3, 4). We have updated this statement accordingly: "In practice, GP predictions can be less accurate for later principal components that explain a small fraction of the ensemble variance (e.g., Higdon et al., 2008). Since including GPs for these later PCs does not meaningfully improve predictions, we will select a model with a modest number of principal components that nonetheless has similar performance obtained by using more components.

Section 3.1: In general, I think that more details are needed in this Section.
We have added details throughout this section as detailed in the following responses.

L192: Add one sentence to explain what the K-transect is.
We have described the K-transect as part of the western Greenland Ice Sheet.

L192-193: "adjusted from 0 m" and "increased to 40–1560 m": are these adjectives with respect to the SHMIP configuration? If so, please specify.
We have clarified these adjustments to: "The synthetic geometry consists of a flat bed with an elevation of 350 m (adjusted from 0 m in the SHMIP experiment) and surface elevation between 390–1909 m (adjusted from 1–1520 m in the SHMIP experiment) to match the observed elevation range of this part of the ice sheet."

L196: Please put the basal melt rate imposed into a glaciological context. For example, how does it compare with basal melt rate estimates in Greenland, or with the SHMIP forcing?
We have added that "This basal melt rate is in line with modelled basal melt rates in western Greenland (e.g., 0.001–0.1 m w.e. a−1, Karlsson et al., 2021), but lacks the seasonality associated with basal sliding". We have omitted a comparison to the SHMIP basal melt forcing since comparing the prescribed melt rates to the Karlsson et al. (2021) model results is a more robust comparison.

L199: " following a moulin density that varies with elevation computed from a satellite-derived supraglacial drainage map": is it possible to provide the formulation of the moulin density as a function of elevation?
We have added that "moulin density is parameterized by a normal distribution with mean 1138 m and standard deviation 280 m".

L200: "within each sub-catchment": this is not explained.
We have added an explanation that surface catchments are defined by a Voronoi diagram: "Surface melt is accumulated within sub-catchments surrounding each moulin defined by a Voronoi diagram and instantaneously routed to the bed."

Figure 1 Is it possible to add the melt rate using the right y-axis in Fig. 1a? What does the color scheme represent in Fig. 1b? Is it possible to indicate the moulin locations in Fig. 1d?
We have updated Figure 1 to include:
- A right y-axis on (a) to indicate surface melt rate
- A colorbar for (b) (surface elevation, m asl.)
- Moulin positions on (b) and (d)

[Figure]

L203: Replace "posed" by configured.
We have kept "posed" as this is the language used by Werder et al. (2013) to originally describe the discretization of the model on a triangular unstructured mesh.

L204: The variable x is already used to denote the input to the GPs. Please do not use the same symbol for two different variables.
Good point, this reference to $x$ is unnecessary since we do not refer to $x$ elsewhere, so we have removed "$x$=0 km" from the description of boundary conditions.

L216: "Following the vocabulary of Higdon et al. (2008) and Verjans and Robel (2024)": this is not needed here.
Following our response to General comment (2), it has sometimes been helpful to use "emulator inputs" rather than "GlaDS parameters" when talking about GP emulators more abstractly. We have moved this statement ahead to the beginning of Section 2 to explain these different terms before they are used.

L222: Concerning the parameter ranges, the ranges provided in Table 2 are most likely not intuitive to a majority of readers. I recommend adding a column in Table 2 specifying the ranges of parameter values used in previous studies focused on uncertainty quantification from GlaDS parametric uncertainty (e.g., Brinkerhoff et al., 2021).
We fear such an addition would be misleading, rather than helpful, as only two of the parameters (channel conductivity $k_c$ and bed bump aspect ratio $r_b$) correspond one-to-one with Brinkerhoff et al. (2021), who varied fewer subglacial drainage model parameters and used different cavity-opening and sheet-flow parameterizations. We have therefore not made this change.

L223: "flotation fraction fw < −10": in principle, any fw < 0 is nonphysical because it implies pw < 0. Are all these simulations rejected from the training data? And/or is the GP constrained to predict pw > 0?
It is correct that any flotation fraction $f_w$<0 is nonphysical since the model does not include physics for open-channel flow. We do not reject simulations from the training data, we sample from the entire

hyper-rectangle described by Table 2. We do not constrain the GP predictions since the goal of this work is to produce an emulator that mimics the model of choice (GlaDS) as closely as possible. We have clarified this description in the text: "We sample from the entire region described by the bounds listed in Table 2 without filtering or discarding nonphysical training runs (c.f., Jantre et al., 2024). The ensemble still contains some instances of negative or extremely high flotation fraction, but these do not appear to negatively impact the principal component decomposition nor the emulator predictions."

L234: As I understand, the test data do not include any extrapolation beyond the parameter space used for training. Therefore, it should be mentioned here and in the Discussion that the extrapolation capability of the GP emulator has not been evaluated.
This is correct, we have not evaluated the parametric extrapolation capability. When modelling a deterministic process (e.g., computer model outputs) with scalar GPs, there is a clear notion of interpolation. While it is not as clear in our case using the truncated basis representation, we still view the GP as interpolating between the training simulations to make predictions for new inputs. With this interpretation, we explicitly want to avoid extrapolating outside the range of parameters used in the GlaDS ensemble. We have acknowledged this in Section 6.1 after discussing the performance of the GP on the test data:
"Emulator performance has not been assessed when extrapolating outside of the range of parameters used for training the model. For predictions far outside the training range, the zero-mean GP that we have used will revert to predicting the mean of the ensemble of simulations, likely producing significantly higher error than we have found on the test data. Predictions should therefore only be made within the parameter ranges used in the GlaDS simulation ensemble."

L242: Please change this sentence to "In addition to emulating the spatiotemporal flotation fraction, (...)".
Done.

L277: Remove "small".
Removed since this section acts as the Results and we want to avoid much interpretation here.

L279: "perhaps since the input space has been explored more thoroughly": I do not think this is the case. In my view, more PCs are needed simply because the rank of the output space increases. For example, if a single simulation is run, it is fully characterized by a single PC. As more simulations are included, the number of PCs required to fully characterize the outputs increases, and thus the number of PCs to characterize a given % of the output variance also increases.
I think that we may have the same ideas about the size of the output space and the number of PCs. By exploring the input space more (i.e., running more simulations, including further towards the edges and the corners of parameter space), we obtain more linearly independent simulations, increasing the rank of the output space. We have found that the number of PCs needed to obtain a certain RMSE or cumulative variance threshold does not exactly scale with the number of simulations in the ensemble. For example, by doubling the number of simulations from 256 to 512, we do not need to double the number of PCs to maintain a consistent variance threshold. We have therefore not made any changes to this description.

L284: "only the absolute value, not the sign, of the PC basis vectors should be interpreted": I disagree. Opposite signs indicate opposite phasing of variability. It would be more correct to say that the sign of any given PC basis vector is arbitrary, but only looking at the absolute value would be wrong.

That is correct, we have updated this statement as suggested: "Note that the sign of the PCs and basis vectors are arbitrary since inverting the sign of both the basis […]"

Figure 3: Specify if the lines show the mean or median of RMSE and MAPE taken across the test simulations.

Lines show the median RMSE and MAPE across the test set. This caption has been revised to "Median root mean square error (RMSE, a) and mean absolute percent error (MAPE, b) for GP emulator predictions across the 100 test inputs [...]"

Figure 4c,f: I think that the presentation of the 95% prediction intervals is both unclear and misleading. Firstly, I understand that the RMSE and MAPE boxplots show the errors averaged in both time and space. But for the 95% prediction intervals, do the boxplots show the entire population of 95% prediction intervals taken at each grid cell and each time step of each test simulation?

We have clarified that a–c "show the distribution of the RMSE, MAPE and the spatiotemporally averaged 95% prediction interval evaluated on the test ensemble"

Secondly, this Figure suggests that broad prediction intervals are a bad thing. However, the purpose of a prediction interval is to communicate about the uncertainty in the output. Thus, it is a good thing that prediction intervals are broader for cases with high RMSE (i.e., the simulations with low PC numbers in this Figure). This means that the true GlaDS value may still lie within the 95% prediction interval despite the larger error in the mean estimate. For this reason, I recommend to show the percentage of GlaDS values falling outside of the 95% prediction intervals in Figure 4c,f, rather than the 95% prediction intervals themselves. If the GPs are well-calibrated, this percentage should be 5%.

We agree that broad prediction intervals are good if the breadth is necessary to encompass the expected proportion of the simulated values. We have added Fig. B4 (copied below) that presents the proportion of emulator-predicted values that overlap the simulated values.

We agree with the reviewer that the most important point is accurately communicating uncertainty in the emulator predictions. In presenting Fig. 4 and interpreting its results, we have not interpreted higher prediction uncertainty to be necessarily good or bad. Considering that Fig. B4 (copied below) shows the emulator uncertainty intervals contain the expected proportion of simulated values (i.e., are well-calibrated), we use the width of the prediction intervals to illustrate how adding more training simulations narrows the spread of predictions. If the intervals are properly calibrated (as shown by Fig. C3), then the balance between prediction uncertainty and cost to run the ensembles can be an important choice.

[Figure]

Figure: Proportion of GP emulator-predicted flotation fraction values that overlap the simulated values within the 95% prediction interval across the test set. (a) For emulators constructed using different numbers of principal components. (b) For emulators using subsets of the training ensemble. Horizontal lines indicate the median and black dots indicate the mean across the test set. The right axis in (b) indicates the number of CPU-hours associated with running each subset of the GlaDS training ensemble.

Figure 4d,e,f: Please mention in the caption that the x-axis uses a logarithmic scale.
Added.

Figure 4 caption: "Black circles indicate the total integrated prediction uncertainty": I do not understand this.
Since GP uncertainty varies with the distance to training points, we wondered if using the median uncertainty at the test points would overestimate the true mean if the test points were unusually far from training points. We assessed this by a Monte Carlo integral of the 95% prediction spread across the space of 8 GlaDS parameters. It is encouraging that the black circles (Monte Carlo-integrated) are near the median, as this shows that the test points are representative in terms of their prediction uncertainty. We have added the following explanation to the text: "Since GP prediction uncertainty varies across the space of emulator inputs depending on the distance to training runs, we assess the overall prediction uncertainty by computing a Monte Carlo integral across the space of GlaDS parameters, indicated as black circles in Fig. 4c"

L320: I find that it is worth mentioning that the RMSE for the three GPs is very similar for the late-September melt event, and I suggest to provide a succinct explanation of why this is.
This is a good observation. We have added an acknowledgement of this in the text: "All models have a similar RMSE during the September melt event, with relatively little improvement obtained by including more PCs." We do not have a concise explanation of why this event appears to be hard for the emulator to capture. Late-season melt events have persistently been difficult for the emulator, including in more recent work that uses different melt forcing and realistic geometry (preprint: https://doi.org/10.31223/X5GQ68). Observations of ice-sheet surface velocity show a similar type of strong response to moderate surface melt rates at the end of the melt season (e.g., Andrews et al. (2014)

Extended Data Figure 1c) as the drainage system is shutting down, suggesting this is an especially sensitive time for the drainage system.

L321: Typo: "reduces by the height".
Corrected: "Using 8 PCs reduces the height of"

L335: Table C1 should be referenced here.
We have added this reference.

L337: Please explain why fw > 2 is considered unrealistic.
We mean to say that any f_w>1 other than short-duration, highly localized events is unrealistic. We were using f_w>2 as an example. To clarify, we have revised this to say "many of the GlaDS simulations produce unrealistically high water pressure exceeding overburden (fw $\gg$ 1) for long periods of time and over a large portion of the domain (e.g., Fig. 6a2 and Fig. 7a2, b2)".

L343: "suggesting the emulator has reasonably accounted for basis truncation error": please note that this also suggests that the GP can correctly estimate uncertainty due to interpolation towards unseen parameter values.
Correct, this has been changed to "interpolation and basis truncation error"

L345: Add comma after "spring".
Thank you for the suggestion, we have added the indicated comma and revised this sentence to: "In the higher-error simulation, however, the prediction intervals do not overlap the simulation outputs in the spring, when the mean prediction significantly overestimates flotation fraction".

L345: "the mean prediction significantly overestimates flotation fraction": this suggests that the GP tends to further amplify the unrealistic GlaDS output. Please mention this explicitly.
We have added: "In other words, the emulator has amplified the unrealistically high GlaDS flotation fraction in the case of the 95th-percentile RMSE test simulation."

Table 3: Specify: Single GlaDS simulation.
Added.

L353 and Figure 8: Please use coefficient of determination ($R^2$ ) as an evaluation metric, rather than the squared correlation coefficient ($r^2$ ).
We have updated the analysis to compute the coefficient of determination ($R^2$) instead of the squared correlation coefficient ($r^2$).

L350-359 and Figure 8: These comparisons are misleading, because the GP emulator has been trained to reproduce $f_w$. It is impossible to know what the performance of the GP would be if it had been trained to reproduce $\phi$ or $N$. The discussion here should be rephrased as an evaluation of the error introduced by the conversion from fw to $\phi$ and/or $N$, rather than "indicators of GP prediction performance" or "different prediction skill" (L355).

Our main claim is that converting from one measure of water pressure to another will change the performance metrics of emulator predictions, as shown by Fig. 8. Based on this evidence, we suggest to emulate and train using the variable of interest for the study at hand. By converting emulator and simulator outputs to different units, we find that the hydraulic potential is the weakest of the three indicators of prediction skill since this comparison yields the highest proportion of explained variance (99.3%) when identical simulations and predictions are being compared. We have revised the text to ensure the accuracy of our statements. We would like to emphasize that we do not make any claims about the behaviour of hypothetical emulators constructed to predict effective pressure or hydraulic potential directly.

Figure 9 column c: Same comment concerning the prediction intervals as for Figure 4c,f.
As with our discussion of Fig. 4, the results and discussion section corresponding to this figure use reductions in prediction uncertainty as an example of why one might want to run additional simulations even as RMSE and MAPE are nearly stationary as long as the uncertainty estimates are well-calibrated.

L368: "Based on RMSE, MAPE and bias": I am not sure that these performances can be compared so easily from Table 4. For example, RMSE and bias have different units for the three variables. MAPE could serve as a better comparison basis, but Ts has been log-transformed, and MAPE may not be representative of the error on Ts itself. In addition, as mentioned by the authors, the MAPE values depend on the degree of variability in each quantity and on the values themselves (e.g., the percentage error when log Ts is 0 tends to infinity). Finally, when considering the ranges of values provided in Table 4, it is not clear to me that "the channel discharge fraction emulator has the best performance".
This is a good point. As the later part of the reviewer's comment indicates, this comparison should only be based on MAPE since the units of these quantities are different. We have refined this statement to more precisely relate to percent error, and not include broad statements about performance for such different quantities:
"We obtain the lowest percent error for the channel discharge fraction emulator (MAPE 5.02%), with similar percent error for the sheet transit time and channel network length emulators (8.7%; Table 4)"

L389: "Sensitivity indices for the flotation field are defined as a variance-weighted sum of the sensitivity indices for each principal component": This one-sentence explanation is not clear to me. If possible, I recommend providing the mathematical formulation for the sensitivity indices. That is, which formula is used to compute the values shown in Figure 10? Furthermore, the difference between first-order and total sensitivity indices should be explained.
We have added equations defining the sensitivity indices and expanded the description of how we compute sensitivity indices for the flotation fraction field.

Figures 10 and 11: Some whiskers extend beyond the value of 1.0. While the value of 0 can be intuitively interpreted as no sensitivity, what do values >1.0 mean?
These are errors from using only a finite number of GP predictions in estimating the sensitivity indices. The Saltelli et al. (2010) estimators that we use are guaranteed to converge as the number of simulations tends to infinity. In practice, the bootstrap sampling finds subsets where predictions holding certain parameters fixed have a higher variance than predictions with all parameters varying. We have added a brief description of the error bars, which was missing in the earlier version

"Confidence intervals for the sensitivity estimates are computed by bootstrap resampling. Confidence intervals extending >1 are a result of numerical errors in the estimators, which are only guaranteed to converge in the limit of infinite simulation runs (Saltelli et al., 2010)"

L418: "prediction RMSE is < 20% of the ensemble standard deviation": across the 100 test simulations? Yes, we have clarified this detail: "Across the input space, prediction errors are small relative to the variations across the ensemble of simulations: prediction RMSE is <20% of the standard deviation of the 100-member test ensemble"

L422: "PC truncation RMSE on the test set for the reference model with 8 PCs is 0.034, while GP prediction RMSE is 0.054, suggesting the PC truncation error contributes more than half of the prediction error.": I think that this statement requires a more thorough justification, and I am also not sure that I agree with the authors about it. First, does the 0.034 value correspond to the case of 8 PCs for the curve 256 simulations in Figure 2a1? If so, please refer to Fig. 2a1 in the text. Second, Figure 4 shows that there seems to be a baseline RMSE of the GP predictions of about 0.05, which does not decrease when going from 7 to 11 PCs. On the other hand, the PC truncation error must decrease when going from 7 to 11 PCs. As such, this indicates that there is a balance between (1) using only the first few PCs that seem to be relatively easy to predict for the GP, and (2) including low-variance PCs that allow to reduce the truncation error, but that seem to be harder to predict for the GP. As a consequence of this balance, the baseline RMSE stagnates at 0.05. But saying that "Of the two error sources, PC truncation error is the larger contributor" is misleading. I believe that if more PCs had been included, PC truncation error would decrease, but GP error would increase. Thus, this conclusion seems to be due to the choice of truncating at 8 PCs, rather than an inherent attribute of the GP. At least, this is how I understand the results. I would welcome any thoughts from the authors about this.
The reviewer is correct, and we were not clear in the text that these statements are only about the p=8 reference emulator. We have identified that we are talking about the reference emulator and acknowledged that "the balance between basis truncation error and GP error depends on the number of principal components used".

L434: "$5 - 10 \times 10^4$ time steps": why such a range? I thought that all simulations had been performed over the same time period and with the same temporal discretization.
This was unclear: we were adding up the time steps across the whole ensemble, i.e. up to 512 x 365 time steps. We have clarified: "This ensemble contain a large volume of data: the ensemble consists of up to 512 simulations, each with 365 time steps and ~4000 nodes"

L446: "The impact of large errors in predicting the spring pressure maximum is also reduced for ice-flow modelling applications": I disagree with this statement. For example, Fig. 4d (magenta curve) and Fig. S4c of Verjans and Robel (2024) show that the highest ice flow velocity errors due to the subglacial hydrology emulation occurs in the spring pressure maximum (their Fig. 4d) and at the ice velocity peaks (their Fig. S4c), which generally coincide with the spring pressure maximum. So, it is impossible to verify this claim from the authors if they do not actually compare ice flow model realizations forced with the GlaDS versus the GP output.
Thank you for pointing out that our original statement, definitively saying that errors associated with the spring pressure maximum will be reduced for ice-flow applications, was too strong. We have changed this

to suggest that errors "may be reduced" and clarified that "The impact of large errors in predicting the spring pressure maximum may be reduced for ice-flow modelling applications if the difference is only in the amplitude and not the duration of the pressure maximum.", with the explicit acknowledgement that "errors in the duration of water pressure exceeding the prescribed cap would propagate through the ice-sheet model to produce discrepancy to some extent in modelled velocity fields relative to using GlaDS directly".

L454-455: " the model of Verjans and Robel (2024), who report squared correlations ($r^2$)": this is not correct. Verjans and Robel (2024) report the coefficient of determination ($R^2$), which is not the same metric as $r^2$.
Thank you for correcting us, we have updated this to the coefficient of determination ($R^2$).

L474: "since PC truncation typically preferentially dampens high-frequency variations": I disagree with the authors here. PC truncation selects the components of variability with maximum variance. If most of the variance lies in high-frequency bands of the spectrum, there would not be any damping of high-frequency variations. Whether PC truncations dampens high-frequency variability or not depends on the power spectrum of the data.
The reviewer is correct. This statement was coming from our observations in previous work that using higher-frequency melt forcing results in larger differences between simulations with different parameter values relative to lower-frequency melt forcing (Hill et al., 2024). Based on this, we expect that using lower-frequency melt forcing would lead to a more accurate truncated basis representation in terms of RMSE. We have changed this statement to: "Smoother, averaged melt inputs (e.g., monthly, Table 5) would likely lead to reduced PC truncation error and therefore more accurate GP predictions since GlaDS simulations tend to have smaller variations in time and between simulations with lower-frequency melt inputs (Hill et al., 2024)"

L477-479 It is also important for resolving sub-annual ice flow variability.
That is correct, but since we do not include an ice flow component to this study, this paragraph focuses on implications for subglacial drainage models only.

L487: I believe that it would be relevant to add a sentence here about the propensity of GlaDS to produce nonphysical output.
This is another compelling reason that physics-based machine learning for GlaDS outputs has uncertain benefits. We have added this point to this section: "Considering the discontinuous nature of the channelized drainage system and the tendency of GlaDS to produce unrealistically high water pressure, it remains an open problem to apply physical constraints, such as mass conservation, to the subglacial drainage model emulation task and determine the applications which would benefit from such constraints."

L493: Citing Verjans and Robel (2024) here is misleading, because their emulator is transferable to different domains or melt inputs, i.e., the opposite of the sentence given here.
That is correct, this citation should be (c.f., Verjans and Robel, 2024) to highlight that we are contrasting these approaches. As part of answering another reviewer's comments, we have added a more nuanced

discussion to Section 6.4 of the applications that each of the three emulation studies (this study; Brinkerhoff et al., 2021; Verjans and Robel, 2024) are best-suited for.

L521-528: I found this entire paragraph a little vague and hand-waving. I recommend that the authors focus on the current work and future developments. For example, why discussing emulator predictions of global mean sea-level rise? This is clearly not the focus of this study.
This paragraph was intended to convey some of the decision points reached when designing an emulator, using applications taken from a wider swath of glaciology. We have removed the reference to sea-level rise contributions since that was distracting. As part of answering another reviewer's comments, we have added a paragraph discussing specific applications that are well-suited for the GP emulator approach that we have presented. Hopefully this paragraph will help address the vagueness highlighted by the reviewer.

L542: Typo: "is is".
Corrected.

L548-549: "fully Bayesian time-dependent calibration to provide observationally constrained distributions of subglacial drainage variables": I do not understand what the authors mean here.
We have clarified that we mean "Bayesian calibration with spatiotemporally resolved data to infer constrained distributions of subglacial drainage variables."

L561-562: Add regimes after "laminar and turbulent".
Done.

Eq. A3: To make the notation less clumsy, I suggest replacing the third and fourth terms on the left-hand side by $\partial h/\partial t$ .
Since both forms are technically correct, we have intentionally duplicated the terms to provide the most explicit representation of the governing equations.

L603: Please specify: of a multivariate Normal distribution.
Added.

Eq. B3: For this equation to be valid, there needs to be an additional constant term on the right-hand-side.
Thank you for highlighting that we were slightly abusing notation here. As part of reviewer 3's comments, we have brought the appendix B content into the main text and we no longer include this log-likelihood equation. In the corresponding likelihood equation in the main text (now Eq. (11)), we have provided the complete likelihood including all constants to provide a direct visual analogy to the univariate GP likelihood, Eq. (9).

L616: Typo: "includes" should be include.
Corrected.

L625: Typo: "by condition" should be by conditioning.
Corrected.

Eq. B5: θ should be boldfaced.
Corrected.

L629: then sampling from equation (B5)
Thank you for checking our equation references. Equation (5), the GP posterior predictive distribution, is the correct reference here.

L634: a major benefit of using
Corrected.

L635: Can the authors please remind here what are p and d so that the reader does not need to go back to the main text?
This is a good suggestion, and we should also not be using $p$ (the number of PCs) here since this is the extension of the univariate GP section. We have changed this to $d+1$ "where d=8 is the number of GlaDS parameters".

Figure C2: It seems to me that some of the Markov Chains are not well-mixed, although it is hard to tell from the scale of the y-axes. Did the authors compute convergence diagnostics? I recommend providing R-hat values and effective sample sizes (Gelman et al., 2013).
Thank you for the good suggestion. We have now computed the R-hat values and effective sample size $N_{eff}$ according to Gelman et al. (2013, p. 284–287). The median R-hat (1.065) and median effective sample size $N_{eff}$ (58 using 4 chains) indicate reasonable convergence. However, chains for individual parameters have not converged with R-hat as high as 1.3 (corresponding to $h_b$ for PC2) and $N_{eff}$ as low as 13 (corresponding to $k_c$ for PC8). Based on these diagnostics, we will:
1. re-investigate the step sizes used in the MCMC to ensure that the acceptance rates are in an appropriate range (~20–50%),
2. extend the length of the MCMC chains to ensure that chains are reasonably converged (as measured by R-hat) and that they contain a sufficient number of independent samples (as measured by $N_{eff}$),
3. report the acceptance rate, R-hat and effective sample sizes in the revised manuscript.

Figure C3: These figures should be shown in two dimensions rather than 3 for better clarity.
Part of the intention of this figure was to visualize the parameter interactions, which is best presented in 3D perspective. We understand the ambiguity in 3D plots with a single perspective, but we hope the surfaces are clear enough since we have added the colour scale to indicate the value (i.e., height) of the scalar quantities.

**References**
Douglas Brinkerhoff, Andy Aschwanden, and Mark Fahnestock. Constraining subglacial processes from surface velocity observations using surrogate-based bayesian inference. *Journal of Glaciology*, 67(263):385–403, 2021.

Andrew Gelman, John B Carlin, Hal S Stern, David Dunson, Aki Vehtari, and Donald B Rubin. *Bayesian data analysis*. Chapman and Hall/CRC, 2013.

Eric Larour, Helene Seroussi, Mathieu Morlighem, and Eric Rignot. Continental scale, high order, high spatial resolution, ice sheet modeling using the ice sheet system model (ISSM). *Journal of Geophysical Research: Earth Surface*, 117(F1), 2012.

Vincent Verjans and Alexander Robel. Accelerating subglacial hydrology for ice sheet models with deep learning methods. *Geophysical Research Letters*, 51(2):e2023GL105281, 2024.

Mauro A Werder, Ian J Hewitt, Christian G Schoof, and Gwenn E Flowers. Modeling channelized and distributed subglacial drainage in two dimensions. *Journal of Geophysical Research: Earth Surface*, 118(4):2140–2158, 2013.

Andrews, L. C., Catania, G. A., Hoffman, M. J., Gulley, J. D., Lüthi, M. P., Ryser, C., ... & Neumann, T. A. (2014). Direct observations of evolving subglacial drainage beneath the Greenland Ice Sheet. *Nature*, 514(7520), 80-83. doi: 10.1038/nature13796

Ehrenfeucht, S., Dow, C., McArthur, K., Morlighem, M., & McCormack, F. S. (2025). Antarctic wide subglacial hydrology modeling. *Geophysical Research Letters*, 52(1), e2024GL111386. doi:10.1029/2024GL111386

Hill, T., Flowers, G. E., Hoffman, M. J., Bingham, D., & Werder, M. A. (2024). Improved representation of laminar and turbulent sheet flow in subglacial drainage models. *Journal of Glaciology*, 70, e24, doi: 10.1017/jog.2023.103.

Hoffman, M., & Price, S. (2014). Feedbacks between coupled subglacial hydrology and glacier dynamics. *Journal of Geophysical Research: Earth Surface*, 119(3), 414-436. doi: 10.1002/2013JF002943

Kazmierczak, E., Gregov, T., Coulon, V., & Pattyn, F. (2024). A fast and simplified subglacial hydrological model for the Antarctic Ice Sheet and outlet glaciers. *The Cryosphere*, 18(12), 5887-5911. doi: 10.5194/tc-18-5887-2024

Saltelli, A., Annoni, P., Azzini, I., Campolongo, F., Ratto, M., & Tarantola, S. (2010). Variance based sensitivity analysis of model output. Design and estimator for the total sensitivity index. *Computer physics communications*, 181(2), 259-270. doi: 10.1002/9780470725184

Sommers, A., Meyer, C., Morlighem, M., Rajaram, H., Poinar, K., Chu, W., & Mejia, J. (2023). Subglacial hydrology modeling predicts high winter water pressure and spatially variable transmissivity at Helheim Glacier, Greenland. *Journal of Glaciology*, 69(278), 1556–1568. doi:10.1017/jog.2023.39

Warburton, K. L. P., Meyer, C. R., & Sommers, A. N. (2024). Predicting the onset of subglacial drainage channels. *Journal of Geophysical Research: Earth Surface*, 129(12), e2024JF007758. doi: 10.1029/2024JF007758

Zoet, L. K., & Iverson, N. R. (2020). A slip law for glaciers on deformable beds. *Science*, 368(6486), 76-78. doi: [10.1126/science.aaz1183](10.1126/science.aaz1183)

---

## Author Comment (AC2)

**Author Response to RC2: "Computationally efficient subglacial drainage modelling using Gaussian Process emulators: GlaDS-GP v1.0"**

Reviewer: Jacob Downs

Tim Hill, Derek Bingham, Gwenn E. Flowers, Matthew J. Hoffman

Reviewer comments are in black and we provide our responses in blue.

This work outlines a Gaussian Process based emulator of the GlaDS subglacial hydrology model. Inputs to the emulator consist of 8 scalar model parameters, while outputs include either spatio-temporal flotation fraction fields or scalar metrics describing bulk properties of the subglacial drainage system. Performance of the emulator is thoroughly examined by comparing emulators using different training sets as well as using different numbers of principal components to represent flotation fraction fields.

Overall, this manuscript is well written, the methods are clear, and I appreciate the rigorous evaluation of the emulator's accuracy. In light of this, I believe this work represents a good contribution to the field. However, I believe that this work would benefit from a more detailed discussion of how the GP emulator compares to existing emulators. In particular, I think the discussion should outline more of the key differences between this work and Verjans and Robel [2024], including differences in their potential use cases.

The authors should outline in specific terms where their emulator could be applied versus other existing methods. For instance, is this purely a tool for calibrating subglacial hydrology parameters or assessing sensitivity? Is there a path for using this as a substitute for GlaDS to predict effective pressure when coupled with an ice sheet model? What do the authors intend to do with this emulator, or what might future work on the emulator entail? As someone interested in this space, I was hoping the authors might delve more deeply into some of these topics in the discussion.

Thank you for taking the time to review our work in detail and for the constructive suggestions. We have responded to your comments individually below.

Line 52 Might benefit from rewording to "The Gaussian Process emulators we develop take subglacial drainage model parameters as inputs and predict spatially and seasonally resolved flotation fraction ... " We have revised this wording as suggested to improve the clarity of this sentence.

Line 63 Maybe this could be reworded as "where the radius of channels is modeled as a balance between the creep closure of ice and opening by melt. " We have revised this to say, "[...], with the channel  **radius determined by** the balance between creep closure of ice and opening by melt".

Line 65 "The continuum distributed (sheet) drainage system is defined on the mesh nodes, with possible channel locations defined by element edges." This could be clarified. Maybe you could say that variables describing the distributed system are represented by linear finite elements with degrees of freedom on mesh nodes? This would help clarify that

the distributed drainage system isn't "confined" to the mesh nodes, but that's where the degrees of freedom are defined.

We have used the suggested, more accurate statement: "The governing equations, arising from conservation of mass and energy, are discretized on an unstructured triangular mesh. Variables describing the continuum distributed (sheet) drainage system are represented using finite elements with degrees of freedom located on the mesh nodes, with possible channel locations defined by element edges."

Line 83 "For details on Gaussian Processes"
Corrected, we have revised these citations to more precisely point the reader to the relevant references: "For background on Gaussian Processes see Jones et al. (1998) and Rasmussen and Williams (2005), and see Higdon et al. (2008) for a complete description of the emulators constructed here"

Line 130 Please mention why there are d+1 hyperparameters. Also, depending on the kernel, this number could vary.
Thank you for highlighting this. The reviewer is correct, while it is usually d+1 hyperparameters, the number can vary. Since we have not introduced a particular kernel function, it's not appropriate here to say how many hyperparameters there are yet. We have removed the number of hyperparameters from this statement: "The fact that the GP model is simple enough to allow for Bayesian inference of the emulator hyperparameter values $\theta$, where uncertainty in the hyperparameters is reflected in the uncertainty in the emulator predictions, is a key advantage compared to a neural network for uncertainty quantification"

Line 170 That seems reasonable. If I understand, would it be fair to say that you hope that much of the spatial/temporal complexity is captured in the principal components while the response of the coefficients of the principal components to variations in parameters is expected to be smooth?
Yes, this is precisely what we are trying to say, and we have revised this section to read: "While the flotation fraction field need not be smooth in space and in time, the principal components $w_{ij}$ ($\theta$) tend to vary smoothly with respect to the GlaDS parameters since the the spatiotemporal complexity is captured by the principal component basis."

Line 285 Interesting, I like the commentary on the first couple principal components, and I think interpretability is a really nice advantage of this method.
Thank you for the nice comment, and we agree that the interpretability of this method is a nice feature.

Line 321 "Using 8 PCs reduces the height..."
Corrected.

Figure 4 I'm not sure what is meant by 95% prediction interval in c and f. Is this the prediction uncertainty of the emulator integrated through time and space? Also, when considering the prediction error, is this accounting for uncertainty in $\theta$ (sampled via MCMC) as well as the GP prediction uncertainty? Or is it just the latter, and you are using the most probable estimate for the hyperparameters? Presumably the advantage of using MCMC on $\theta$ as opposed to maximum likelihood estimation is to characterize its effect on uncertainty as well?
Fig. 4c and 4f show the prediction interval averaged through time and space. We have added an explanation of this figure to the text: "Fig.4a--c show the distribution of the RMSE, MAPE and the

spatiotemporally averaged 95% prediction interval across the test ensemble for emulators constructed using different numbers of PCs. Since GP prediction uncertainty varies across the space of emulator inputs depending on the distance to training runs, we assess the overall prediction uncertainty by computing a Monte Carlo integral across the space of GlaDS parameters, indicated as black circles in Fig. 4c, f).”

The 95% intervals include (1) uncertainty in the hyperparameters, (2) GP prediction uncertainty, and (3) uncertainty arising from the truncated basis via the error term and $\lambda_{\text{sim}}$. We have added a Figure providing an overview of the methods, including a summary of the algorithm used to draw posterior realizations from the GP. We hope this figure helps to clarify the uncertainties included in the emulator predictions.

Line 340 Before discussing results for the test cases with different levels of error, please introduce what the "high-error" and "median-error" simulations are. I think I found this information in the figure 6 and 7 captions, but it would be helpful to include it in the text.

Good suggestion, we have explained these test cases in the text: "To assess how emulator performance varies across the test set, we evaluate the performance on test simulations with 95th-percentile ("high error"), 50th-percentile ("median error") and 5th-percentile ("low error") RMSE."

Figure 9 It's difficult to see the differences in the median error across different numbers of simulations due to the scale of the outliers.

This is true, especially for the log-sheet transit time (row 2). We would also like to point out that the difficulty in seeing changes in the mean is a useful interpretation of the experiment. For these scalar metrics, it requires only ~32 simulations to obtain predictions with a median error (bias) near zero. Adding more simulations primarily reduces the error in the worst-case scenarios (outliers and the spread of the whiskers) and the uncertainty in predictions.

Section 6.4 I think some of this discussion could be expanded to highlight more of the nuances of the different approaches. For example, the emulator in Verjans and Robel [2024] aims to be fairly general purpose, and it's more of a one-to-one substitute for GLaDS. Hence, I see their neural network based approach as fundamentally different from yours in its intent. This also makes the comparison of the number of parameters difficult as the input / output spaces in Verjans and Robel [2024] is very broad, meaning more parameters are likely needed.

There are certainly a number of appealing elements to your GP approach, including interpretability, the speed at which it can be trained, and how you also obtain uncertainty estimates without additional work. But I feel like its difficult to directly compare your emulator to Verjans and Robel [2024], and I have a hard time seeing the two emulators being used in the same way. In this sense, I see your emulator as being far more comparable to Brinkerhoff et al. [2021].

Thank you for the suggestion, and we agree with your assessment that our approach is most similar in spirit and applications to Brinkerhoff et al. (2021). We have added a discussion about the utility of these three studies to Section 6.4:

> The GP emulator approach that we have described is closest in spirit and in practical applications to that of Brinkerhoff et al. (2021). By emulating model outputs for different model parameter values, the GP emulator constructed in this study and the Brinkerhoff et al. (2021) neural network emulator are well-suited for quantifying parametric uncertainty, calibrating model parameters

given data and exploring parameter sensitivity (e.g., Fig. 11). Both approaches use a principal component decomposition that nicely introduces interpretability for the emulator (e.g., Fig. 3, 11). Aside from structural differences in the type of emulator, the major differences between our work and that of Brinkerhoff et al. (2021) is that we explicitly resolve subglacial water pressure and drainage characteristics and we obtain a built-in prediction uncertainty estimate, whereas Brinkerhoff et al. (2021) implicitly represent subglacial conditions through the influence on surface velocities and take extra steps to estimate prediction uncertainty. Both approaches are tied to a particular study area, limiting their utility for large-scale forward modelling. On the other hand, Verjans and Robel (2024) use a convolutional neural network that can generalize to arbitrary melt forcing and study areas, making it an ideal tool for forward modelling of ice-sheet evolution forced with a basal boundary condition that is influenced by the hydrology emulator. Since Verjans and Robel (2024) do not predict water pressure for different model parameters, their emulator is not ideally suited for uncertainty quantification, calibration of drainage model parameters or sensitivity analysis.

Line 481 To me, saying that IGM enforces conservation of momentum sounds as if it is enforced strictly. Although IGM uses a physics-based loss function, conservation of momentum is only upheld approximately. Contrast this with something like Horie and Mitsume [2024], in which a value is strictly conserved in a neural network.

This is a good point, we have updated this wording to clarify that "Jouvet et al. (2023) use a loss function that is based on conservation of momentum as part of a neural network ice-flow velocity emulator", rather than strictly enforcing their PDE constraint.

Section 6.6
I think this manuscript would significantly benefit from elaborating on specific use cases for this emulator. For instance, do you see the emulator being used more or less as is, or do you think the value of this work is in the general approach that you present? You mention uncertainty in future sea level rise, but not a clear application of the emulator for this purpose.

There is a pretty clear use case for using the emulator to do Bayesian calibration of subglacial hydrology model parameters, but what other use cases might it have? Do you see a path forward for coupling effective pressure fields from the emulator to an ice sheet model? Clarifying the intended use of this emulator or more concrete pathways for other applications would really strengthen the discussion.

Thank you for the suggestion to improve this discussion. We have added two paragraphs to Section 6.6 to more clearly lay out where we see this methodology being used:
- Calibrating parameters of the subglacial drainage model using observations of quantities corresponding to GlaDS outputs, such as borehole water pressure or moulin water level, tracer transit times, or channel characteristics inferred from passive seismic measurements.
- Using emulated effective-pressure fields as inputs to a sliding law to characterize the sensitivity and related uncertainty of ice flow (e.g., solid-ice discharge to the oceans) to drainage model parameters.

- More broadly, the GP methods have not been extensively tailored to the subglacial drainage application, they have a place as part of uncertainty quantification across a range of glaciological processes.

For the reviewer's information, we would like to highlight that we have a recent preprint that applies the GP approach described in this manuscript to the calibration task: https://doi.org/10.31223/X5GQ68. We have added a reference to this preprint when we point to Bayesian calibration of drainage model parameters as an extension of the present work.

**References**

Douglas Brinkerhoff, Andy Aschwanden, and Mark Fahnestock. Constraining subglacial processes from surface velocity observations using surrogate-based bayesian inference. 67 (263):385–403, 2021. ISSN 0022-1430, 1727-5652. doi: 10.1017/jog.2020.112. URL https://www.cambridge.org/core/journals/journal-of-glaciology/article/constraining-subglacial-processes-from-surface-velocity-observations-using-surrogatebased-bayesian-inference/1E03CA805D8CE6A0C310108540D9457E. Publisher: Cambridge University Press.

Masanobu Horie and Naoto Mitsume. Graph neural PDE solvers with conservation and similarity-equivariance. 2024. URL https://openreview.net/forum?id=WajJf47TUi.

Vincent Verjans and Alexander Robel. Accelerating subglacial hydrology for ice sheet models with deep learning methods. 51(2):e2023GL105281, 2024. ISSN 1944-8007. doi: 10.1029/2023GL105281. URL https://onlinelibrary.wiley.com/doi/abs/10.1029/2023GL105281. eprint: https://onlinelibrary.wiley.com/doi/pdf/10.1029/2023GL105281.

---

## Author Comment (AC3)

**Author Response to RC3: "Computationally efficient subglacial drainage modelling using Gaussian Process emulators: GlaDS-GP v1.0"**

Tim Hill, Derek Bingham, Gwenn E. Flowers, Matthew J. Hoffman

Reviewer comments are in black and we provide our responses in blue.

This paper describes a Gaussian Process emulator of the GlaDS subglacial drainage model and its testing on a synthetic ice-sheet margin setup. Modelling subglacial drainage is starting to become an important aspect of ice dynamics simulations as that system impacts the basal boundary condition significantly. However, subglacial drainage models are relatively costly to evaluate and in particular operate on different, shorter time scales compared to ice flow. Thus running coupled ice-flow drainage simulations is typically difficult and costly at the moment. Emulating the subglacial drainage model using a statistical representation is likely an important step in making these types of coupled models readily applicable.

Whilst emulations of GlaDS with neural network based emulators have been achieved over the last few years, this is the first time a Gaussian Process based emulator has been put forward. The advantage of GP emulators is their greatly reduced number of parameters to fit compared to a neural network as well as built-in capability to quantify uncertainties of the emulation.

The manuscript lays out the procedure to construct the GP emulator; of note is that this construction is relatively involved as it also entails, for instance, decomposition of the GlaDS training data into principal components, fitting of hyperparameters using Bayesian schemes, etc. The emulator is then tested extensively on a synthetic setup and the authors discuss the pros and cons relative to neural network based emulators.

The study and manuscript are carefully constructed. As I am not an expert in statistical emulators, I cannot judge the appropriateness and correctness of the approach to implement the GP emulator. The testing and assessment of the emulator is certainly fine and the discussion is interesting and relevant. Thus, with above caveat, I recommend to publish this manuscript in GMD with the minor corrections outlined below.

Thank you for the detailed and constructive review. We have responded to your comments individually below.

Comments

I think it would be useful to discuss a bit more how this emulator could be used for inversions or for coupled ice-flow & drainage simulations as, in my opinion, this are the most sought after usages of such tools. This can just be in the Discussion and/or Introduction, no need for more simulations or an implementation.

Thank you for the suggestion. We have expanded Section 6.6 "Applications and considerations" to describe Bayesian inference of subglacial drainage model parameters as an appealing direct extension of this work. We have described the steps needed to use the emulator for coupled ice-flow and subglacial

drainage simulations and highlight some of the additional uncertainties related to the basal slip relationship and the ice flow law that could be addressed by extending the current work.

The construction of the emulator has many steps. Looking through the manuscript, I can see:

- training data construction using parameter design matrix

- running the simulations with GlaDS

- principal component decomposition and component selection or (reduction of variables to scalars)

- fit the GP emulator to the data using an MCMC scheme

Then using the GP in different ways for predictions and analysis is then yet another step. Would it make sense to somehow graphically represent this, flow-chart or some such? Or maybe a numbered list?

Thank you for suggesting ways to make the construction of the GPs more accessible. We have designed and added the following summary of the steps involved in the emulator construction.

[Figure]

Figure: Overview of steps involved in constructing the Gaussian Process emulators. X is the design matrix of GlaDS parameters (defined in Table 2) with corresponding GlaDS outputs Y. The Gaussian Process emulator is constructed as a truncated linear combination of p principal components $w_i(\theta)$ and basis vectors $v_j$ for i = 1, . . . , p, where $\theta$ are Gaussian Process hyperparameters that are inferred by Markov Chain Monte Carlo (MCMC) sampling. Emulators are fit using *m*-member subsets of the training data and constructed using different numbers of principal components *p*. The performance is evaluated on the independent set of 100 test simulations. The emulator is used to compute the sensitivity of model outputs to model parameters (Section 5).

Irrespective of the lack of such a graphical overview, I struggled to understand the GP emulator from the description. I am not sure whether I should expect to understand GP emulation from reading about it in such a publication or whether I should just need to go elsewhere to learn it. I see that the authors try to keep the reading smooth by moving quite a bit of the explanations to the appendix but I wonder whether that makes it even harder to understand as now the content is disjoint? Maybe if this layout is kept, then make it even more high level in the main text and have the full description in the appendix which then could be in one place; or, alternatively, move all into the main text? In fact, I think that would be my preferred option and, I think, would fit GMD well as this journal is mostly about methods and not science. As it is, I think it is a bit of a difficult split.

Thank you for highlighting that Section 2.2 was not as accessible to non-experts as we had intended and for suggesting improvement in the content and structure. We have expanded Section 2.2 to integrate the content from Appendix B so that the reader has all the information in one place. We have also expanded the high-level description of the terms in the equations and defined all statistical terminology. At the beginning of the GP section, we have also clarified our intention to provide a high-level overview with only the details necessary to understand our application of the method and the differences compared to other statistical models (e.g., neural networks):

> "This section briefly provides a high-level overview of the Gaussian Process (GP) model and the architecture that we use to emulate spatially and temporally resolved GlaDS outputs. For background on Gaussian Processes seeJones et al. (1998) and Rasmussen and Williams (2005), and see Higdon et al. (2008) for a complete description of the emulators constructed here."

The authors state the principal component decomposition will make the representation necessarily smooth (line 170). Around the channels the hydraulic potential is often not smooth but has the channel as a kink, is that a problem (i.e. a spatial non-smoothness)? Also related to smoothness: in setups like the one presented, where there is no lateral variation in topography, channel position is not necessarily stable with parameter variation but they can jump around (and, for certain, channels move if the mesh is varied). Is that a problem for GP?

The perturbation in hydraulic potential (or more precisely flotation fraction for our work) near a channel is not a problem for the principal component-based GP. The spatial and temporal variations themselves do not need to be smooth since this complexity is encoded by the basis, which has no smoothness constraints. What the GP requires is that the principal components ($w_{ij}$ in Eq. (9)) vary smoothly with respect to the GlaDS parameters. We have tried to explain this more clearly: "While the flotation fraction field need not be smooth in space and in time, the principal components $w_{ij}$ ($\theta$) tend to vary smoothly with

respect to the GlaDS parameters since the the spatiotemporal complexity is captured by the principal component basis."

The comment about unstable channel positions is interesting. It's possible that "boundaries" in parameter space that cause changes in channel position would be reflected as discontinuities in the principal components. This would show up as simulations with unusually high error when evaluating predictions on the test set. We have not seen evidence of such issues.

Line-by-line

L4: "the combination of the number" is not clearly formulated. Reword.
We have revised this sentence to read "While they are used to understand processes such as the relationship between surface melt and ice flow, the number of uncertain model parameters and the computational cost of running models makes it difficult to [..]"

L8: "daily representation" is not clear to me. Maybe "diurnally averaged"?
Thank you for the suggestion, we have updated the text as suggested since "diurnally averaged" is more accurate.

L66: I would cite the ISSM GlaDS implementation here too, I think that is Ehrenfeucht&al 2023.
Correct, we have added this citation.

L83: "see B" -> "see Appendix B"
Corrected.

L84: "fast predictions" is a bit sloppy, they are fast to run but not fast themselves.
That is correct, this sentence has been simplified to say: "Following tuning and evaluation of the emulators, we apply them to compute the sensitivity of model outputs to parameters."

Table 2: r_b is not defined in the original GlaDS paper nor in this manuscript. Needs to be defined, at least in Appendix A.
Thank you for the suggestion. We have added the definition of the aspect ratio $r_b$ as the ratio of the bump length $l_b$ to the bump height $h_b$, such that the aspect ratio should be roughly >1, following equation (A2).

L108: state here that theta is what is fitted and maybe also state the (approximate) size of theta.
We have expanded the description of the GP hyperparameters: "The hyperparameters typically control the variance of the Gaussian Process and the sensitivity to each input, but their interpretation depends on the type of covariance function that is used. Gaussian Processes typically have a similar number of hyperparameters as inputs to the emulator. The hyperparameters must be optimized to obtain an accurate emulator."

L110: "The second choice" really needs a clear statement above of what the first choice is (namely k), otherwise the reader will stumble over this.

Thank you for the suggestion, since we do not clearly articulate the covariance function as the first choice, we have removed the language about "the second choice". Instead, this paragraph begins with "We make the common choice to set the prior mean to zero everywhere…"

L124: $x$ is not defined, or if its definition is "prediction input", then that is not clear enough.
We have defined $x$ on line 88 as the vector of GlaDS model parameters. Since using both "model parameters" and "inputs" to refer to the same thing is confusing, we have referred to $x$ as model parameters throughout (see also response to reviewer 1).

L127: the "posterior distribution" comes out of the blue here
Thank you for highlighting this. As detailed in our response to your third comment (clarity of the GP exposition), we have revised this section to more fully explain the posterior distribution and posterior predictions, keeping in mind to explain these statistical terms.

L130: are there d+1 hyperparameters for any k? Couldn't it be less as well? Or more?
The fact that there are d+1 hyperparameters is specific to how we have written the covariance function $k$. Since we have not yet introduced a particular covariance function $k$ it is not appropriate yet to provide a specific length of the hyperparameter vector. We have revised this sentence to remove the precise number of hyperparameters: "the fact that the GP model is simple enough to allow for Bayesian inference is a key advantage compared to a neural network for uncertainty quantification."

L223: A negative floation fraction implies negative water pressure, right? But how can the water pressure go negative in the presented setting? I don't think it can drop below the value of the Dirichlet BC which corresponds to zero water pressure.
This is a good question, and the reviewer is correct that negative flotation fraction implies negative water pressure. The negative flotation fraction (and water pressure) happens during the melt season in response to a rapid drop in surface melt rates and only lasts for one to a few days. We have clarified that this is a transient issue: "We have found that broadening the parameter ranges results in numerous nonphysical simulations with nearly zero water pressure during the melt season, transient negative flotation fraction as low as $f_w < -10$ or extremely high flotation fraction as high as $f_w \gg 100$ which degrade the performance of the principal component decomposition."

L274: RMSE is not defined yet. But then it gets defined in L296.
Thank you for highlighting, we have defined root mean square error at the first instance of RMSE.

Fig 2: state to which fields the PCs are encoding
The principal components (a) and basis vectors (b) represent the spatiotemporal flotation fraction field. We have added "flotation fraction" to the caption related to (b): "Width-averaged representation of the first seven **flotation fraction** $f_w$ spatiotemporal principal component basis vectors [...]"

L283: It would be nice to have some snapshots of the PC fields and the GlaDS fields side by side (probably in the appendix). So similar to Fig 2 panel b, but not width averaged but instead just a few instances in time. This would allow to get a bit of a feel on how accurate the spatial fidelity of the PCs are.

This is a good idea, it will provide some intuition of how the PCs and GP behave. We have added the following figure to Appendix C. We would also like to clarify that Fig. 2b does not show the PC low-rank representation of the GlaDS-simulated flotation fraction. Fig. 2b shows the principal component basis, $v_j$ in Eq. (9). We have referenced $v_j$ and Eq. (9) in the Fig. 2 caption to minimize confusion about the quantity shown in (b).

[Figure]

Figure: Principal component truncation error and GP prediction error. (a) GlaDS-simulated flotation fraction for the test simulation with median RMSE on 29 July, (b) corresponding principal component representation of the GlaDS flotation fraction using 8 PCs, and (c) Gaussian Process (GP) emulator prediction. Difference maps show the principal component representation (d) and the Gaussian Process emulator prediction (e) minus the GlaDS output.

Fig 4: Eyeballing the convergence of the two error metrics (panel a,b,d,e), it looks like that the errors do not go to zero but approach some non-zero asymptote. Is that expected? If so, why? Maybe this could be briefly mentioned in the text.
This is correct and expected. There are errors in the GP predictions from two main sources:
- Basis truncation error: using 1–11 PCs obtains only an approximation of the full set of simulations (e.g., Fig. 1a)
- GP prediction error adds to the basis truncation error. The GP is only be expected to be a perfect predictor of the principal component representation of the data in the theoretical limit of infinite training runs
We have added a brief explanation where we present the results from Fig. 4 (now numbered Fig. 5):
"Figure 5 seems to suggest that the RMSE and MAPE are converging to nonzero values. This is an expected outcome since the total error represents the sum of the basis truncation error associated with using at most 11 PCs (Fig. 3) and error in the GP predictions of the principal components"

L394: "supporting the interpretation of PC1 as representing water pressure in the absence of surface melt inputs": to me Fig2b1 shows that PC1 has a clear seasonal signal which the basal melt does not. So, I'm not sure this statement is correct or at least needs some more information.

Thank you for the question. While the first basis function (PC1 in Fig. 2b1) does indeed have a clear seasonal signal, Fig. 2b1 shows that PC1 "turns off" by being nearly 0 in the melt season and especially at lower elevations, so that PC1 does not contribute to the surface melt-forced drainage system. PC1 is consistently "turned on" with absolute values >1 during winter and above the maximum melt extent. From this, we argue that PC1 mostly represents subglacial drainage in the absence of surface melt. It turns off when surface melt dominates the drainage system, allowing other PCs to dominate at these times. We have expanded our explanation where we propose an explanation for PC1:

"Based on the first PC basis vector being nonzero in winter and upstream of the maximum surface melt extent (~80 km), **and not contributing to the solution at low elevations during the melt season,** the first and most important PC in terms of its explained variance (80.3%) appears to control the baseline water pressure in the absence of surface melt inputs"

L444: Formulating more clearly what "in ice-flow modelling" means would be helpful
We have added the explanation: "if the emulated fields were used as part of the basal boundary condition for ice-flow modelling".

Tab5: here the typesetting seems a bit off: in the fields spanning multiple lines, the line spacing should be less than between different rows.
We have had to adjust the formatting of Table 5 to force it to fit on a single page. We will ensure that the formatting is correct in the final typeset version of this table.

L552: ideally a DOI and stable archived version of SEPIA and ISSM should also be provided. At the very least the version of ISSM used needs to be stated.
We have used ISSM version 4.24, which is available as a release on GitHub (https://github.com/ISSMteam/ISSM/releases/tag/v4.24), but not with a DOI. We have added "v4.24" to the code and data availability statement. Since SEPIA and ISSM are not our code to archive, we have provided the best publicly accessible links that we can.

L554: the air-temp dataset needs to be clearly specified. The provided link points to very many datasets. Do note that this data-repository provides DOIs for each dataset.
The correct link to the Greenland weather station data is https://doi.org/10.22008/FK2/IW73UU and the text has been amended accordingly

Eq A2: I would expect r_b to feature here.
Thank you for highlighting this mistake, this equation (and the following equation for time-evolution of hydraulic potential) has been corrected to include the bed bump aspect ratio r_b instead of the bed bump length l_b:

$$\frac{\partial h_{\mathrm{s}}}{\partial t} = f_{\mathrm{b}} \frac{h_{\mathrm{b}} - h_{\mathrm{s}}}{h_{\mathrm{b}} r_{\mathrm{b}}} u_{\mathrm{b}} - \tilde{A} h_{\mathrm{s}} |N|^{n-1} N$$

The same correction has been made to Eq. (A3).

L613: "maximing" -> "maximising"
Corrected.